# Nearest-Neighbor Imputation with Error Guarantees and Extensions for Mixed-Type Data and Joint Learning

## Abstract

Missing feature values are pervasive in real-world applications, and remain a significant hurdle for downstream machine-learning tasks such as classification. Imputation methods combined with downstream tasks are often also time-consuming for high-dimensional data, and offer few theoretical guarantees on imputation error, especially for not-missing-at-random mechanisms. We first show that (weighted) nearest-neighbor approaches remain competitive on real-world data sets compared to the state-of-the-art, while being orders of magnitude faster. Second, we derive a novel concentration inequality from which we obtain theoretically-supported bounds on the imputation error for several types of missingness mechanisms in nearest-neighbor algorithms. Third, we show that nearest-neighbor algorithms can be adapted to mixed-type imputation and extended to joint training with downstream tasks by introducing a data-distribution-preserving function and tuning the weights with an online learner. We validate our theoretical bounds on synthetic data sets, and empirical results on nine real-world data sets. This paper demonstrates the strength of nearest-neighbor imputation and opens the way towards more theoretically-backed approaches for imputation.

## 1 Introduction

Most machine-learning approaches assume full access to the features of the input data points. However, missing values might arise due to the incompleteness of public databases or measurement errors. Research on the imputation of missing values and inference on missing data is motivated by the fact that naive approaches would not fare well. Indeed, ignoring samples with missing values might lead to severe data loss and meaningless downstream models, for classification or regression (Liao et al., 2014; Shadbahr et al., 2023). Yet, replacing missing values with zeroes (or any "simple" univariate approach such as taking the mean or the median) can considerably distort the distribution of data values, as there is a more significant weight on the default value for missing entries, and then perhaps bias the training of a downstream model for classification or regression tasks (Khan & Hoque, 2020). Nonetheless, multivariate approaches are often time-consuming and prohibitive for high-dimensional data sets, as in biology (Brini & van den Heuvel, 2024; Gu et al., 2025).

The literature often distinguishes three main categories of missingness mechanisms (Rubin, 1976) depending on the relationship between the probability $p^{\text{miss}}$ of a missing value and the data. The simplest one is Missing-Completely-At-Random (MCAR), where that probability is independent of the data. This can be applied when the measurement tools fail at some probability, regardless of the analyzed data. The second, more complex, setting is Missing-At-Random (MAR), where $p^{\text{miss}}$ depends solely on the observed (not missing) data. An example of MAR is when male patients drop out more often from a clinical study than female patients. Finally, the Missing-Not-At-Random setting (MNAR), where $p^{\text{miss}}$ depends on both the observed and missing data, is widely regarded as the most challenging setting for analysis because the actual values might not be identifiable.

As previously mentioned, the fastest approaches to imputation are often univariate because they are simple operations applied feature-wise to a dataset. For instance, the missing value for a feature corresponding to a column of the data matrix might be replaced by the mean or the median of all non-missing values or even by

zeroes in that column. Yet, such naive approaches can severely distort the distribution of values (Morvan & Varoquaux, 2025).

The risk of distorting distributions with naive approaches opened the path to multiple multivariate methods, such as MICE (van Buuren & Groothuis-Oudshoorn, 2011), MissForest (Stekhoven & Bühlmann, 2012), RF-GAP (Rhodes et al., 2023) resorting to random forests; MIDAS (Seu et al., 2022) using denoising auto-encoders; Optimal Transport-based algorithms (Muzellec et al., 2020a); but also matrix factorizations (Mazumder et al., 2010), penalized logistic regression methods (van Loon et al., 2024), Bayesian network-based approaches (for instance, MIWAE (Mattei & Frellsen, 2019) for MAR mechanisms and its MNAR counterpart not-MIWAE (Ipsen et al., 2021)). Some recent works also provide a pipeline for the automated finetuning and refinement of imputers, such as MIRACLE (Kyono et al., 2021) or HyperImpute (Jarrett et al., 2022). We dwell further on newer diffusion model and deep learning-based approaches in Appendix (Section A). However, as the number of features increases, so does the computation time, making most of those approaches intractable on practical data sets. For instance, in genomic data, the feature set (genes) can amount to as many as $20,000$ genes in humans.

Moreover, many published imputation methods come without any guarantee on the quality of the imputation or often on the more straightforward settings such as MCAR (Mazumder et al., 2010) and MAR (Śmieja et al., 2018); with a few exceptions such as (Tang et al., 2003; Mohan et al., 2018; Sportisse et al., 2020) for pure imputation tasks, and NeuMiss networks (Le Morvan et al., 2020), which tackle a classification task in the presence of missing values. However, the MCAR and MAR settings are usually not applicable to real-world data, and (Tang et al., 2003; Mohan et al., 2018; Sportisse et al., 2020) rely on an assumption of data generation through low-rank or linear random models instead of simpler data distributions.

Finally, nearest-neighbor imputers (Troyanskaya et al., 2001) are known to have good performance in practice and are relatively fast (Emmanuel et al., 2021; Seu et al., 2022; Joel et al., 2024). This comes at the price of some distortion in high-dimensional data sets (Beretta & Santaniello, 2016). This observation led us to consider an improvement of a nearest-neighbor imputation that preserves the data distribution even in larger dimensions while remaining computationally fast.

We first propose a general proof strategy for deriving theoretical bounds on the imputation quality for a general description of nearest-neighbor approaches for several types of missingness mechanisms (Section 2). Then, we describe a novel nearest-neighbor algorithm named Fast Iterative Improvement for Imputation (F3I) (Section 3), which can be used to automatically tune neighbor-dependent weights to a specific downstream task and take into account mixed-type variables (Section 3.1). Finally, we show in our experiments that nearest-neighbor approaches are competitive performance-wise to the state-of-the-art, while being much faster, which is an aspect often overlooked for real-world applications (Section 4).

Throughout the paper, we denote $N$, $F$ and K the number of samples, features, and nearest neighbors. $i$, $j$, $f$ are integer indices. For all other alphabet letters ($x$, $z$, etc.), $v$ is a scalar, $\boldsymbol{v}$ a vector, and $V$ a matrix. $\boldsymbol{v}^i$ is the $i^{th}$ column and $\boldsymbol{v}_j$ is the $j^{th}$ row of matrix $V$, and $v_i^j$ is the coefficient at position $(i, j)$ in $V$ for $i, j > 0$. For any $K \geq 2$, $\triangle_K \triangleq \{\boldsymbol{p} \in [0, 1]^K \mid \sum_{k \leq K} p_k = 1\}$ is the simplex of dimension K. The initial data matrix with missing values is $X \in (\mathbb{R} \cup \{\texttt{NaN}\})^{N \times F}$, where $X^\star \in \mathbb{R}^{N \times F}$ is the full (unavailable) data matrix. Finally, $m_i^f \in \{0, 1\}$ is the random variable that indicates whether the value at position $(i, f)$ is missing in the input data matrix, where $m_i^f = 1$ means it is missing.

## 2 Theoretical guarantees for nearest-neighbor imputation algorithms

We propose a pseudo-code for a weighted nearest-neighbor (NN) imputation meta-algorithm with convex weights for which our proof strategy and analysis hold (Algorithm 1). Given any weight update function returning a vector of K convex weights in $\triangle_K$, and a stopping criterion, this meta-algorithm iteratively retrieves the K nearest neighbors of each data point, and imputes missing values with the convex combinations of corresponding non-missing feature values from nearest-neighbors. This structure retrieves known NN-based imputation approaches (Troyanskaya et al., 2001). For instance, the classical K-NN imputation algorithm with uniform weights just stops after one iteration, and the weight function produces the K-sized vector $1/K\mathbf{1}_K$.

---

**Algorithm 1** General weighted nearest-neighbor (NN) imputation meta-algorithm.

---

**Input:** Data $X \in (\mathbb{R} \cup \{\texttt{N/A}\})^{N \times F}$
**Parameters:** Number of neighbors $K \geq 2$, Weight update function $f_w : \triangle_K \times \mathbb{R}^{N \times F} \to \triangle_K$, Stopping criterion $s : \triangle_K \times \mathbb{R}^{N \times F} \to \{\text{True}, \text{False}\}$, Nearest-neighbor search algorithm $\mathcal{A}$
**Output:** Imputed data $\widehat{X} \in \mathbb{R}^{N \times F}$
Initialize $\mathcal{A}$ with $Z = \{\boldsymbol{x}_1, \boldsymbol{x}_2, \ldots, \boldsymbol{x}_N\}$ and $\widehat{X} \leftarrow X$       # Initialize NN search
$\boldsymbol{\alpha} \leftarrow 1/K \mathbf{1}_K \triangleq \{1/K, 1/K, \ldots, 1/K\}$       # Initialize the K neighbor-dependent weights
$t \leftarrow 0$
**while** not $s(\boldsymbol{\alpha}, \widehat{X}, t)$ **do**
    $\boldsymbol{\alpha} \leftarrow f_w(\boldsymbol{\alpha}, \widehat{X})$       # Update the K neighbor-dependent weights
    **for** $i \leq N$ and $f \leq F$ **do**
        $(\boldsymbol{z}^{i,f})_1, (\boldsymbol{z}^{i,f})_2, \ldots, (\boldsymbol{z}^{i,f})_K \leftarrow \mathcal{A}(\widehat{\boldsymbol{x}}_i)$ and $\widehat{x}_i^f \leftarrow \sum_{k \leq K} \alpha_k (z^{i,f})_k^f$ **if** $m_i^f = 1$, $\widehat{x}_i^f$ otherwise
    **end for**
    $t \leftarrow t + 1$
**end while**
**return** $\widehat{X}$

---

One of the most common metrics to evaluate the imputation quality is the Mean Squared Error (MSE).

**Definition 2.1.** Mean squared error. *We define the mean squared error as $MSE(\widehat{\boldsymbol{x}}, \boldsymbol{x}^\star) = 1/F \sum_{f \leq F} (\widehat{x}_i^f - (x^\star)_i^f)^2$ and the associated loss as $\mathcal{L}^{\text{MSE}}(\widehat{X}, X^\star) \triangleq 1/N \sum_{i \leq N} \text{MSE}(\widehat{\boldsymbol{x}}_i, \boldsymbol{x}_i^\star)$.*

*The root-mean-squared error (RMSE) is then defined as $\mathcal{L}^{RMSE}(\widehat{X}, X^\star) \triangleq \sqrt{\mathcal{L}^{MSE}(\widehat{X}, X^\star)}$.*

Note that none of the considered imputation approaches in our experimental study in Section 4 have access to the ground truth values, nor need to compute the mean squared error during training. However, we can still derive useful properties of NN imputation approaches on the MSE.

## 2.1 Novel concentration bound on the difference between NN-imputed and true points

We now informally state our assumptions about the data generation procedure to derive theoretical guarantees. Formal statements can be found in Appendix (Section B and Algorithm 4). Note that the algorithms still work well in practice, even without those assumptions being satisfied, and as reported in the literature and in Section 4, NN approaches can be successfully applied to real-world data.

We provide an analysis of NN imputation algorithms for three types of missingness mechanisms: a MCAR mechanism where the random indicator variables are drawn from a Bernoulli distribution with fixed mean (Assumption B.1); a MAR mechanism where the probability of missingness depends only on observed values of the data set (Assumption B.2); and, finally, a MNAR mechanism called Gaussian self-masking (Assumption 4 from Le Morvan et al. (2020)) where the probability of $m_i^f = 1$ is proportional to $\exp(-\zeta^{-2}((x^\star)_i^f - \mu_f)^2)$ where $\mu_f$ is specific to feature $f$ and $\zeta$ is fixed (Assumption B.3). In addition, we rely on assumptions about the data. First, we assume that each value in the full data matrix is drawn from independent fixed-variance Gaussian distributions (Assumption B.4). Second, the random indicator variables $m_i^f$ are then independently drawn according to the missingness mechanism with probability $p^{\text{miss}}$. If $m_i^f = 1$, then the covariate at position $(i, f)$, $x_i^f$ in $X$ is unavailable, otherwise, $x_i^f = (x^\star)_i^f$. Third, we also ensure that there are at least $K$ neighbors for every data point (Assumption B.5), and that we know a constant upper bound on the norm of any feature vectors (Assumption B.6).

The following novel concentration inequality deals with the norm of the difference between any complete data point $(\boldsymbol{x}^\star)_i$ and *any* NN-based imputed data point $\widehat{\boldsymbol{x}}_j$ with a K-nearest neighbor imputation approach. We defer the proof to Appendix F, which leverages subgaussian random variables to control the tail decay.

**Corollary 2.2.** Concentration bound on the norm of the difference between $\widehat{\boldsymbol{x}}_j$ and $(\boldsymbol{x}^\star)_i$. *Under all assumptions listed previously, for $0 < \delta \leq 1/N$ and a quantity $C_\delta^{miss} > 0$ such that, as $\delta$ tends to 0,*

$C_\delta^{miss} = \mathcal{O}((\sigma^{miss})^2 F + \ln\frac{1}{\delta})$, where $\sigma^{miss}$ is linked to the variance of the data distribution and depends on the missingness mechanism, $\|\widehat{\boldsymbol{x}}_j - (\boldsymbol{x}^\star)_i\|_2^2 \leq C_{\delta/N^2}^{miss}$ for all $i,j \leq N$ with probability $1-\delta$.

## 2.2 General proof strategy for controlling the imputation error in NN algorithms

Provided any concentration bound on the norm of the difference between NN-imputed and complete points, we can now derive guarantees on the imputation quality.

**Theorem 2.3.** Bounds in high probability and in expectation on the MSE for Algorithm 1. *Under Assumptions B.4-B.6 and under any of the missingness mechanisms described in Assumptions B.1-B.3, if $\widehat{X}$ is a NN imputed matrix, and $X^\star$ is the corresponding complete matrix (unavailable in practice),*

$$\mathbb{P}\left(\mathcal{L}^{MSE}(\widehat{X}, X^\star) \leq \mathcal{O}((\sigma^{miss})^2 + \ln N/F)\right) \geq 1 - 1/N \ ,$$

*where $\sigma^{miss}$ is linked to the variance of the data distribution and depends on the missingness mechanism.*

*Proof.* First, we denote $\mathcal{K}(\boldsymbol{x}, X, f, k)$ the index of the $k^{\text{th}}$ nearest neighbor (identified by a nearest neighbor search algorithm $\mathcal{A}$) to vector $\boldsymbol{x}$ on feature $f$ among the rows of $X$, that is, $\{\boldsymbol{x}_1, \boldsymbol{x}_2, \ldots, \boldsymbol{x}_N\}$. We recall that for any imputation step, $(\widehat{x})_i^f \triangleq (x^\star)_i^f$ if $m_i^f = 0$, $\sum_{k \leq K} \alpha_k \widehat{x}_{\mathcal{K}(\widehat{\boldsymbol{x}}_i, X, f, k)}^f$ otherwise. Then, for any $i \leq N$,

$$
\begin{aligned}
\mathrm{MSE}(\widehat{\boldsymbol{x}}_i, \boldsymbol{x}_i^\star) &= \frac{1}{F} \sum_{\substack{f \leq F \\ m_i^f = 1}} \left(\sum_{k \leq K} \alpha_k \widehat{x}_{\mathcal{K}(\widehat{\boldsymbol{x}}_i, X, f, k)}^f - (x^\star)_i^f\right)^2 = \frac{1}{F} \sum_{\substack{f \leq F \\ m_i^f = 1}} \left(\sum_{k \leq K} \alpha_k \left(\widehat{x}_{\mathcal{K}(\widehat{\boldsymbol{x}}_i, X, f, k)}^f - (x^\star)_i^f\right)\right)^2 \\
&\leq \frac{1}{F} \sum_{\substack{f \leq F \\ m_i^f = 1}} \sum_{k \leq K} \alpha_k \left(\widehat{x}_{\mathcal{K}(\widehat{\boldsymbol{x}}_i, X, f, k)}^f - (x^\star)_i^f\right)^2 = \frac{1}{F} \sum_{k \leq K} \alpha_k \sum_{\substack{f \leq F \\ m_i^f = 1}} \left(\widehat{x}_{\mathcal{K}(\widehat{\boldsymbol{x}}_i, X, f, k)}^f - (x^\star)_i^f\right)^2 \ ,
\end{aligned}
$$

where the inequality is obtained by the Jensen's inequality on convex function $x \mapsto x^2$ with $\sum_{k \leq K} \alpha_k^t = 1$.

**Case 1: First imputation step.** $\widehat{\boldsymbol{x}}_i$ has missing values, and possibly the $k^{\text{th}}$ neighbor $\mathcal{K}(\boldsymbol{x}, X, f, k)$ of data point $\widehat{\boldsymbol{x}}$ with respect to feature $f$ might be different for a different feature $f'$ if the data point corresponding to index $\mathcal{K}(\boldsymbol{x}, X, f, k)$ has a missing value for feature $f'$. We define the $k^{\text{th}}$ *pseudo-neighbor* of data point $\widetilde{\boldsymbol{x}}_i$ as $\widetilde{x}_i^f \triangleq (x^\star)_i^f$ if $m_i^f = 0$, $\widetilde{x}_i^f \triangleq \sum_{k \leq K} \alpha_k \widehat{x}_{\mathcal{K}(\widehat{\boldsymbol{x}}_i, X, f, k)}^f$ otherwise. Going back to the derivation on the MSE

$$\mathrm{MSE}(\widehat{\boldsymbol{x}}_i, \boldsymbol{x}_i^\star) \leq \frac{1}{F} \sum_{k \leq K} \alpha_k \sum_{\substack{f \leq F \\ m_i^f = 1}} \left(\widehat{x}_{\mathcal{K}(\widehat{\boldsymbol{x}}_i, X, f, k)}^f - (x^\star)_i^f\right)^2 = \frac{1}{F} \sum_{k \leq K} \alpha_k \|\widetilde{\boldsymbol{x}}_i - (\boldsymbol{x}^\star)_i\|_2^2 \leq \frac{1}{F} C_{1/N^3}^{\mathrm{miss}} \ ,$$

with probability $1 - 1/N$, by applying Corollary 2.2 with $\delta = 1/N$ and $\sum_{k \leq K} \alpha_k = 1$, where $C_{1/N^3}^{\mathrm{miss}} = \mathcal{O}((\sigma^{\mathrm{miss}})^2 F + \ln N)$. This leads to $\mathbb{P}\left(\mathcal{L}^{\mathrm{MSE}}(\widehat{X}, X^\star) \triangleq 1/N \sum_{i \leq N} \mathrm{MSE}(\widehat{\boldsymbol{x}}_i, \boldsymbol{x}_i^\star) \leq \frac{1}{F} C_{1/N^3}^{\mathrm{miss}}\right) \geq 1 - 1/N$.

**Case 2: Subsequent imputation steps.** That is, $\widehat{\boldsymbol{x}}_i$ has no longer any missing values, since they have been imputed in the previous step. That means in particular that the identity of neighbors by the NN search algorithm $\mathcal{A}$ does not depend on the feature, *i.e.*, $\mathcal{K}(\boldsymbol{x}, X, f, k) = \mathcal{K}(\boldsymbol{x}, X, f', k) = \mathcal{K}(\boldsymbol{x}, X, k)$ for any $f \neq f'$. We can then use the derivation from Case 1 for $\widetilde{\boldsymbol{x}}_i \leftarrow \widehat{\boldsymbol{x}}_{\mathcal{K}(\boldsymbol{x}, X, f, k)}$. □

In particular, this theorem means that the imputation quality in any NN imputation algorithm described in Algorithm 1 decreases linearly with the variance in the data, which is what we expect, as convex imputations would hardly be able to generate outlier data points as empirically shown in Section 4. Note that this proof does not rely on a specific definition of nearest neighbors (*e.g.*, with the Chebyschev or Euclidean distance) nor a specific weight function. This means that our theoretical results will still hold for the extensions of NN algorithms described in the next section, so long as the data satisfies our assumptions.

# 3 Extensions of NN imputation algorithms

An additional perspective on imputation is how well the imputed values align with a fixed downstream task. The most common use case for imputation is regularizing data so that the results of appropriate downstream analysis on the imputed data are similar to if the full data was available (Bertsimas et al., 2018). However, Morvan & Varoquaux (2025) show that simply improving imputation often offers limited benefits for downstream predictive performance. We thus design a method called F3I (Fast Iterative Improvement for Imputation) in Section 3.1 that allows us to optimize kNN-based imputation for both data consistency and a downstream task of choice, by appropriately scaling and adding the downstream loss to a novel joint imputation-downstream task objective function, as well as performing gradient surgery to aid convergence when the losses are misaligned. Moreover, a well-known limitation of nearest-neighbor imputation algorithms is that they are restricted to numerical values, whereas one might be interested in imputing categorical data as well. We describe in Section 3.2 how to extend further F3I to mixed-type imputation.

In both extensions, the key idea is that we would like to replace missing positions in $X$ with the most probable values by iteratively applying a "good" *weighted combination of elements in the data set*, starting from a guess, *e.g.*, through K-nearest neighbor imputation. For each value $x_i^f$, we are looking to tune the weights $\boldsymbol{\alpha} = (\alpha_1, \alpha_2, \ldots, \alpha_K)$ of a convex combination of the K closest neighbors of $\widehat{\boldsymbol{x}}_i$ in a reference set $Z \in \mathbb{R}^{N \times F}$ without missing values, $\boldsymbol{z}_{i(1)}, \boldsymbol{z}_{i(2)}, \ldots, \boldsymbol{z}_{i(K)}$, ordered by their increasing Chebyshev distance to $\widehat{\boldsymbol{x}}_i$. When the reference set is obvious, we define $\boldsymbol{x}(\boldsymbol{\alpha}) \triangleq \sum_{k \leq K} \alpha_k \boldsymbol{z}_{i(k)}$.

## 3.1 Joint NN imputation with a downstream task

Ideally, if we had access to the true distribution $\mathcal{D}$ on feature vectors, we would like to set the weights $\boldsymbol{\alpha} \in \triangle_K$, the simplex of dimension K, such that the quantity $\mathbb{E}_{\boldsymbol{x} \sim \mathcal{D}} \left[ \mathbb{1}(\Phi_\Theta(\boldsymbol{x}(\boldsymbol{\alpha})) > \Phi_\Theta(\boldsymbol{z})) \right]$ is maximized, where $\mathbb{1}$ is the Kronecker symbol, $\boldsymbol{x}^0 \in \mathbb{R}^F$ is an initial guess on the missing values in $\boldsymbol{x} \in (\mathbb{R} \cup \{\texttt{NA}\})^F$, and $\Phi_\Theta$ is the parametrized true data distribution according to Assumption B.4. That is, we want to choose $\boldsymbol{\alpha}$ so that the imputed values are more probable than the current guesses. Considering a *full* data set of $N$ $F$-dimensional points $\mathcal{X} = \{\widehat{\boldsymbol{x}}_1, \widehat{\boldsymbol{x}}_2, \ldots, \widehat{\boldsymbol{x}}_N\} \subset \mathbb{R}^F$ of initial guesses on the missing values in $X$ and approximating the true distribution $\mathcal{D}$ by a density kernel $D_0$ on $Z$, we would like to maximize $\mathbb{1}\left(D_0(\widehat{\boldsymbol{x}}_i(\boldsymbol{\alpha})) > D_0(\widehat{\boldsymbol{x}}_i)\right)$ in $\boldsymbol{\alpha} \in \triangle_K$ for each sample $\boldsymbol{x}_i$, $i \leq N$. That quantity can be approximated by

$$\max \left( 0, \frac{D_0(\widehat{\boldsymbol{x}}_i(\boldsymbol{\alpha}))}{D_0(\widehat{\boldsymbol{x}}_i)} - 1 \right) \approx \log \left( \frac{D_0(\widehat{\boldsymbol{x}}_i(\boldsymbol{\alpha}))}{D_0(\widehat{\boldsymbol{x}}_i)} \right) ,$$

We define for any $\boldsymbol{\alpha} \in \triangle_K$, $X \in \mathbb{R}^{N \times F}$ and $\eta \geq 0$, the function $G$ defined as

$$G : \boldsymbol{\alpha}, X \mapsto 1/N \sum_{i \leq N} \log D_0(\boldsymbol{x}_i(\boldsymbol{\alpha}))/D_0(\boldsymbol{x}_i) - \eta \|\boldsymbol{\alpha}\|_2^2 . \tag{1}$$

An intuitive interpretation of $G$ is that if $G(\boldsymbol{\alpha}, X) \leq 0$, then the imputed points with $\boldsymbol{\alpha}$ are, on average, less probable than the previous imputations. We now show that $G$ can be maximized through standard convex optimization techniques since $G$ is concave, detailed in the Appendix (Section C). Moreover, another interesting property of $G$ is that its gradient is Lipschitz-continuous with respect to $\boldsymbol{\alpha}$ (Proposition C.5), which allows us to derive theoretical guarantees when $G$ is combined with a loss function of a downstream task (Section 3.1). Then the simplest approach to imputation based on $G$ (Equation 1) consists of first imputing the missing values with K-nearest neighbors with uniform weights, and then iteratively improving the imputed values by fine-tuning the weights to maximize $G$. Note that the neighbors might change for the same initial sample across iterations. However, solving a full convex optimization problem at each iteration might be time-consuming. Similarly to prior works in other research fields (Degenne et al., 2020), we advocate for learning the optimal weight vector on the fly by resorting to an online learner. This permits the leverage of powerful online learners from the literature, for instance, AdaHedge (De Rooij et al., 2014) or EXP3 (Auer et al., 2002), to obtain theoretical guarantees while having a computationally fast imputation. Those two ingredients are the keys to our kNN extension F3I, described in Algorithm 2.

As noticed by several prior works (Le Morvan et al., 2021; Morvan & Varoquaux, 2025; Vo et al., 2024), a good imputation quality does not necessarily go hand in hand with an improved performance in a downstream task

---

**Algorithm 2** The Fast Iterative Improvement for Imputation (F3I) algorithm.

---

**Parameters:** Maximum budget $T > 0$
$\widehat{X} \leftarrow \texttt{KNN\_imputer}(X, \text{weights}=1/K\mathbf{1}_K)$
\# Define the weight function
**def** $f_{\text{F3I}}(\boldsymbol{\alpha}, \widehat{X}, \mathcal{L} = (0, 0, \ldots, 0) \in \mathbb{R}^K)$:                    \# Initialize the AdaHedge learner
    Impute $\widehat{X}$ with the uniformly weighted NN approach with Chebyschev distance (see Algorithm 1)
    Update $\mathcal{L}$ with the loss $-\langle \boldsymbol{\alpha}, \nabla_{\boldsymbol{\alpha}} G(\boldsymbol{\alpha}, \widehat{X}) \rangle$                    \# Update the online learner $\mathcal{L}$
    Return $\boldsymbol{\alpha} \leftarrow \mathcal{L}$                    \# Get the new weight vector
\# Define the stopping criterion
**def** $s(\boldsymbol{\alpha}, \widehat{X}, t)$ :
    Return $((t > T) \text{ or } (G(\boldsymbol{\alpha}, \widehat{X}) \leq 0))$
\# Run the K-NN imputation algorithm
Run Algorithm 1 with $f_w \leftarrow f_{\text{F3I}}$, $s \leftarrow s_{\text{F3I}}$ and $X \leftarrow \widehat{X}$

---

run on the imputed data set, *e.g.*, for classification (Le Morvan et al., 2021), regression (Ayme et al., 2023), or structure learning (Vo et al., 2024). That might explain why, in some cases, data sets imputed with naive constant imputations that are known to distort the initial data distribution might yield better performance metrics than those with more sophisticated approaches (Morvan & Varoquaux, 2025). In addition to the imputation algorithm in Algorithm 2, we propose a generic approach that optimizes both for an imputation task and a specific downstream task, by learning the optimal (convex) imputation pattern for some model parameters. Assuming that there is a convex, differentiable pointwise loss function $\ell$ for the downstream task, we now consider the maximization problem $\max_{\boldsymbol{\alpha} \in \triangle_K} \mathcal{G}(\boldsymbol{\alpha}, X; \beta)$ on $X \in \mathbb{R}^{N \times F}$ on $\boldsymbol{\alpha}$, where

$$\mathcal{G}(\boldsymbol{\alpha}, X; \beta) \triangleq (1 - \beta)G(\boldsymbol{\alpha}, X) - \beta/N \sum_{i \leq N} \ell(\boldsymbol{x}_i(\boldsymbol{\alpha})) , \tag{2}$$

where $\beta \in [0, 1]$ is a positive regularization parameter related to the importance of the downstream task. As reported in many papers on multi-task learning (Chen et al., 2018; Yu et al., 2020; Liu et al., 2021), simply replacing the gradient of $G$ by the weighted sum of the gradient of $G$ and $\ell$ might lead to optimization issues, for instance, stalling update due to orthogonal gradients. This issue is solved by using PCGrad (Yu et al., 2020) to perform gradient surgery to align gradients. We call PCGrad-F3I this joint training version of F3I obtained by replacing $G$ in the loss function of the online learner in Algorithm 2 by $\mathcal{G}$. In addition to the high-probability upper bound derived on the MSE loss in Theorem 2.3, we can derive the following theoretical upper bound on imputation-downstream task performance with respect to $\mathcal{G}$. This upper bound again relies on the concentration bound (Lemma 2.2) derived in the previous section.

**Theorem 3.1.** High-probability upper bound on the joint imputation-downstream task performance. *Under Assumptions B.4-B.6, for any $X \in (\mathbb{R} \cup \{\textit{N/A}\})^{N \times F}$, convex pointwise loss $\ell$ [1] such that $\nabla \ell$ is Lipschitz-continuous, and $\beta \in [0, 1]$, under the conditions in Theorem 2 from (Yu et al., 2020), $\max_{\boldsymbol{\alpha} \in \triangle_K} \sum_{s=1}^{t} \mathcal{G}(\boldsymbol{\alpha}, X^{s-1}; \beta) - \mathcal{G}(\boldsymbol{\alpha}^s, X^{s-1}; \beta) \leq C_{(G,\ell)}^{AH} \sqrt{t} + (1 - \beta)H^{miss}h^{-1}t$ w.h.p. $1 - 1/N$, where $H^{miss} = \mathcal{O}(F + \ln N)$ depends on the missingness mechanism, $h$ is chosen to guarantee that $G$ is concave, and $C_{(G,\ell)}^{AH}$ is the constant related to AdaHedge being applied with gains $\overline{g}_s(\cdot)$.*

Finally, note that nearest neighbor search $\mathcal{A}$ in NN-based imputation approaches might be costly when the number of samples is very large. We suggest (and implement in practice) an approximate neighbor-finding algorithm such as FAISS (Johnson et al., 2019), LSH (Zhao et al., 2014; Tsai & Yang, 2014) or Annoy (Bernhardsson, 2018) and leverage the use of GPUs to accelerate F3I. Here, we use FAISS.

### 3.2 Mixed-type imputation with a NN algorithm

Like other NN-based approaches, F3I (Algorithm 2) can only be applied to numerical variables, as the imputation procedure features a convex sum of values from nearest neighbors. However, the general idea behind F3I can actually be adapted to mixed-type data, *i.e.*, data with categorical and continuous variables.

---

[1] Such a loss function exists for classification, and we give an example in appendix (Example 1).

---

**Algorithm 3** Imputation in the F3I algorithm on mixed-type variables.

---

> **Input:** K-nearest neighbors $(\boldsymbol{z}^{i,f})_1, \ldots, (\boldsymbol{z}^{i,f})_K$ of data point $\widetilde{\boldsymbol{x}}_i$ with respect to missing feature $f$, weight vector $\boldsymbol{\alpha} \in \triangle_K$
> **Output:** New value of $\widetilde{x}_i^f$
> **if** $f$ is a categorical variable **then**
>     $v_1, v_2, \ldots, v_\ell$ are the unique values among $(\boldsymbol{z}^{i,f})_1, \ldots, (\boldsymbol{z}^{i,f})_K$
>     $s_1, \ldots, s_\ell \leftarrow 0, \ldots, 0$
>     **for** $u = 1, 2, \ldots, \ell$ **do**
>         $s_u \leftarrow \sum_{k \leq K} \alpha_k \delta\left( (\boldsymbol{z}^{i,f})_k^f = v_u \right)$
>     **end for**
>     $u^\star \leftarrow \arg\max_{u \leq \ell} s_u$                 # Ties are broken arbitrarily
>     $x \leftarrow v_{u^\star}$                       # Assign the value with highest cumulative weight
>     **else** $x \leftarrow \sum_{k \leq K} \alpha_k (z^{i,f})_k^f$          # Regular convex imputation
> **end if**
> **return** $x$

---

First, mixed-type nearest-neighbor search relies on the Gower's distance (Gower, 1971) instead of the Chebyshev distance. This allows F3I to compute the K nearest neighbors even when the point features categorical variables. Given two $F$-dimensional points $\boldsymbol{x}$ and $\boldsymbol{y}$, Gower's distance between those two points is $d^{\text{Gower}}(\boldsymbol{x}, \boldsymbol{y}) \triangleq F^{-1} \sum_{f \leq F} \text{dist}_f(x_f, y_f)$, where the distance function for a categorical variable $f$ is $\text{dist}_f(x, y) = \delta(x \neq y)$ and $\delta$ is the Kronecker symbol, whereas the distance function for a continuous variable $g$ is $\text{dist}_g(x, y) = |x - y|$. Second, the kNN imputation approach for continuous variables can naturally extend to categorical variables by adopting a consensus procedure. For each categorical value for a given variable in a given point, the weights of the K nearest neighbors to the point featuring that value are summed. Then, the categorical value with highest weight sum is selected as the imputed value for the considered feature of the current data point. The pseudocode related to this version of F3I, that we called MI-F3I, is shown in Algorithm 3. Due to the corresponding imputation not being continuous any longer, Theorem 3.1 does not hold for this version of F3I.

## 4 Experimental study

This section focuses on the empirical validation of the upper bound on the MSE loss in synthetic data sets (Theorem 2.3), the comparison of nearest neighbor-based imputation algorithms to baselines for imputation-only tasks, and comparing our extensions of nearest neighbor-based imputation to relevant baselines for the imputation-only and joint imputation-binary classification tasks, both on real-world data sets. In the appendix (Section G), in addition to presenting the full numerical tables for the experiments in this section, we also empirically validate the high-probability regret upper bounds Theorem 3.1 (and its counterpart for F3I without downstream task, which is Theorem D.1), shown in appendix (Section D). Further information about hyperparameter tuning and the computing infrastructure is also available in Section G.

Synthetic data is generated by fixing a value for the number of samples N (here, we selected $N = 50$) and the number of features of $F$ (here, $F = 100$), and generating a corresponding multivariate mean parameter $\boldsymbol{\mu}$ from a multivariate normal distribution $\mathcal{N}(\mathbf{0}_F, 10^{-2}\boldsymbol{I}_{F \times F})$. The different synthetic datasets are then generated from a multivariate normal distribution $\mathcal{N}(\mathbf{0}_F, \sigma^2 \boldsymbol{I}_{F \times F})$ for $\sigma = \{0.01, 0.1, 0.15, 0.2, 0.25, 0.5\}$. Missing data is then masked according to the missingness mechanism, as shown in Algorithm 4 in appendix. This synthetic data satisfies all our assumptions in Section B.

### 4.1 Empirical validation of error bounds for nearest neighbor approaches to imputation

We first empirically show that Theorem 2.3 is valid. We select as missingness rate $p^{\text{miss}} = 25\%$ and $K = 5$. We report in Figure 1 the MSE loss $\mathcal{L}^{\text{MSE}}(\widehat{X}, X^\star)$ (where $\widehat{X}$ is the last imputed data set in Algorithm 2) along with the corresponding $\sigma$-dependent upper bound $C^{\text{miss}} = \mathcal{O}((\sigma^{\text{miss}})^2 F + \ln N)$ depending on the value of $\sigma^{\text{miss}} \approx \sigma \in \{0, 0.1, 0.2, \ldots, 0.5\}$. The exact definition of $C^{\text{miss}}$ is in the full expression of the

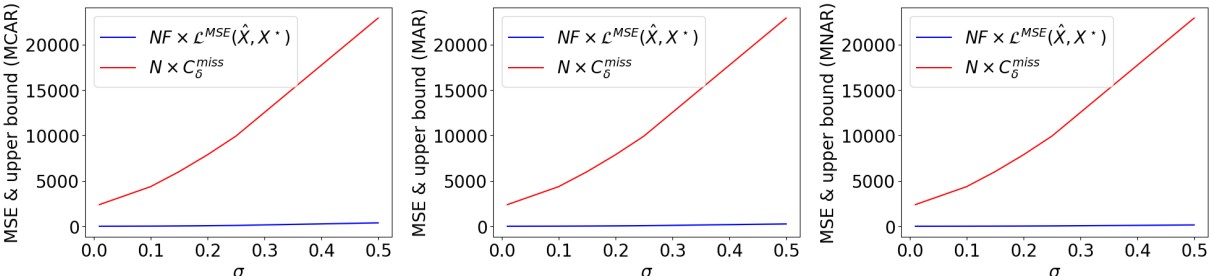

Figure 1: Empirical validation of Theorem 2.3 by comparing the value of the upper bound $N \times C^{\mathrm{miss}}$ and $NF \times \mathcal{L}^{\mathrm{MSE}}(X^t, X^\star)$ where $t$ is the final round for Algorithm 2 and $p^{\mathrm{miss}} = 25\%$. Left: MCAR setting. Center: MAR setting. Right: MNAR setting.

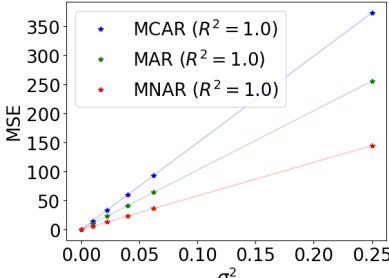

Figure 2: $NF \times \mathcal{L}^{\mathrm{MSE}}(X^t, X^\star)$ is linear in $\sigma^2$ regardless of the missingness mechanism.

concentration bound in Corollary F.6. Figure 1 confirms that the theoretical upper bound on the MSE loss is largely above the empirical MSE value for any value of $\sigma$, and similarly increases as $\sigma$ grows (see Table 3 in appendix). Moreover, $C^{\mathrm{miss}}$ recovers interesting dependencies as an upper bound of $\mathcal{L}^{\mathrm{MSE}}(X^t, X^\star)$. Indeed, we empirically observe that $\mathcal{L}^{\mathrm{MSE}}(\widehat{X}, X^\star)$ is roughly linear in $\sigma^2 \approx (\sigma^{\mathrm{miss}})^2$ regardless of the missingness mechanism (see Figure 2), which matches the dependencies shown in the high-probability upper bound given by Theorem 2.3. We also validate the bounds for different missingness rates and for other NN algorithms (uniform and distance-proportional weighted kNN) in the appendix.

### 4.1.1 Evolution of nearest neighbor-associated weights learnt during the F3I procedure

We compare NN-associated weights $\boldsymbol{\alpha}$ learnt on synthetic data (same as previously described) during the F3I procedure (Algorithm 2 applied to an imputation-only task) to weights in uniform or distance-dependent kNN algorithms. In the first round of F3I, the weights are uniform, and updated at each round by maximizing the

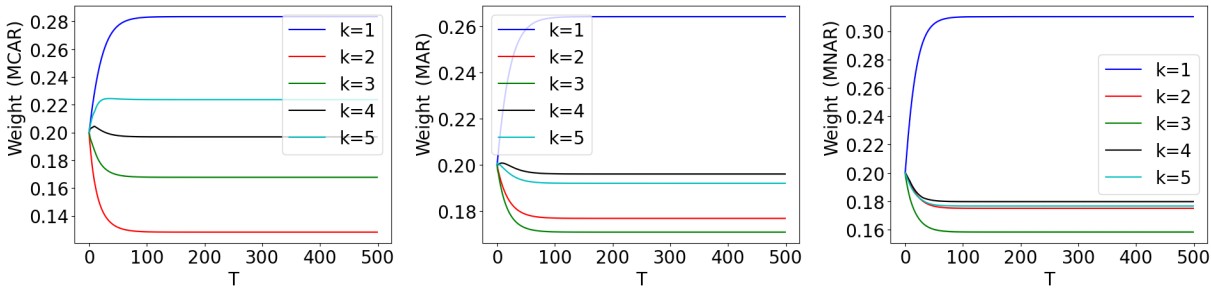

Figure 3: Evolution of the weight of each of the K-nearest neighbors for each sample as computed by F3I, where neighbor $k$ is the $k^{\mathrm{th}}$-nearest point, depending on the round $T$. Left: MCAR setting. Center: MAR setting. Right: MNAR setting.

value of function $G$. The relative weights point towards which neighbors are more important for imputation. Figure 3 displays the evolution of weight $\alpha_k$ for each $k$-nearest neighbor at each round $T$. Surprisingly enough, the optimal weight vector is completely different from the ones in uniform or distance-proportional k-NN algorithms, as it is not proportional to the rank of the neighbor; that is, the closer the neighbor, the higher the weight, which often motivates some heuristics about k-nearest neighbor algorithms. Optimality (preserving the data distribution) puts higher weights on the first and last closest neighbors. This observation suggests that the strategy that preserves the data distribution best would be to put the highest importance on the closest neighbor but mitigate this effect by putting a stronger weight on farthest neighbors, contrary to a strategy where all nearest neighbors have the same importance (uniform k-NN), hence not leveraging the similarities across neighbors, or where the closest ones have the highest weight (distance-proportional k-NN), which might lead to distribution bias.

## 4.2 Empirical performance of nearest neighbor-based imputers compared to other baselines

We also study the empirical performance in imputation tasks of nearest neighbor approaches compared to other baselines. We resort to the framework HyperImpute (Jarrett et al., 2022) to implement and run the benchmark for an imputation task across different performance metrics (including RMSE) on nine standard real-world data sets: BreastCancer (Wolberg et al., 1993), Diabetes (from scikit-learn (Pedregosa et al., 2011)), HeartDisease (Janosi et al., 1989), Ionosphere (selva86, 2024), Libras (Dias et al., 2009), Spambase (Hopkins et al., 1999), Wine Quality (Red and White) (Cortez et al., 2009) and a data set for drug repurposing, Gottlieb (Luo et al., 2016). In this benchmark, we included recent methods from the literature which benefited from open-source, modular `scikit-learn` (Pedregosa et al., 2011)-like implementations: along with the kNN implementations with uniform (kNN-u) and distance-dependent (kNN-d, where weights are inversely proportional to the distance to the neighbor) weights in (Pedregosa et al., 2011), and F3I (Section 3.1), and we also included baselines GAIN (Yoon et al., 2018a), GRAPE (You et al., 2020), HyperImpute (Jarrett et al., 2022), MIRACLE (Kyono et al., 2021), NewImp (Chen et al., 2024), Remasker (Du et al., 2023) and TDM (Zhao et al., 2023).

We first consider the MNAR scenario with a 30% missingness rate to add missing values, and report the RMSE averaged across 10 iterations (with different random seeds) across all data sets in Figure 4. These results show that nearest neighbor-based approaches are able to compete with the top contenders in the simple imputation task. These observations extend to other metrics (see Table 9 in the appendix). We also look at the performance of these imputation methods under different missingness rates and mechanisms.

Figure 5 shows the average RMSE over 10 iterations across missingness rates from 10%-90% and all three missingness mechanisms (MCAR, MAR, MNAR). We again observe that, similar to Figure 4, nearest neighbor approaches are able to compete with the top contenders across all missingness scenarios and missingness rates. Corresponding full numerical results are located in Table 12 in the appendix. We also evaluate on different missingness mechanisms in Table 11 in all nine datasets.

Most importantly, not only do nearest-neighbor imputation approaches have good performance compared to the state-of-the-art, but they also are faster by orders of magnitude than the top baseline HyperImpute (see the numerical results), which raises the question of computational cost-performance tradeoff in practical imputation tasks.

## 4.3 Empirical evaluation of the extension to downstream tasks

Additionally, we tested the performance of our extension to F3I on classification on missing data, by tackling both an imputation and a classification task. To do so, we chain an imputation model to a MLP classifier that returns logits, and either train separately or jointly the imputer and classifier depending on the imputation model. The considered loss is the log-loss function and sigmoid classifier $\ell(\boldsymbol{x}) \triangleq -y \log C_{\boldsymbol{\omega}}(\boldsymbol{x})$ in all datasets except Wine (Red), where we use the Cross Entropy loss $\ell(\boldsymbol{x}) \triangleq -\sum_{s=1}^{\mathcal{S}} y_s \log C_{\boldsymbol{\omega}}(\boldsymbol{x})_s$, where $y \in \{0, \ldots, \mathcal{S}\}$ is the class associated with sample $\boldsymbol{x} \in \mathbb{R}^F$.

We report the classification performance in terms of Area Under the ROC Curve (AUC) of several imputation models: *imputation methods with separate classifier training*: imputing by the mean (Mean), a K nearest-

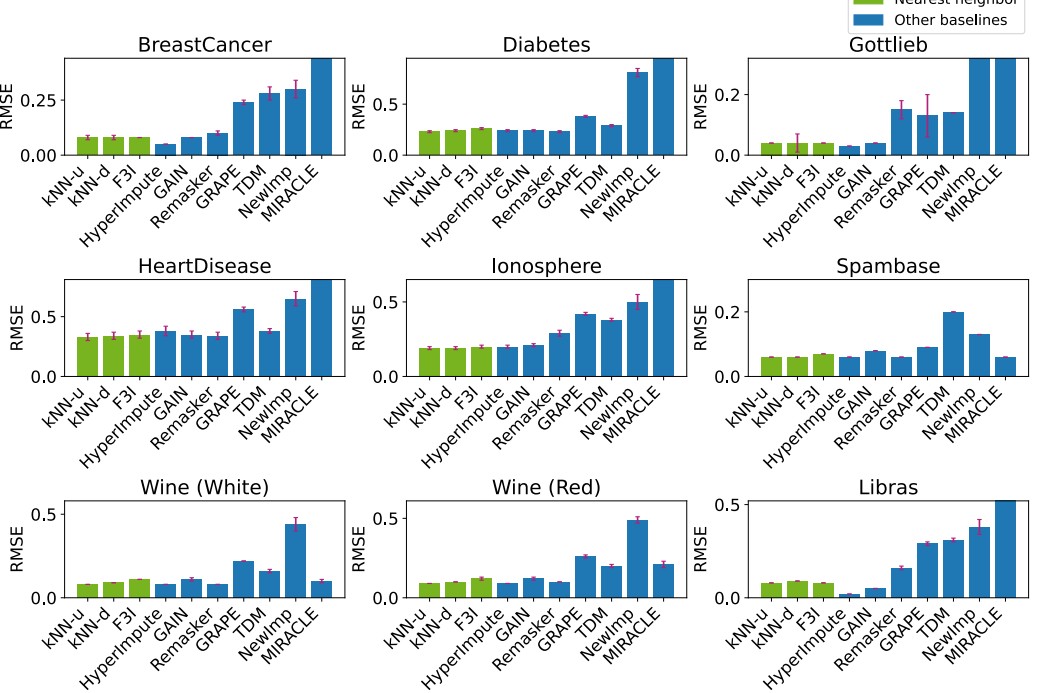

Figure 4: Average and standard deviation of RMSE across 10 iterations. MIRACLE vastly underperformed all baselines on BreastCancer, Diabetes, Gottlieb, Ionosphere and Libras, placing the top of its bar well beyond the scaled y-axis.

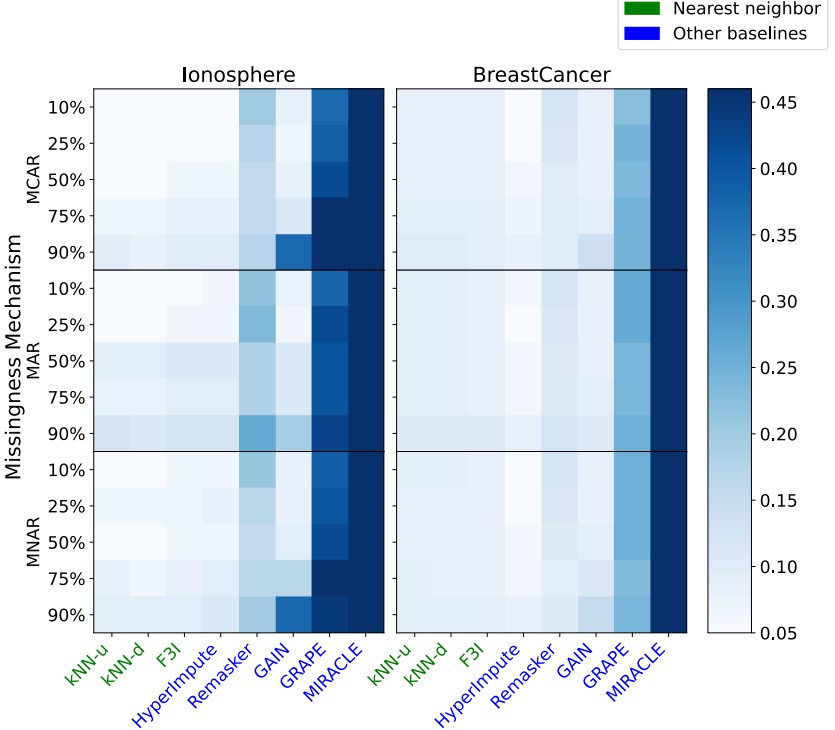

Figure 5: Average of RMSE score over 10 iterations on the Ionosphere and BreastCancer datasets for varying missing rate (in %) and missingness mechanisms (MCAR, MAR, MNAR). MIRACLE vastly underperformed in all scenarios, and is shown as solid blue block.

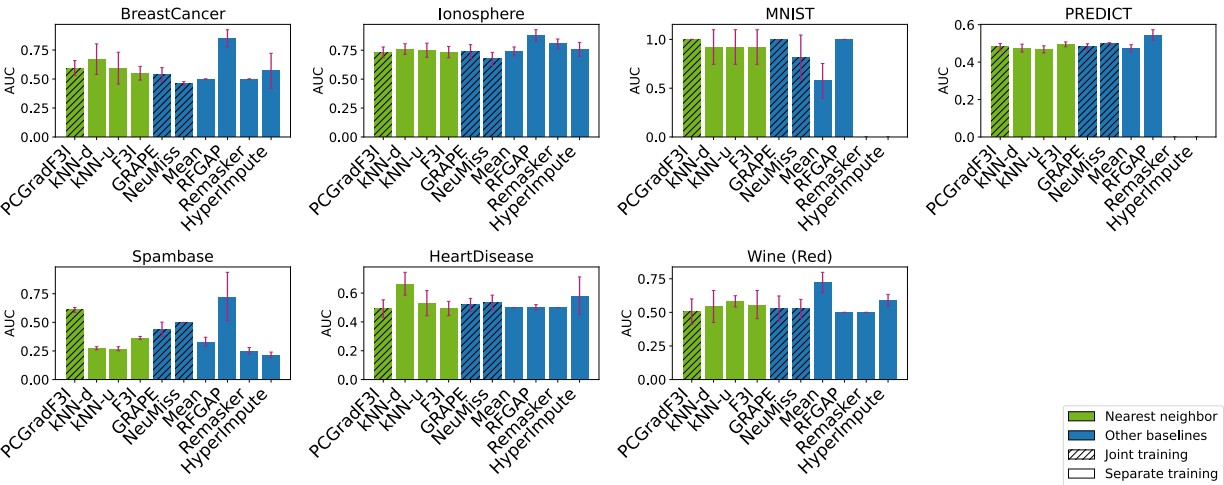

Figure 6: Average and standard deviation of Area Under the Curve (AUC) values across 20 iterations on the joint imputation-classification task (MCAR scenario, $p^{\mathrm{miss}} = 50\%$). *Remasker and HyperImpute are extremely slow when combined with an MLP on the largest datasets MNIST and PREDICT, which is why the values are missing.

neighbor algorithm with weights inversely proportional to the distance to neighbors (Troyanskaya et al., 2001), a random forest classifier (RF-GAP (Rhodes et al., 2023)), or one of the previous imputers (F3I, HyperImpute (Jarrett et al., 2022) and Remasker (Du et al., 2023)) prior to applying the MLP classifier; and *imputation methods with* joint *classifier training* like PCGradF3I, adding a NeuMiss block (Le Morvan et al., 2020) to the MLP classifier, or training simultaneously GRAPE (You et al., 2020) and the MLP classifier. We use the same MLP architecture across all imputation techniques. We split the samples into training (70%), validation (20%), and testing (10%) sets, where the former two sets are used for training the MLP and hyperparameter finetuning, and the AUC is computed on the latter set. We consider the MNIST dataset (LeCun et al., 1998), which we restrict to images annotated with class 0 or 1 to get a binary classification problem, and another drug repurposing data set, PREDICT (Réda, 2023). Moreover, we also include again the previous data sets for imputation, with their native classification labels. In all data sets, we remove features at random with probability 50% using a MCAR mechanism. Figure 6 displays the AUC values across 20 iterations with different random seeds.

The empirical performance of NN imputation algorithms is often comparable and sometimes better than state-of-the-art approaches such as Remasker and HyperImpute, whereas the performance of PCGradF3I (our extension of NN-based imputation to downstream tasks) is the best across joint-training approaches. Again, NN algorithms turn out to be competitive with respect to the performance-computation cost tradeoff.

### 4.4 Empirical evaluation of the extension to mixed-type imputation

We implemented MI-F3I, that is, the mixed-type extension of F3I, using the Python packages `kdtree` (Kögl, 2013) and `gower` (Yan, 2019) to implement a KD Tree compatible with Gower's distance. Table 1 reports the average and standard deviations of the Gower's distance computed between the imputed data set and the corresponding complete data set across 10 iterations. Missing values were added with an MCAR mechanism with frequency $p^{\mathrm{miss}} = 30\%$. On this data set, F3I achieves a significant improvement over HyperImpute across all data sets, which was the top baseline in our experiments on imputation-only tasks and can natively handle mixed-type data.

Table 1: Average and standard deviation of Gower's distance between the imputed data set and the corresponding complete dataset across 10 iterations (MCAR mechanism, $p^{\text{miss}} = 30\%$).

|  | Titanic | Airfoil | Blood | Compression | Iris |
|---|---|---|---|---|---|
| MI-F3I | **0.349 ±0.168** | **0.063 ±0.069** | **0.361 ±0.138** | **0.081 ±0.036** | **0.040 ±0.029** |
| HyperImpute | 0.488 ±0.144 | 0.137 ±0.081 | 0.486 ±0.128 | 0.136 ±0.045 | 0.085 ±0.092 |

## 5 Limitations

We highlight here some limitations of our contributions. First, NN-based algorithms might be prone to the curse of dimensionality. An approach to mitigate this effect is to apply (invertible) dimensionality reduction techniques prior to the imputation, for instance, using a Principal Component Analysis (PCA). Note that in this case, the assumption of independence in the data distribution for our theoretical results do not hold. Second, the theoretical guarantees derived in Theorems 2.3-3.1 require strong assumptions on the data distribution (*i.e.*, Gaussian distributed data), which may not accurately reflect the statistical properties of data sets in practical applications. Nonetheless, as shown in Section 4, NN-based imputation approaches can still be applied to real-world data and be competitive compared to the state-of-the-art. A interesting future work would be to extend our concentration bound to a larger set of data distributions and to non-continuous imputation procedures (such as in MI-F3I), allowing us to obtain similar high-probability upper bounds on the MSE loss.

## 6 Discussion

This paper studies on nearest-neighbor algorithms for imputation. First, we proposed a general proof strategy for deriving upper bounds on the mean-squared-error loss on imputed values, which we obtained through a new concentration bound on the norm between the imputed and corresponding complete data points. This result is valid for various missingness mechanisms, weight functions and neighbor search algorithms. Then, we extended nearest-neighbor-based imputation through the introduction of Fast Iterative Improvement for Imputation (F3I) (Section 3), which can be used to automatically tune neighbor-dependent weights to a specific downstream task and take into account mixed-type variables. derived high-probability guarantees on the optimization of the underlying data distribution-preserving function. Finally, our experimental study highlights the good performance-computational cost tradeoff achieved by nearest-neighbor algorithms for imputation-only and joint imputation-classification tasks despite their simplicity, while empirically confirming our error bounds. The code used in the experiments is provided as supplementary material.

Combining online learning and density ratio estimation is a simple and flexible idea that could be improved further. For instance, the density ratio estimation step might benefit from the classifier-based approach developed in BORE (Tiao et al., 2021). One might also wonder whether imputation could be improved by considering sample-specific weights, as, currently, $\boldsymbol{\alpha}$ is shared by all samples. This creates a new computational challenge as the number of samples grows. Finally, for a new sample $\boldsymbol{x} \in (\mathbb{R} \cup \{\texttt{N/A}\})^F$, is there a way not to re-run the full F3I procedure? If we assume that the new sample comes from the data set, the simplest idea is to apply on $\boldsymbol{x}$ the imputation improvement procedure in Algorithm 2 with the weight vector $\boldsymbol{\alpha}$ obtained at the final round of F3I. However, this out-of-sample imputation loses the theoretical guarantees in Section 2. Finding a theoretically-backed approach for out-of-sample imputation would also be an interesting question to investigate.

### Broader Impact Statement

The contributions of this paper are essentially theoretical. However, by the definition of our imputation algorithm, possibly sensitive information can leak to imputed values without specific steps taken to avoid it, potentially undermining fairness in downstream analyses. This might be mitigated by modifying the imputation model to protect sensitive variables.

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

## A    Other related works

**Related works**    Relevant methods developed for multivariate time-series data might also be adapted to single-timepoint data sets: Conditional Score-based Diffusion Models (CDSI) (Tashiro et al., 2021a), Generative Adversarial Networks (Luo et al., 2018; Yoon et al., 2018a) or Last Observation Carried Forward (LOCF), where the last non-missing value is duplicated until the next non-missing time point. However, those methods are best-suited in the case where temporal connections can be made, and we restrict our study to single-timepoint data.

Due to diffusion models' impressive generative modeling capacity, many imputation approaches based on them have been developed in recent years. One of the first approaches in this direction was Conditional Score-Based Diffusion models (Tashiro et al., 2021b), which attempts to impute values at missing time points in multivariate time-series data by conditioning the diffusion model on the observed time points and then denoising from white noise using the conditional diffusion model. Another similar approach (Chen et al., 2023) constructs and solves a Schrödinger Bridge problem with conditional constraints to impute missing values in time-series data, where the conditional constraints are derived from the observed values. However,

both these approaches are only available for imputation in multivariate time-series data, which is out of the scope of our paper.

An interesting approach for tabular data (Zheng & Charoenphakdee, 2022) consists of removing the temporal transformer layers from CSDI (Tashiro et al., 2021b) and constructing embeddings for categorical variables, after which the same diffusion procedure as CSDI is applied. Another diffusion-based approach (Jolicoeur-Martineau et al., 2024) constructs multiple copies of the dataset with different noise samples added, on which gradient-boosted trees are trained to perform the reverse diffusion process to reconstruct the missing data values. The NewImp (Chen et al., 2024) model, on the other hand, attempts to learn the missing value distribution by learning the score function of the joint distribution using a Denoising Score Matching approach and then by stepwise reconstructing the missing values using an ODE simulation of the joint distribution based on these learned scores. Another similar approach, MissDiff (Ouyang et al., 2025), attempts to learn a score-based model by modifying the loss in a conventional Variance Preserving SDE for diffusion models to only consider the observed values.

A variational autoencoder-based approach (Peis et al., 2022) generates the missing data by training multiple separate variational autoencoders to encode the latent space of each feature and the dependencies among them. This training is helpful in the presence of diverse feature types. New approaches also explore the use of transformer models in missing data imputation. For instance, ReMasker (Du et al., 2024) is an extension of the Masked AutoEncoder (He et al., 2021) where the encoder and decoder models consist of a sequence of transformer layers, and the model learns by masking additionally available values and learning to predict them, similar to CSDI (Tashiro et al., 2021b). Another approach based on a modification of OTImputer (Muzellec et al., 2020b) attempts to minimize the Wasserstein distance between latent representations of the original and imputed data set generated via a neural network, where the latent representation generating neural network is fit to maximize the Mutual Information to prevent model collapse.

Other generative approaches involve using Generative Adversarial Networks. The simplest GAN-based approach (Li et al., 2019) jointly trains two pairs of generator-discriminator networks, one to predict the data and the other to predict the missing masks. Another GAN-based approach masks certain values in the data set besides the missing ones. It uses the generator to generate all masked values, after which the discriminator is trained to predict the mask matrix from the imputed data set (Yoon et al., 2018b).

Finally, GRAPE is a somewhat more novel approach (You et al., 2020), which consists of representing the data set as a labeled bipartite graph, with the two node types representing the samples and the features and each edge label representing the value of the particular feature for that sample. One then uses a GraphSAGE (Hamilton et al., 2017) inspired Graph Neural Network to predict the missing edge labels, which are the imputed values.

## B    Theoretical assumptions

In this section, we state the formal expression of our assumptions about the data generation procedure to derive theoretical guarantees from our algorithm. We explicitly write the pseudo-code for the full synthetic data generation procedure (matching our theoretical assumptions) in Algorithm 4. We provide an analysis of nearest-neighbor imputation algorithms for three types of missingness mechanisms.

**Assumption B.1.** MCAR mechanism: Bernouilli distribution. The random indicator variables for missing values $m_i^f$ are drawn *iid* from $\mathcal{B}(p^{\mathrm{miss}}(\boldsymbol{x}))$, where $p^{\mathrm{miss}}(\boldsymbol{x}) \in (0,1)$ is a constant value for any $\boldsymbol{x}$.

**Assumption B.2.** MAR mechanism. We assume a subset $\mathcal{F}_o$ of size $F_o < F$ features is always observed. We denote $(x^\star)_i^{|\,\mathrm{obs}} \triangleq (x_i^\star[f])_{f \in \mathcal{F}_o}$. Then there exist a function $p^{\mathrm{miss}}$, $\forall \boldsymbol{x} \in \mathbb{R}^{F_o}$, $\mathbb{P}(m_i^f = 1 \mid (x^\star)_i^{|\,\mathrm{obs}} = \boldsymbol{x}) = p^{\mathrm{miss}}(\boldsymbol{x})$ .

**Assumption B.3.** MNAR mechanism: Gaussian self-masking (Assumption 4 from Le Morvan et al. (2020)). The probability of event $\{m_i^f = 1\}$ depends on $(x^\star)_i^f$: $\exists K_f \in (0,1)$, $\forall x \in \mathbb{R}$, $\mathbb{P}(m_i^f = 1 \mid (x^\star)_i^f = x) = K_f e^{-\frac{1}{\sigma^2}(x - \mu_f)^2} = p^{\mathrm{miss}}(\boldsymbol{x})$ .

First, we assume that each value in the full data matrix is drawn from independent fixed-variance Gaussian distributions.

---

**Algorithm 4** Data generation procedure according to Assumptions B.4-B.6

---

**Input:** $N$ number of samples, $F$ number of features
**Output:** Initial data $X^{\text{miss}} \in (\mathbb{R} \cup \{\texttt{NaN}\})^{N \times F}$ and naively imputed $\widehat{X} \in \mathbb{R}^{N \times F}$
  # Generation of the complete data set
**for** $i = 1, 2, \ldots, N$ **do**
    **for** $f = 1, 2, \ldots, F$ **do**
      $(\boldsymbol{x}^{\star})_i \sim \mathcal{N}_F(\mu_f, \sigma^2)$ (where $\Theta \triangleq (\boldsymbol{\mu} = (\mu_1, \ldots, \mu_F), \sigma^2 \boldsymbol{I}_{F \times F}) \in \mathbb{R}^F \times \mathbb{R}^{F \times F}$)
    **end for**
**end for**
  # Missingness mechanism
**for** $i = 1, 2, \ldots, N$ **do**
    **for** $f = 1, 2, \ldots, F$ **do**
      $m_i^f \sim_{\text{iid}} p^{\text{miss}}((\boldsymbol{x}^{\star})_i, f)$
      **if** $m_i^f = 1$ **then**
        **then** $(x^{\text{miss}})_i^f \leftarrow \texttt{NaN}$
        **else** $(x^{\text{miss}})_i^f \leftarrow (x^{\star})_i^f$
      **end if**
    **end for**
**end for**
  # Create the naively imputed data set
**for** $i = 1, 2, \ldots, N$ **do**
    **for** $f = 1, 2, \ldots, F$ **do**
      **if** $m_i^f = 1$ **then**
        # K-nearest neighbor imputation with uniform weights
        # $k^{\text{th}}$ closest neighbor for $(\boldsymbol{x}^{\text{miss}})_i^f$ is denoted $\mathcal{K}((\boldsymbol{x}^{\text{miss}})_i, f, k)$
        **then** $\widehat{x}_i^f \leftarrow \frac{1}{K} \sum_{k \leq K} x^f_{\mathcal{K}((\boldsymbol{x}^{\text{miss}})_i, f, k)} = \frac{1}{K} \sum_{k \leq K} (x^{\star})^f_{\mathcal{K}((\boldsymbol{x}^{\text{miss}})_i, f, k)}$
        **else** $\widehat{x}_i^f \leftarrow (x^{\star})_i^f$
      **end if**
    **end for**
**end for**

---

**Assumption B.4.** Independent Gaussian distributions. There exist $\boldsymbol{\mu} \in \mathbb{R}^F$ and $\sigma > 0$ such that, for any sample $i \leq N$ and any feature $f \leq F$, $(x^{\star})_i^f \sim_{\text{iid}} \mathcal{N}(\mu_f, \sigma^2)$.

The random indicator variables $m_i^f$ are then independently drawn according to the missingness mechanism with probability $p^{\text{miss}}$. If $m_i^f = 1$, then the covariate at position $(i, f)$ $x_i^f$ in $X$ is unavailable, otherwise, $x_i^f = (x^{\star})_i^f$. We also ensure that there are exactly $K$ neighbors for the initial simple guesses and that we know a (constant) upper bound on the norm of any feature vectors.

**Assumption B.5.** Number of neighbors K. In the remainder of the paper, if $\{i \leq N \mid m_i^f = 0\}$ is the set of data point indices for which the feature $f$ is not missing, then $K \leq \min_{f \leq F} |\{i \leq N \mid m_i^f = 0\}|$. Without a loss of generality, $\min_{f \leq F} |\{i \leq N \mid m_i^f = 0\}| \geq 2$ (otherwise, we can ignore the corresponding feature).

**Assumption B.6.** Upper bound on any of the $(\|\boldsymbol{x}_i\|_2^2)_{i \leq n}$. We assume a constant $S > 0$ exists, such that for any $i \leq N$, $\|\boldsymbol{x}_i\|_2^2 \leq S$ (ignoring potential missing values). Up to renormalization, we assume that $S = 1$. Moreover, the initial imputation step (the "simple guess") preserves that condition, meaning that for any $i \leq N$ and $t \geq 0$, $\|\boldsymbol{x}_i^t\|_2^2 \leq S$, where $\widehat{X}$ is the imputed data matrix with the initial imputation step, and $X^t$ for $t \geq 1$ is obtained through Algorithm 1.

*Remark* B.7. Assumption B.6 can hold. Indeed, Assumption B.6 is satisfied by the K-nearest neighbor imputation with uniform weights, where the imputed value equals the mean of all feature-wise values from the $K$ neighbors.

**Assumption B.8.** Assumptions on $G$ and $\ell$ for Theorem 3.1 Let $\ell$ be a convex pointwise loss such that $\nabla \ell$ is Lipschitz-continuous and $\beta \in [0, 1]$. Define $\mathcal{G}$ as in 2. Also define $\boldsymbol{H}(\mathcal{G}; \boldsymbol{\alpha}, \boldsymbol{\alpha}') = \int_0^1 \nabla \mathcal{G}(\boldsymbol{\alpha}, X; \beta)^\intercal \nabla^2 \mathcal{G}(\boldsymbol{\alpha} + $

$a(\boldsymbol{\alpha}' - \boldsymbol{\alpha}), X; \beta)da$. Let $\boldsymbol{g_1} = \nabla_\alpha(1 - \beta)G(\boldsymbol{\alpha})$, $\boldsymbol{g_2} = -\nabla_\alpha \frac{\beta}{N} \sum_{i \leq N} \ell(\boldsymbol{x_i}(\boldsymbol{\alpha}))$, $\phi_{12}$ be the angle between $\boldsymbol{g_1}$ and $\boldsymbol{g_2}$, and $\boldsymbol{g} = \boldsymbol{g_1} + \boldsymbol{g_2}$. Let $\boldsymbol{\alpha}^t$ be the updated value of $\boldsymbol{\alpha}$ and let $\lambda$ be the step-size for this update. Lemma E.1 shows that $-\mathcal{G}$ is also Lipschitz-continuous w.r.t. $\boldsymbol{\alpha}$, so let this Lipschitz constant be $L$. We assume that there exists $w \leq L$ such that $\boldsymbol{H}(-\mathcal{G}, \boldsymbol{\alpha}, \boldsymbol{\alpha}^t) \geq w\|g\|_2^2$. We also assume that $\cos \phi_{12} \leq \frac{2\|g_1\|_2\|g_2\|_2}{\|g_1\|_2^2\|g_2\|_2^2}$, $w \geq (1 - \cos^2(\phi_{12})\frac{\|g_1 - g_2\|_2^2}{\|g_1 + g_2\|_2^2}W$ and $\lambda \geq \frac{2}{w - (1 - \cos^2(\phi_{12}))\frac{\|g_1 - g_2\|_2^2}{\|g_1 + g_2\|_2^2}W}$.

## C  Properties of the objective function G

**Proposition C.1.** Continuity and derivability of $G$. $G$ is continuous and infinitely derivable with respect to $\boldsymbol{\alpha} \in \triangle_K$.

*Proof.* $G$ is a composition and sum of indefinitely derivable functions on their respective domains, which are compatible: log on $\mathbb{R}^{+*}$, exp on $\mathbb{R}$ of image domain $\mathbb{R}^{+*}$, $\|\cdot\|_2$ and the convex imputation model on $\mathbb{R}$ with image $\mathbb{R}$. $\square$

**Proposition C.2.** Strict concavity of $G$ in $\boldsymbol{\alpha}$. *Assume that $\eta < 4KN$. Then there exists $h_0 > 0$ such that for all $h \geq h_0$, $G$ is strictly concave in $\boldsymbol{\alpha}$.*

We aim to show that a value of $h_0$ always exists such that, for $h \geq 0$, the Hessian matrix of $G$ with respect to $\boldsymbol{\alpha}$ is negative (semi-)definite. First, we compute the Hessian matrix of $G$.

**Lemma C.3.** Gradient of $G$ with respect to $\boldsymbol{\alpha}$. *The gradient $\nabla_{\boldsymbol{\alpha}} G(\boldsymbol{\alpha}, X) \in \mathbb{R}^K$ at $\boldsymbol{\alpha} \in \mathbb{R}^K$ and fixed $X \in \mathbb{R}^{N \times F}$ is*

$$\nabla_{\boldsymbol{\alpha}} G(\boldsymbol{\alpha}, X) = -\sum_{i \leq N} \frac{D_0(\boldsymbol{x_i}(\boldsymbol{\alpha}))^{-1}}{2hN^2(\sqrt{2\pi h})^F} \left( \sum_{j \leq N} e^{-\frac{1}{4h}\|\boldsymbol{x_i}(\boldsymbol{\alpha}) - \widehat{\boldsymbol{x}}_j\|_2^2} (\boldsymbol{x_i}(\boldsymbol{\alpha}) - \widehat{\boldsymbol{x}}_j) \right)^{\mathsf{T}} \widetilde{Z}^{n_i} - 2\eta\boldsymbol{\alpha} \ ,$$

*where $n^i \triangleq (n_1^i, n_2^i, \ldots, n_K^i)$ is the set of indices of the $K$-nearest neighbors of $\boldsymbol{x_i}$ among the reference set $Z = \{\widehat{\boldsymbol{x}}_1, \widehat{\boldsymbol{x}}_2, \ldots, \widehat{\boldsymbol{x}}_N\}$, $\widetilde{Z}^{n_i} \in \mathbb{R}^{F \times K}$ where the $k^{th}$ column of $\widetilde{Z}^{n_i}$ is defined as $(\widetilde{\boldsymbol{z}}^{n_i})_k^f = 0$ if $m_i^f = 0$, $(\widetilde{\boldsymbol{z}}^{n_i})_k^f = \widehat{x}_{n_k^i}^f$ otherwise. That is, $(\widetilde{\boldsymbol{z}}^{n_i})_k$ is equal to the $k^{th}$ closest neighbor of $\boldsymbol{x_i}$ (by increasing order of distance) on missing coordinates of $\boldsymbol{x_i}$, and equal to zero otherwise.*

*Proof.* The gradient of $G$ at $\boldsymbol{\alpha} \in \triangle_K$ for a fixed $X \in \mathbb{R}^{N \times F}$ is

$$\begin{aligned}
\nabla_{\boldsymbol{\alpha}} G(\boldsymbol{\alpha}, X) &= \sum_{i \leq N} \frac{D_0(\boldsymbol{x_i}(\boldsymbol{\alpha}))^{-1}}{N} \nabla_{\boldsymbol{\alpha}} D_0(\boldsymbol{x_i}(\boldsymbol{\alpha})) - 0 - 2\eta\boldsymbol{\alpha} \\
\nabla_{\boldsymbol{\alpha}} D_0(\boldsymbol{x_i}(\boldsymbol{\alpha})) &= -\frac{1}{4hN(\sqrt{2\pi h})^F} \sum_{j \leq N} e^{-\frac{1}{4h}\|\boldsymbol{x_i}(\boldsymbol{\alpha}) - \widehat{\boldsymbol{x}}_j\|_2^2} \nabla_{\boldsymbol{\alpha}} \|\boldsymbol{x_i}(\boldsymbol{\alpha}) - \widehat{\boldsymbol{x}}_j\|_2^2 \\
\nabla_{\boldsymbol{\alpha}} \|\boldsymbol{x_i}(\boldsymbol{\alpha}) - \widehat{\boldsymbol{x}}_j\|_2^2 &= 2(\boldsymbol{x_i}(\boldsymbol{\alpha}) - \widehat{\boldsymbol{x}}_j) \nabla_{\boldsymbol{\alpha}} \boldsymbol{x_i}(\boldsymbol{\alpha}) \text{ and } \nabla_{\boldsymbol{\alpha}} \boldsymbol{x_i}(\boldsymbol{\alpha}) = \widetilde{Z}^{n_i} \ .
\end{aligned}$$

$\square$

**Lemma C.4.** Hessian matrix of $G$ with respect to $\boldsymbol{\alpha}$. *Let us denote for any $i, j \leq N$ and $\boldsymbol{\alpha} \in \triangle_K$*

- *$u_{\boldsymbol{\alpha}}^{ij} \triangleq e^{-\frac{1}{4h}\|\boldsymbol{x_i}(\boldsymbol{\alpha}) - \widehat{\boldsymbol{x}}_j\|_2^2}$ and $U_{\boldsymbol{\alpha}}^i \triangleq \sum_{j \leq N} u_{\boldsymbol{\alpha}}^{ij} = N(\sqrt{2\pi h})^F D_0(\boldsymbol{x_i}(\boldsymbol{\alpha}))$,*

- *$S_{\boldsymbol{\alpha}}^i \triangleq \sum_{j \leq N} u_{\boldsymbol{\alpha}}^{ij}(\boldsymbol{x_i}(\boldsymbol{\alpha}) - \widehat{\boldsymbol{x}}_j)$ and $T_{\boldsymbol{\alpha}}^i \triangleq \sum_{j \leq N} u_{\boldsymbol{\alpha}}^{ij}(\|\boldsymbol{x_i}(\boldsymbol{\alpha}) - \widehat{\boldsymbol{x}}_j\|_2^2 - 2h)$.*

*Then the covariate at position $(k, q)$ of Hessian matrix $\nabla_{\boldsymbol{\alpha}}^2 G(\boldsymbol{\alpha}, X) \in \mathbb{R}^{K \times K}$ at $\boldsymbol{\alpha}$ and fixed $X$ is*

$$\frac{\partial^2 G(\boldsymbol{\alpha}, X)}{\partial \alpha_k \partial \alpha_q} = \sum_{i \leq N} \left( \frac{T_{\boldsymbol{\alpha}}^i}{NU_{\boldsymbol{\alpha}}^i} - \frac{(S_{\boldsymbol{\alpha}}^i)^{\mathsf{T}} S_{\boldsymbol{\alpha}}^i}{4h^2(U_{\boldsymbol{\alpha}}^i)^2} \right) (\widetilde{\boldsymbol{z}}^{n_i})_q^{\mathsf{T}} (\widetilde{\boldsymbol{z}}^{n_i})_k - \eta \mathbb{1}(q = k) \ .$$

*Proof.* According to Lemma C.3, for any $k \leq K$

$$\frac{\partial G(\boldsymbol{\alpha}, X)}{\partial \alpha_k} = -\sum_{i \leq N} \frac{D_0(\boldsymbol{x}_i(\boldsymbol{\alpha}))^{-1}}{2hN^2(\sqrt{2\pi}h)^F} (S_{\boldsymbol{\alpha}}^i)^{\mathsf{T}}(\widetilde{\boldsymbol{z}}^{n_i})_k - 2\eta\alpha_k \ .$$

This implies that

$$\frac{\partial^2 G(\boldsymbol{\alpha}, X)}{\partial \alpha_k \partial \alpha_q} + 2\eta\mathbb{1}(q=k) = \sum_{i \leq N} \frac{-D_0(\boldsymbol{x}_i(\boldsymbol{\alpha}))^{-1}}{2hN^2(\sqrt{2\pi}h)^F} \left( \left( \frac{\partial S_{\boldsymbol{\alpha}}^i}{\partial \alpha_q} \right)^{\mathsf{T}} (\widetilde{\boldsymbol{z}}^{n_i})_k \right.$$
$$\left. - \frac{(S_{\boldsymbol{\alpha}}^i)^{\mathsf{T}}(\widetilde{\boldsymbol{z}}^{n_i})_k}{D_0(\boldsymbol{x}_i(\boldsymbol{\alpha}))} \frac{\partial D_0(\boldsymbol{x}_i(\boldsymbol{\alpha}))}{\partial \alpha_q} \right)$$

And then

$$\frac{\partial S_{\boldsymbol{\alpha}}^i}{\partial \alpha_q} = \frac{-1}{4h} \sum_{j \leq N} e^{-\frac{1}{4h}\|\boldsymbol{x}_i(\boldsymbol{\alpha}) - \widehat{\boldsymbol{x}}_j\|_2^2} \left( \frac{\partial\|\boldsymbol{x}_i(\boldsymbol{\alpha}) - \widehat{\boldsymbol{x}}_j\|_2^2}{\partial \alpha_q} \right)^{\mathsf{T}} (\boldsymbol{x}_i(\boldsymbol{\alpha}) - \widehat{\boldsymbol{x}}_j)$$
$$+ \sum_{j \leq N} e^{-\frac{1}{4h}\|\boldsymbol{x}_i(\boldsymbol{\alpha}) - \widehat{\boldsymbol{x}}_j\|_2^2} \frac{\partial \boldsymbol{x}_i(\boldsymbol{\alpha})}{\partial \alpha_q}$$
$$= \frac{-1}{2h} \sum_{j \leq N} e^{-\frac{1}{4h}\|\boldsymbol{x}_i(\boldsymbol{\alpha}) - \widehat{\boldsymbol{x}}_j\|_2^2} (\widetilde{\boldsymbol{z}}^{n_i})_q^{\mathsf{T}} (\boldsymbol{x}_i(\boldsymbol{\alpha}) - \widehat{\boldsymbol{x}}_j)^{\mathsf{T}} (\boldsymbol{x}_i(\boldsymbol{\alpha}) - \widehat{\boldsymbol{x}}_j)$$
$$+ \sum_{j \leq N} e^{-\frac{1}{4h}\|\boldsymbol{x}_i(\boldsymbol{\alpha}) - \widehat{\boldsymbol{x}}_j\|_2^2} (\widetilde{\boldsymbol{z}}^{n_i})_q^{\mathsf{T}}$$

That is,

$$\frac{\partial S_{\boldsymbol{\alpha}}^i}{\partial \alpha_q} = (\widetilde{\boldsymbol{z}}^{n_i})_q^{\mathsf{T}} \underbrace{\left( \sum_{j \leq N} e^{-\frac{1}{4h}\|\boldsymbol{x}_i(\boldsymbol{\alpha}) - \widehat{\boldsymbol{x}}_j\|_2^2} \left( 1 - (2h)^{-1}\|\boldsymbol{x}_i(\boldsymbol{\alpha}) - \widehat{\boldsymbol{x}}_j\|_2^2 \right) \right)}_{=-2hT_{\boldsymbol{\alpha}}^i}$$

$$\frac{\partial D_0(\boldsymbol{x}_i(\boldsymbol{\alpha}))}{\partial \alpha_q} = -\frac{(S_{\boldsymbol{\alpha}}^i)^{\mathsf{T}}(\widetilde{\boldsymbol{z}}^{n_i})_q}{2hN(\sqrt{2\pi}h)^F} \text{ according to Lemma C.3 }.$$

Moreover, since $S_{\boldsymbol{\alpha}}^i, (\widetilde{\boldsymbol{z}}^{n_i})_k \in \mathbb{R}^F$ for any $k \leq K$

$$(S_{\boldsymbol{\alpha}}^i)^{\mathsf{T}}(\widetilde{\boldsymbol{z}}^{n_i})_k (S_{\boldsymbol{\alpha}}^i)^{\mathsf{T}}(\widetilde{\boldsymbol{z}}^{n_i})_q = \underbrace{(S_{\boldsymbol{\alpha}}^i)^{\mathsf{T}}(\widetilde{\boldsymbol{z}}^{n_i})_q}_{=(\widetilde{\boldsymbol{z}}^{n_i})_q^{\mathsf{T}} S_{\boldsymbol{\alpha}}^i} (S_{\boldsymbol{\alpha}}^i)^{\mathsf{T}}(\widetilde{\boldsymbol{z}}^{n^i})_k = (\widetilde{\boldsymbol{z}}^{n_i})_q^{\mathsf{T}}(S_{\boldsymbol{\alpha}}^i)^{\mathsf{T}} S_{\boldsymbol{\alpha}}^i (\widetilde{\boldsymbol{z}}^{n_i})_k \ .$$

$\square$

Then, to show that $G$ is (strictly) concave, it is enough to show that the Hessian matrix $\nabla_{\boldsymbol{\alpha}}^2 G(\boldsymbol{\alpha}, X)$ is negative semi-definite (or definite). We assume that $\eta < 4S^2 K = 4K$ (using Assumption B.6), which is the case for most realistic settings.

*Proof.* Let us denote $\boldsymbol{x}_{\boldsymbol{\alpha}}^{ij} \triangleq x_{\boldsymbol{\alpha}}^i - \widehat{\boldsymbol{x}}_j$ for any $i, j \leq N$. Then

$$\frac{T_{\boldsymbol{\alpha}}^i}{NU_{\boldsymbol{\alpha}}^i} - \frac{(S_{\boldsymbol{\alpha}}^i)^{\mathsf{T}} S_{\boldsymbol{\alpha}}^i}{4h^2(U_{\boldsymbol{\alpha}}^i)^2} = \frac{T_{\boldsymbol{\alpha}}^i}{NU_{\boldsymbol{\alpha}}^i} - \frac{1}{4h^2} \left( \sum_{j,j' \leq N} \frac{u_{\boldsymbol{\alpha}}^{ij}}{U_{\boldsymbol{\alpha}}^i} \frac{u_{\boldsymbol{\alpha}}^{ij'}}{U_{\boldsymbol{\alpha}}^i} (\boldsymbol{x}_{\boldsymbol{\alpha}}^{ij})^{\mathsf{T}} \boldsymbol{x}_{\boldsymbol{\alpha}}^{ij'} \right) \ . \tag{3}$$

Now consider $(\boldsymbol{x}_{\boldsymbol{\alpha}}^{ij})^{\mathsf{T}} \boldsymbol{x}_{\boldsymbol{\alpha}}^{ij'} = \langle \boldsymbol{x}_i(\boldsymbol{\alpha}) - \widehat{\boldsymbol{x}}_j, \ \boldsymbol{x}_i(\boldsymbol{\alpha}) - \widehat{\boldsymbol{x}}_{j'} \rangle$. From the triangle equality, we have, for all $j, j' \leq N$

$$\langle \boldsymbol{x}_i(\boldsymbol{\alpha}) - \widehat{\boldsymbol{x}}_j, \ \boldsymbol{x}_i(\boldsymbol{\alpha}) - \widehat{\boldsymbol{x}}_{j'} \rangle = \frac{1}{2}(\|\boldsymbol{x}_i(\boldsymbol{\alpha}) - \widehat{\boldsymbol{x}}_j\|_2^2 + \|\boldsymbol{x}_i(\boldsymbol{\alpha}) - \widehat{\boldsymbol{x}}_{j'}\|_2^2 - \|\widehat{\boldsymbol{x}}_j - \widehat{\boldsymbol{x}}_{j'}\|_2^2) \ .$$

We plug this inequality into 3

$$
\begin{aligned}
\frac{T_{\boldsymbol{\alpha}}^i}{NU_{\boldsymbol{\alpha}}^i} - \frac{(S_{\boldsymbol{\alpha}}^i)^\intercal S_{\boldsymbol{\alpha}}^i}{4h^2(U_{\boldsymbol{\alpha}}^i)^2} &= \frac{T_{\boldsymbol{\alpha}}^i}{NU_{\boldsymbol{\alpha}}^i} + \frac{1}{8h^2} \sum_{j,j' \leq N} \frac{u_{\boldsymbol{\alpha}}^{ij}}{U_{\boldsymbol{\alpha}}^i} \frac{u_{\boldsymbol{\alpha}}^{ij'}}{U_{\boldsymbol{\alpha}}^i} \|\widehat{\boldsymbol{x}}_j - \widehat{\boldsymbol{x}}_{j'}\|_2^2 \\
&\quad \underbrace{- \frac{1}{8h^2} \sum_{j,j' \leq N} \frac{u_{\boldsymbol{\alpha}}^{ij}}{U_{\boldsymbol{\alpha}}^i} \frac{u_{\boldsymbol{\alpha}}^{ij'}}{U_{\boldsymbol{\alpha}}^i} (\|\boldsymbol{x}_i(\boldsymbol{\alpha}) - \widehat{\boldsymbol{x}}_j\|_2^2 + \|\boldsymbol{x}_i(\boldsymbol{\alpha}) - \widehat{\boldsymbol{x}}_{j'}\|_2^2)}_{\geq 0} \\
&\leq \frac{T_{\boldsymbol{\alpha}}^i}{NU_{\boldsymbol{\alpha}}^i} + \frac{1}{8h^2} \sum_{j,j' \leq N} \frac{u_{\boldsymbol{\alpha}}^{ij}}{U_{\boldsymbol{\alpha}}^i} \frac{u_{\boldsymbol{\alpha}}^{ij'}}{U_{\boldsymbol{\alpha}}^i} \|\widehat{\boldsymbol{x}}_j - \widehat{\boldsymbol{x}}_{j'}\|_2^2 \, .
\end{aligned}
$$

Obviously $\frac{u_{\boldsymbol{\alpha}}^{ij}}{U_{\boldsymbol{\alpha}}^i} \leq 1$. Moreover, since Assumption B.6 gives $\|\widehat{\boldsymbol{x}}_j\|_2^2 \leq S$ and $\|\boldsymbol{x}_j(\boldsymbol{\alpha})\|_2^2 \leq S$ (using Jensen's inequality) for any $j \leq N$,

$$
\begin{aligned}
\frac{T_{\boldsymbol{\alpha}}^i}{NU_{\boldsymbol{\alpha}}^i} - \frac{(S_{\boldsymbol{\alpha}}^i)^\intercal S_{\boldsymbol{\alpha}}^i}{4h^2(U_{\boldsymbol{\alpha}}^i)^2} &\leq \frac{T_{\boldsymbol{\alpha}}^i}{NU_{\boldsymbol{\alpha}}^i} + \frac{2N^2 S}{8h^2} = \frac{T_{\boldsymbol{\alpha}}^i}{NU_{\boldsymbol{\alpha}}^i} + \frac{N^2 S}{4h^2} \\
&= \sum_{j \leq N} \frac{u_{\boldsymbol{\alpha}}^{ij}}{NU_{\boldsymbol{\alpha}}^i} (\|\boldsymbol{x}^i(\boldsymbol{\alpha}) - \widehat{\boldsymbol{x}}_j\|_2^2 - 2h) + \frac{NS}{4h^2} \\
&\leq \frac{1}{N} \sum_{j \leq N} \|\boldsymbol{x}^i(\boldsymbol{\alpha}) - \widehat{\boldsymbol{x}}_j\|_2^2 - 2h + \frac{N^2 S}{4h^2} \\
&\leq \frac{1}{N} \sum_{j \leq N} (\|\boldsymbol{x}^i(\boldsymbol{\alpha})\|_2^2 + \|\widehat{\boldsymbol{x}}_j\|_2^2) - 2h + \frac{N^2 S}{4h^2} \\
&\leq 2S - 2h + N^2 S(4h^2)^{-1} \, .
\end{aligned}
$$

We set $C(h) \triangleq h^{-2}(-2h^3 + 2Sh^2 + N^2 S/4)$, and fix $\boldsymbol{v} \in \mathbb{R}^K$. Then

$$
\begin{aligned}
\boldsymbol{v}^\intercal \nabla_{\boldsymbol{\alpha}}^2 G(\boldsymbol{\alpha}, X) \boldsymbol{v} &= -\eta \|\boldsymbol{v}\|_2^2 + \sum_{i \leq N} \left( \frac{T_{\boldsymbol{\alpha}}^i}{NU_{\boldsymbol{\alpha}}^i} - \frac{(S_{\boldsymbol{\alpha}}^i)^\intercal S_{\boldsymbol{\alpha}}^i}{4h^2(U_{\boldsymbol{\alpha}}^i)^2} \right) (\boldsymbol{v}^\intercal (\widetilde{Z}^{n_i})^\intercal \widetilde{Z}^{n_i} \boldsymbol{v}) \\
&\leq -\eta \|\boldsymbol{v}\|_2^2 + C(h) \sum_{i \leq N} \|\widetilde{Z}^{n_i} \boldsymbol{v}\|_2^2
\end{aligned}
$$

so, using Technical lemma 1 (proven below),

$$
\boldsymbol{v}^\intercal \nabla_{\boldsymbol{\alpha}}^2 G(\boldsymbol{\alpha}, X) \boldsymbol{v} \leq \|\boldsymbol{v}\|_2^2 (2KSC(h) - \eta) = \|\boldsymbol{v}\|_2^2 \times h^{-2} \left( -4KSh^3 + (4S^2 K - \eta)h^2 + \frac{KN^2 S^2}{2} \right) \, .
$$

Then we choose $h > 0$ such that $\boldsymbol{v}^\intercal \nabla_{\boldsymbol{\alpha}}^2 G(\boldsymbol{\alpha}, X) \boldsymbol{v} < 0$. That is

$$
-4KSh^3 + (4S^2 K - \eta)h^2 + \frac{KN^2 S^2}{2} < 0 \Leftrightarrow -2h^3 + \frac{4S^2 K - \eta}{2KS}h^2 + \frac{N^2 S}{4} < 0 \, . \tag{4}
$$

Under the assumption of $\eta < 4S^2 K$, this is equivalent to analyzing the following cubic equation

$$
-2h^3 + bh^2 + c = 0 \text{ where } b, c > 0 \, .
$$

The cubic equation above admits three roots and at least one real root. We show that at least one real root is positive (thus, corresponds to a valid bandwidth). To show that there exists $h > 0$ such that $-2h^3 + bh^2 + c < 0$, it is enough to show that there exists $h' \in \mathbb{R}$ such that on $[h', +\inf)$, continuous and infinitely derivable function $x \mapsto -2x^3 + bx^2 + c$ is strictly decreasing. We have $\frac{\mathrm{d}}{\mathrm{d}h}(-2h^3 + bh^2 + c) = -6h^2 + 2bh$ with roots 0

and $b/3$, and $\frac{\mathrm{d}^2}{\mathrm{d}^2 h}(-2h^3 + bh^2 + c) = -12h + 2b$, and then $-12 \times 0 + 2b = 2b > 0$ and $-12 \times \frac{b}{3} + 2b = -2b < 0$. The analysis of the behavior of $x \mapsto -2x^3 + bx^2 + c$ then shows that the condition is fulfilled for $h' = b/3 > 0$.

Finally, the value of $h$ can be found through the known closed-form expressions of roots of the rightmost cubic polynomial in $h$ in 4. $\qquad\square$

**Proposition C.5.** The gradient of $G(\cdot, X)$ for any $X \in \mathbb{R}^{N \times F}$ is Lipschitz-continuous. *There exists a positive constant $H$ such that*

$$\|\nabla_{\boldsymbol{\alpha}} G(\boldsymbol{\alpha}, X) - \nabla_{\boldsymbol{\alpha}} G(\boldsymbol{\alpha}', X)\|_2 \leq H \|\boldsymbol{\alpha} - \boldsymbol{\alpha}'\|_2 .$$

*Proof.* According to Lemma C.3, and using notation from Lemma C.4, for any $X \in \mathbb{R}^{N \times F}$ and $\boldsymbol{\alpha} \in \triangle_K$

$$\nabla_{\boldsymbol{\alpha}} G(\boldsymbol{\alpha}, X) = -\frac{1}{2hN} \sum_{i,j \leq N} \frac{u_{\boldsymbol{\alpha}}^{ij}}{U_{\boldsymbol{\alpha}}^i} (\boldsymbol{x}_i(\boldsymbol{\alpha}) - \widehat{\boldsymbol{x}}_j)^{\mathsf{T}} \widetilde{Z}^{n_i} - 2\eta \boldsymbol{\alpha} .$$

Then for any $\boldsymbol{\alpha}, \boldsymbol{\alpha}' \in \triangle_K$

$$\|\nabla_{\boldsymbol{\alpha}} G(\boldsymbol{\alpha}, X) - \nabla_{\boldsymbol{\alpha}} G(\boldsymbol{\alpha}', X)\|_2$$

$$= \| -\frac{1}{2hN} \sum_{i,j \leq N} \Big( \frac{u_{\boldsymbol{\alpha}}^{ij}}{U_{\boldsymbol{\alpha}}^i} (\boldsymbol{x}_i(\boldsymbol{\alpha}) - \widehat{\boldsymbol{x}}_j) - \frac{u_{\boldsymbol{\alpha}'}^{ij}}{U_{\boldsymbol{\alpha}'}^i} (\boldsymbol{x}_i(\boldsymbol{\alpha}') - \widehat{\boldsymbol{x}}_j) \Big)^{\mathsf{T}} \widetilde{Z}^{n_i} - 2\eta(\boldsymbol{\alpha} - \boldsymbol{\alpha}') \|_2$$

$$\leq \frac{1}{2hN} \sum_{i \leq N} \| \sum_{j \leq N} \Big( \frac{u_{\boldsymbol{\alpha}}^{ij}}{U_{\boldsymbol{\alpha}}^i} (\boldsymbol{x}_i(\boldsymbol{\alpha}) - \widehat{\boldsymbol{x}}_j) - \frac{u_{\boldsymbol{\alpha}'}^{ij}}{U_{\boldsymbol{\alpha}'}^i} (\boldsymbol{x}_i(\boldsymbol{\alpha}') - \widehat{\boldsymbol{x}}_j) \Big)^{\mathsf{T}} \widetilde{Z}^{n_i} \|_2 + 2\eta \|\boldsymbol{\alpha} - \boldsymbol{\alpha}'\|_2$$

$$\leq \frac{1}{2hN} \sum_{i \leq N} \| \sum_{j \leq N} \frac{u_{\boldsymbol{\alpha}}^{ij}}{U_{\boldsymbol{\alpha}}^i} (\boldsymbol{x}_i(\boldsymbol{\alpha}) - \widehat{\boldsymbol{x}}_j) - \frac{u_{\boldsymbol{\alpha}'}^{ij}}{U_{\boldsymbol{\alpha}'}^i} (\boldsymbol{x}_i(\boldsymbol{\alpha}') - \widehat{\boldsymbol{x}}_j) \|_2 \sqrt{2KS} + 2\eta \|\boldsymbol{\alpha} - \boldsymbol{\alpha}'\|_2$$

(using the Cauchy-Schwartz inequality and Technical lemma 1)

$$= \frac{\sqrt{KS}}{\sqrt{2}hN} \sum_{i \leq N} \| \sum_{j \leq N} \frac{u_{\boldsymbol{\alpha}}^{ij}}{U_{\boldsymbol{\alpha}}^i} \boldsymbol{x}_i(\boldsymbol{\alpha}) - \frac{u_{\boldsymbol{\alpha}'}^{ij}}{U_{\boldsymbol{\alpha}'}^i} \boldsymbol{x}_i(\boldsymbol{\alpha}') + \Big( \frac{u_{\boldsymbol{\alpha}'}^{ij}}{U_{\boldsymbol{\alpha}'}^i} - \frac{u_{\boldsymbol{\alpha}}^{ij}}{U_{\boldsymbol{\alpha}}^i} \Big) \widehat{\boldsymbol{x}}_j \|_2 + 2\eta \|\boldsymbol{\alpha} - \boldsymbol{\alpha}'\|_2$$

$$\underset{U_{\boldsymbol{\alpha}}^i = \sum_j u_{\boldsymbol{\alpha}}^{ij}}{=} \frac{\sqrt{KS}}{\sqrt{2}hN} \sum_{i \leq N} \| \boldsymbol{x}_i(\boldsymbol{\alpha}) - \boldsymbol{x}_i(\boldsymbol{\alpha}') + \sum_{j \leq N} \Big( \frac{u_{\boldsymbol{\alpha}'}^{ij}}{U_{\boldsymbol{\alpha}'}^i} - \frac{u_{\boldsymbol{\alpha}}^{ij}}{U_{\boldsymbol{\alpha}}^i} \Big) \widehat{\boldsymbol{x}}_j \|_2 + 2\eta \|\boldsymbol{\alpha} - \boldsymbol{\alpha}'\|_2$$

$$\leq \frac{\sqrt{KS}}{\sqrt{2}hN} \sum_{i \leq N} \Big( \sqrt{S} \|\boldsymbol{\alpha} - \boldsymbol{\alpha}'\|_2 + \| \sum_{j \leq N} \Big( \frac{u_{\boldsymbol{\alpha}'}^{ij}}{U_{\boldsymbol{\alpha}'}^i} - \frac{u_{\boldsymbol{\alpha}}^{ij}}{U_{\boldsymbol{\alpha}}^i} \Big) \widehat{\boldsymbol{x}}_j \|_2 \Big) + 2\eta \|\boldsymbol{\alpha} - \boldsymbol{\alpha}'\|_2$$

$$\leq \frac{S\sqrt{K}}{\sqrt{2}hN} \sum_{i \leq N} \Big( \|\boldsymbol{\alpha} - \boldsymbol{\alpha}'\|_2 + \sqrt{2F} \sqrt{\sum_{j \leq N} \Big( \frac{u_{\boldsymbol{\alpha}'}^{ij}}{U_{\boldsymbol{\alpha}'}^i} - \frac{u_{\boldsymbol{\alpha}}^{ij}}{U_{\boldsymbol{\alpha}}^i} \Big)^2} \Big) + 2\eta \|\boldsymbol{\alpha} - \boldsymbol{\alpha}'\|_2 ,$$

(using Cauchy-Schwartz, Assumption B.6, and $\|\widehat{X}\|_F \leq \sqrt{2FS}$)

All that remains is to show that $f_i : \boldsymbol{\alpha} \in \triangle_K \mapsto (\frac{u_{\boldsymbol{\alpha}}^{ij}}{U_{\boldsymbol{\alpha}}^i})_{j \leq N} \in \triangle_N$ (which is a bounded space) is Lipschitz-continuous in $\boldsymbol{\alpha}$. For starters, if all coordinates of $f_i$ $f_{i,j} : \boldsymbol{\alpha} \in \triangle_K \mapsto u_{\boldsymbol{\alpha}}^{ij}/U_{\boldsymbol{\alpha}}^i \in [0, 1]$ are Lipschitz-continuous, each with constant $L_j$, then it is easy to show that $f_i$ is Lipschitz-continuous (always with respect to the $\ell_2$-norm) with constant $L = \sqrt{\sum_{j \leq N} L_j^2}$. Let's consider now any $j \leq N$. For any pair of points $\boldsymbol{\alpha}_1, \boldsymbol{\alpha}_2 \in \triangle_K$, let's introduce the linear path $\gamma : t \in [0, 1] \mapsto t\boldsymbol{\alpha}_1 + (1 - t)\boldsymbol{\alpha}_2$. $f_{i,j} \circ \gamma$ is well-defined, continuous on the closed space $[0, 1]$, differentiable on $(0, 1)$, then by the mean-value theorem

$$f_{i,j}(\boldsymbol{\alpha}_1) - f_{i,j}(\boldsymbol{\alpha}_2) = f_{i,j} \circ \gamma(1) - f_{i,j} \circ \gamma(0) \leq \sup_{t' \in [0,1]} \underbrace{\nabla_t f_{i,j} \circ \gamma(t')}_{= (\nabla_{\boldsymbol{\alpha}} f_{i,j})(\gamma(t'))^{\mathsf{T}}(\boldsymbol{\alpha}_1 - \boldsymbol{\alpha}_2)} (1 - 0) .$$

Meaning that, using the Cauchy-Schwartz inequality

$$|f_{i,j}(\boldsymbol{\alpha}_1) - f_{i,j}(\boldsymbol{\alpha}_2)| \leq \| \sup_{t \in [0,1]} (\nabla_{\boldsymbol{\alpha}} f_{i,j})(\gamma(t)) \|_2 \| \boldsymbol{\alpha}_1 - \boldsymbol{\alpha}_2 \|_2 \leq \| \underbrace{\sup_{\boldsymbol{\alpha} \in \triangle_K} \nabla_{\boldsymbol{\alpha}} f_{i,j}(\boldsymbol{\alpha}) \|_2}_{= L_j} \| \boldsymbol{\alpha}_1 - \boldsymbol{\alpha}_2 \|_2 .$$

Let's show that the value of $L_j$ is bounded (we are not interested in finding the tightest value of $L_j$, simply that $L_j < \infty$). Then for any $\boldsymbol{\alpha} \in \triangle_K$

$$\nabla_{\boldsymbol{\alpha}} f_{i,j}(\boldsymbol{\alpha}) = \left( -\frac{u_{\boldsymbol{\alpha}}^{ij}}{2hU_{\boldsymbol{\alpha}}^i}(\boldsymbol{x}_i(\boldsymbol{\alpha}) - \widehat{\boldsymbol{x}}_j) - \frac{u_{\boldsymbol{\alpha}}^{ij}}{(U_{\boldsymbol{\alpha}}^i)^2} \sum_{\ell \leq N} \frac{-1}{2h} u_{\boldsymbol{\alpha}}^{i\ell}(\boldsymbol{x}_i(\boldsymbol{\alpha}) - \widehat{\boldsymbol{x}}_\ell) \right)^{\mathsf{T}} \widetilde{Z}^{n_i}$$

$$= -\frac{u_{\boldsymbol{\alpha}}^{ij}}{2hU_{\boldsymbol{\alpha}}^i} \left( (\boldsymbol{x}_i(\boldsymbol{\alpha}) - \widehat{\boldsymbol{x}}_j) - \sum_{\ell \leq N} \frac{u_{\boldsymbol{\alpha}}^{i\ell}}{U_{\boldsymbol{\alpha}}^i}(\boldsymbol{x}_i(\boldsymbol{\alpha}) - \widehat{\boldsymbol{x}}_\ell) \right)^{\mathsf{T}} \widetilde{Z}^{n_i} ,$$

Using the Cauchy-Schwartz inequality and Technical lemma 1

$$\|\nabla_{\boldsymbol{\alpha}} f_{i,j}(\boldsymbol{\alpha})\|_2 \leq \frac{1}{2h} \underbrace{\left| \frac{u_{\boldsymbol{\alpha}}^{ij}}{U_{\boldsymbol{\alpha}}^i} \right|}_{\leq 1} \left\| (\boldsymbol{x}_i(\boldsymbol{\alpha}) - \widehat{\boldsymbol{x}}_j) - \sum_{\ell \leq N} \frac{u_{\boldsymbol{\alpha}}^{i\ell}}{U_{\boldsymbol{\alpha}}^i}(\boldsymbol{x}_i(\boldsymbol{\alpha}) - \widehat{\boldsymbol{x}}_\ell) \right\|_2 \sqrt{2KS}$$

Using the triangular inequality on the $\ell_2$-norm

$$\leq \frac{1}{2h} \left( \underbrace{\|\boldsymbol{x}_i(\boldsymbol{\alpha}) - \widehat{\boldsymbol{x}}_j\|_2}_{\leq \sqrt{2S}} + \sum_{\ell \leq N} \underbrace{\left| \frac{u_{\boldsymbol{\alpha}}^{i\ell}}{U_{\boldsymbol{\alpha}}^i} \right|}_{\leq 1} \underbrace{\|\boldsymbol{x}_i(\boldsymbol{\alpha}) - \widehat{\boldsymbol{x}}_\ell\|_2}_{\leq \sqrt{2S}} \right) \sqrt{2KS}$$

$$\leq \frac{1}{2h}(N+1)\sqrt{2S}\sqrt{2KS} = \frac{(N+1)}{h}S\sqrt{K} .$$

All in all, $f_{i,j}$ is $\frac{(N+1)}{h}S\sqrt{K}$-Lipschitz continuous in $\boldsymbol{\alpha}$ and then $f_i$ is $\frac{(N+1)}{h}S\sqrt{NK}$-Lipschitz continuous in $\boldsymbol{\alpha}$. Finally

$$\|\nabla_{\boldsymbol{\alpha}} G(\boldsymbol{\alpha}, X) - \nabla_{\boldsymbol{\alpha}} G(\boldsymbol{\alpha}', X)\|_2 \leq \frac{S\sqrt{K}}{\sqrt{2}hN} \sum_{i \leq N} \left( \|\boldsymbol{\alpha} - \boldsymbol{\alpha}'\|_2 + \sqrt{2F}\|f_i(\boldsymbol{\alpha}') - f_i(\boldsymbol{\alpha})\|_2 \right)$$

$$+ 2\eta\|\boldsymbol{\alpha} - \boldsymbol{\alpha}'\|_2$$

$$\leq \frac{S\sqrt{K}}{\sqrt{2}hN} \sum_{i \leq N} \left( 1 + \sqrt{2F}\frac{(N+1)}{h}S\sqrt{NK} \right)\|\boldsymbol{\alpha} - \boldsymbol{\alpha}'\|_2$$

$$+ 2\eta\|\boldsymbol{\alpha} - \boldsymbol{\alpha}'\|_2$$

$$\leq \underbrace{\left( \frac{S\sqrt{K}}{\sqrt{2}h}\left( 1 + \sqrt{2F}\frac{(N+1)}{h}S\sqrt{K} \right) + 2\eta \right)}_{\triangleq H} \|\boldsymbol{\alpha} - \boldsymbol{\alpha}'\|_2$$

Then $\nabla_{\boldsymbol{\alpha}} G(\boldsymbol{\alpha}, X)$ is $H$-Lipschitz continuous in $\boldsymbol{\alpha}$ with respect to the $\ell_2$-norm. □

# D    Regret analysis of F3I

**Theorem D.1.** High-probability upper bound on the regret for F3I *Under Assumptions B.4-B.6, for any initial matrix $X \in (\mathbb{R} \cup \{\textit{N/A}\})^{N \times F}$,*

$$\max_{\boldsymbol{\alpha} \in \triangle_K} \sum_{s=1}^{t} G_\star(\boldsymbol{\alpha}, X^{s-1}) - G_\star(\boldsymbol{\alpha}^s, X^{s-1}) \quad \leq \quad C_G^{AH} \sqrt{t} + H^{miss} h^{-1} t \,,$$

*with probability $1 - 1/N$, where $H^{miss} = \mathcal{O}(F + \ln N)$ is another value which depends on the missingness mechanism and $h$ is chosen to guarantee that $G$ is concave in its first argument (Proposition C.2). $C_G^{AH} = \mathcal{O}(\sqrt{\log(K)})$ is the constant associated with the regret bound on the gain in F3I incurred by AdaHedge.*

*Proof.* We set the gain in F3I to $g^s(\boldsymbol{\alpha}) \triangleq \sum_{k \leq K} \alpha_k \frac{\partial G}{\partial \alpha_k}(\boldsymbol{\alpha}^s, X^{s-1})$ for $s \leq t$ and $\boldsymbol{\alpha} \in \triangle_K$. We recall the gradient trick for any convex function $\ell$ of gradient $g_t \triangleq \nabla \ell(\boldsymbol{x}_t)$ at a point $\boldsymbol{x}_t$; the following inequality holds thanks to the convexity of $\ell$: for any point $\boldsymbol{x}$, $\ell(\boldsymbol{x}_t) - \ell(\boldsymbol{x}) \leq \nabla g_t^\mathsf{T}(\boldsymbol{x}_t - \boldsymbol{x})$. We use the gradient trick to transfer the regret bound from a linear loss function (the one used in AdaHedge) to the convex loss $-G(\cdot, X^{t-1})$ (the target loss function) for any $t \geq 1$, for all $\boldsymbol{\alpha}, \boldsymbol{\alpha}^s \in \triangle_K$, $s \leq t$,

$$\sum_{s=1}^{t} G(\boldsymbol{\alpha}, X^{s-1}) - G(\boldsymbol{\alpha}^s, X^{s-1}) \leq \sum_{s=1}^{t} (\boldsymbol{\alpha} - \boldsymbol{\alpha}^s)^\mathsf{T} \nabla_{\boldsymbol{\alpha}} G(\boldsymbol{\alpha}^s, X^{s-1}) \,.$$

and note that for any $\boldsymbol{\alpha} \in \triangle_K$,

$$\sum_{s=1}^{t} g^s(\boldsymbol{\alpha}) - g^s(\boldsymbol{\alpha}^s) = \sum_{s=1}^{t} \boldsymbol{\alpha}^\mathsf{T} \nabla_{\boldsymbol{\alpha}} G(\boldsymbol{\alpha}^s, X^{s-1}) - (\boldsymbol{\alpha}^s)^\mathsf{T} \nabla_{\boldsymbol{\alpha}} G(\boldsymbol{\alpha}^s, X^{s-1})$$

$$= \sum_{s=1}^{t} (\boldsymbol{\alpha} - \boldsymbol{\alpha}^s)^\mathsf{T} \nabla_{\boldsymbol{\alpha}} G(\boldsymbol{\alpha}^s, X^{s-1}) \,.$$

Then applying the regret bound of AdaHedge (Technical lemma 2, proven below) to that gain yields at time $t > 1$ (rightmost term) and the gradient trick on the function $G$ which is concave in its first argument (leftmost term) with Proposition C.2

$$\forall \boldsymbol{\alpha} \in \triangle_K, \ \sum_{s=1}^{t} G(\boldsymbol{\alpha}, X^{s-1}) - G(\boldsymbol{\alpha}^s, X^{s-1}) \leq 2\delta_t \sqrt{t \log(K)} + 16\delta_t \left( 2 + \frac{\log K}{3} \right) \,, \tag{5}$$

where $\delta_t \triangleq \max_{s \leq t} \left( \max_{k \leq K} \frac{\partial G}{\partial \alpha_k}(\boldsymbol{\alpha}^s, X^{s-1}) - \min_{q \leq K} \frac{\partial G}{\partial \alpha_q}(\boldsymbol{\alpha}^s, X^{s-1}) \right)$. Now we go from $G$ to $G_\star$ point-wise. Corollary F.6 with $\delta = 1/N$ states that under Assumptions B.4-B.6, there exists $C_{1/N^3}^{\mathrm{miss}} = \mathcal{O}(F + \ln N)$ such that for any $i \leq N$, $\|\widehat{\boldsymbol{x}}_i - (\boldsymbol{x}^\star)_i\|_2^2 \leq C_{1/N^3}^{\mathrm{miss}}$ with probability $1 - 1/N$. By triangle inequality,

$$\forall \boldsymbol{x} \in \mathbb{R}^F \ \forall i \leq N, \ \|\boldsymbol{x} - (\boldsymbol{x}^\star)_i\|_2^2 - \|\boldsymbol{x} - \widehat{\boldsymbol{x}}_i\|_2^2 \leq \|\widehat{\boldsymbol{x}}_i - (\boldsymbol{x}^\star)_i\|_2^2 \leq C_{1/N^3}^{\mathrm{miss}} \ \text{w.p.} \ 1 - 1/N \,.$$

Then, with probability $1 - 1/N$,

$$\forall \boldsymbol{x} \in \mathbb{R}^F, \qquad \|\boldsymbol{x} - (\boldsymbol{x}^\star)_i\|_2^2 \qquad \leq \|\boldsymbol{x} - \widehat{\boldsymbol{x}}_i\|_2^2 + C_{1/N^3}^{\mathrm{miss}}$$

$$\implies \qquad -\frac{1}{4h}\|\boldsymbol{x} - (\boldsymbol{x}^\star)_i\|_2^2 \qquad \geq -\frac{1}{4h}\|\boldsymbol{x} - \widehat{\boldsymbol{x}}_i\|_2^2 - \frac{C_{1/N^3}^{\mathrm{miss}}}{4h}$$

$$\implies \qquad e^{-\frac{1}{4h}\|\boldsymbol{x}-(\boldsymbol{x}^\star)_i\|_2^2} \qquad \geq e^{-\frac{C_{1/N^3}^{\mathrm{miss}}}{4h}} e^{-\frac{1}{4h}\|\boldsymbol{x}-\widehat{\boldsymbol{x}}_i\|_2^2}$$

$$\implies \qquad \sum_{i \leq N} e^{-\frac{1}{4h}\|\boldsymbol{x}-(\boldsymbol{x}^\star)_i\|_2^2} \qquad \geq e^{-\frac{C_{1/N^3}^{\mathrm{miss}}}{4h}} \sum_{i \leq N} e^{-\frac{1}{4h}\|\boldsymbol{x}-\widehat{\boldsymbol{x}}_i\|_2^2}$$

$$\implies \quad \log\left( \frac{\sum_{i \leq N} e^{-\frac{1}{4h}\|\boldsymbol{x}-\widehat{\boldsymbol{x}}_i\|_2^2}}{\sum_{i \leq N} e^{-\frac{1}{4h}\|\boldsymbol{x}-(\boldsymbol{x}^\star)_i\|_2^2}} \right) = \log\left( \frac{D_0(\boldsymbol{x})}{D_\star(\boldsymbol{x})} \right) \quad \leq \frac{C_{1/N^3}^{\mathrm{miss}}}{4h} \,.$$

Symmetrically (by switching the roles of $(\boldsymbol{x}^\star)_i$ and $\widehat{\boldsymbol{x}}_i$ in the previous inequalities), we obtain with probability $1 - 1/N$

$$
\begin{aligned}
\forall \boldsymbol{x} \in \mathbb{R}^F, \ \log\left(\frac{D_\star(\boldsymbol{x})}{D_0(\boldsymbol{x})}\right) &= -\log\left(\frac{D_0(\boldsymbol{x})}{D_\star(\boldsymbol{x})}\right) \leq C^{\text{miss}}_{1/N^3}/(4h) \\
\implies \left|\log\left(\frac{D_0(\boldsymbol{x})}{D_\star(\boldsymbol{x})}\right)\right| &\leq C^{\text{miss}}_{1/N^3}/(4h) \ .
\end{aligned}
$$

That is, for any $\boldsymbol{\alpha} \in \triangle_K$ and $X \in \mathbb{R}^{N \times F}$, with probability $1 - 1/N$

$$
|(G - G_\star)(\boldsymbol{\alpha}, X)| = \frac{1}{N} \sum_{i \leq N} \log\left(\frac{D_0(\texttt{Impute}(\boldsymbol{x}_i; \boldsymbol{\alpha}))}{D_\star(\texttt{Impute}(\boldsymbol{x}_i; \boldsymbol{\alpha}))}\right) - \log\left(\frac{D_0(\boldsymbol{x}_i)}{D_\star(\boldsymbol{x}_i)}\right) \leq \frac{C^{\text{miss}}_{1/N^3}}{2h} \ . \tag{6}
$$

Finally, we combine Equations 5-6 to obtain for any $\boldsymbol{\alpha} \in \triangle_K$, with probability $1 - 1/N$

$$
\sum_{s=1}^{t} G_\star(\boldsymbol{\alpha}, X^{s-1}) - G_\star(\boldsymbol{\alpha}^s, X^{s-1}) \leq \frac{C^{\text{miss}}_{1/N^3}}{h} t + \underbrace{2\delta_t \sqrt{t \log(K)} + 16\delta_t\left(2 + \frac{\log K}{3}\right)}_{=C^{\text{AH}}_G \sqrt{t}} \ .
$$

$\square$

## E  Regret analysis of PCGradF3I

**Lemma E.1.** *Any loss $\ell$ with a Lipschitz continuous gradient allows the use of PCGrad (Yu et al., 2020) combined with F3I. If $\nabla\ell$ is L-Lipschitz continuous with a finite $L > 0$ with respect to its single argument, then for any matrix $X \in \mathbb{R}^{N \times F}$, $\boldsymbol{\alpha} \mapsto \nabla_{\boldsymbol{\alpha}}\left((1-\beta)G(\boldsymbol{\alpha}, X) - \frac{\beta}{N}\sum_{i \leq N} \ell(\texttt{Impute}(\boldsymbol{x}^i, \boldsymbol{\alpha}))\right)$ is also Lipschitz continuous with a positive finite constant.*

*Proof.* Note that Proposition C.5 establishes that the gradient of $G$ with respect to $\boldsymbol{\alpha}$ is $H$-Lipschitz continuous with $H > 0$. Then for all $\boldsymbol{\alpha}, \boldsymbol{\alpha}' \in \triangle_K$

$$
\begin{aligned}
&\left\|\nabla_{\boldsymbol{\alpha}}\left((1-\beta)G(\boldsymbol{\alpha}, X) + \frac{\beta}{N}\sum_{i \leq N}\ell(\boldsymbol{x}_i(\boldsymbol{\alpha})) - \left((1-\beta)G(\boldsymbol{\alpha}', X) + \frac{\beta}{N}\sum_{i \leq N}\ell(\boldsymbol{x}_i(\boldsymbol{\alpha}'))\right)\right)\right\|_2 \\
\leq \ & (1-\beta)\|\nabla_{\boldsymbol{\alpha}}G(\boldsymbol{\alpha}, X) - \nabla_{\boldsymbol{\alpha}}G(\boldsymbol{\alpha}', X)\|_2 + \frac{\beta}{N}\sum_{i \leq N}\|\nabla_{\boldsymbol{\alpha}}\ell(\boldsymbol{x}_i(\boldsymbol{\alpha})) - \nabla_{\boldsymbol{\alpha}}\ell(\boldsymbol{x}_i(\boldsymbol{\alpha}'))\|_2 \\
\leq \ & H(1-\beta)\|\boldsymbol{\alpha} - \boldsymbol{\alpha}'\|_2 + \frac{L\beta}{N}\sum_{i \leq N}\|\boldsymbol{x}_i(\boldsymbol{\alpha}) - \boldsymbol{x}_i(\boldsymbol{\alpha}')\|_2 \\
\leq \ & (H(1-\beta) - L\beta\sqrt{KS})\|\boldsymbol{\alpha} - \boldsymbol{\alpha}'\|_2 \ .
\end{aligned}
$$

The last step holds because of the fact that, for any $i \leq N$, if $\mathcal{K}(\boldsymbol{x}_i, \widehat{X}, k)$ is the index of the $k^{\text{th}}$ nearest neighbor of $\boldsymbol{x}_i$ among $\{\widehat{\boldsymbol{x}}_1, \ldots, \widehat{\boldsymbol{x}}_N\} \subseteq \mathbb{R}^F$ and $\boldsymbol{x}_i^{\mathcal{M}_i}$ is the vector restricted to columns $f$ such that $x_i^f$ is

missing

$$\|\boldsymbol{x}_i(\boldsymbol{\alpha}) - \boldsymbol{x}_i(\boldsymbol{\alpha}')\|_2^2 = \Big\| \sum_{k \le K} (\alpha_k - \alpha'_k) \widehat{\boldsymbol{x}}^{\mathcal{M}_i}_{\mathcal{K}(\boldsymbol{x}_i, \widehat{X}, k)} \Big\|_2^2$$

$$= \sum_{f \in \mathcal{M}_i} \Big( \sum_{k \le K} (\alpha_k - \alpha'_k) \widehat{x}^f_{\mathcal{K}(\boldsymbol{x}_i, \widehat{X}, k)} \Big)^2$$

$$= \sum_{f \in \mathcal{M}_i} \langle \boldsymbol{\alpha} - \boldsymbol{\alpha}', [\widehat{x}^f_{\mathcal{K}(\boldsymbol{x}_i, \widehat{X}, 1)}, ..., \widehat{x}^f_{\mathcal{K}(\boldsymbol{x}_i, \widehat{X}, K)}]^\mathsf{T} \rangle^2$$

$$\le \|\boldsymbol{\alpha} - \boldsymbol{\alpha}'\|_2^2 \sum_{f \in \mathcal{M}_i} \|[\widehat{x}^f_{\mathcal{K}(\boldsymbol{x}_i, \widehat{X}, 1)}, ..., \widehat{x}^f_{\mathcal{K}(\boldsymbol{x}_i, \widehat{X}, K)}]^\mathsf{T} \|_2^2$$

$$= \|\boldsymbol{\alpha} - \boldsymbol{\alpha}'\|_2^2 \sum_{f \in \mathcal{M}_i} \sum_{k \le K} (\widehat{x}^f_{\mathcal{K}(\boldsymbol{x}_i, \widehat{X}, k)})^2$$

$$\le \|\boldsymbol{\alpha} - \boldsymbol{\alpha}'\|_2^2 \sum_{k \le K} \sum_{f \le F} (\widehat{x}^f_{\mathcal{K}(\boldsymbol{x}_i, \widehat{X}, k)})^2$$

$$= \|\boldsymbol{\alpha} - \boldsymbol{\alpha}'\|_2^2 \sum_{k \le K} \|\widehat{\boldsymbol{x}}_{\mathcal{K}(\boldsymbol{x}_i, \widehat{X}, k)}\|_2^2$$

$$\le \|\boldsymbol{\alpha} - \boldsymbol{\alpha}'\|_2^2 KS \text{ using Assumption B.6 } .$$

The first inequality is obtained by applying the Cauchy-Schwartz inequality $|\mathcal{M}_i|$ times, since the selection of neighbors does not depend on $\boldsymbol{\alpha}$. Note that $(x_i(\boldsymbol{\alpha}))^f = (x_i(\boldsymbol{\alpha}'))^f$ for any $f \notin \mathcal{M}_i$. $\qquad \square$

**Example 1.** A simple example of a convex loss function $\ell$ with a Lipschitz-continuous gradient function. *The pointwise log loss $\ell(\boldsymbol{x}) = -y \log C_{\boldsymbol{\omega}}(\boldsymbol{x})$ for the binary classification task is convex and such that $\nabla_{\boldsymbol{x}} \ell$ is Lipschitz continuous, where $y$ is the true class in $\{0, 1\}$ for sample $\boldsymbol{x}$ and $C_{\boldsymbol{\omega}} : \boldsymbol{x} \mapsto 1/(1 + \exp(-\boldsymbol{\omega}^\mathsf{T} \boldsymbol{x}))$ is the sigmoid function of parameter $\boldsymbol{\omega}$.*

*Proof.* $\ell$ is continuous and twice differentiable on $\mathbb{R}^F$. Knowing that $\nabla_{\boldsymbol{x}} C_{\boldsymbol{\omega}}(\boldsymbol{x}) = C_{\boldsymbol{\omega}}(\boldsymbol{x})(1 - C_{\boldsymbol{\omega}}(\boldsymbol{x})) \boldsymbol{\omega}^\mathsf{T}$, the Hessian matrix of $\ell$ in its single argument is

$$\forall \boldsymbol{x} \in \mathbb{R}^F \; \forall y \in \{0, 1\}, \; \nabla_{\boldsymbol{x}}^2 \ell(\boldsymbol{x}) = y C_{\boldsymbol{\omega}}(\boldsymbol{x})(1 - C_{\boldsymbol{\omega}}(\boldsymbol{x})) \boldsymbol{\omega} \boldsymbol{\omega}^\mathsf{T} .$$

In particular, it is easy to see that $\ell$ is convex, because for any $\boldsymbol{v} \in \mathbb{R}^F$,

$$\boldsymbol{v}^\mathsf{T} \nabla_{\boldsymbol{x}}^2 \ell(\boldsymbol{x}) \boldsymbol{v} = \underbrace{y C_{\boldsymbol{\omega}}(\boldsymbol{x})(1 - C_{\boldsymbol{\omega}}(\boldsymbol{x}))}_{\ge 0} (\boldsymbol{v}^\mathsf{T} \boldsymbol{\omega})^2 \ge 0 .$$

Then for any $\boldsymbol{x} \in \mathbb{R}^F$ and $y \in \{0, 1\}$,

$$\|\nabla_{\boldsymbol{x}}^2 \ell(\boldsymbol{x})\|_F^2 = \underbrace{y C_{\boldsymbol{\omega}}(\boldsymbol{x})(1 - C_{\boldsymbol{\omega}}(\boldsymbol{x}))}_{\le 1 \times 1/4} \sum_{f, f' \le F} (\omega^f)^2 \le \frac{1}{4} \|\boldsymbol{\omega}\|_F^2 .$$

Similarly to the proof of Proposition C.5, proving that $\nabla_{\boldsymbol{x}} \ell$ is Lipschitz continuous in each of its $F$ coordinates will be enough to prove that $\nabla_{\boldsymbol{x}} \ell$ is Lipschitz continuous as well. For any pair of points $\boldsymbol{x}_1, \boldsymbol{x}_2 \in \mathbb{R}^F$, we introduce the linear path $\gamma' : t \in [0, 1] \mapsto t\boldsymbol{x}_1 + (1 - t)\boldsymbol{x}_2$. For any $f \le F$, $t \in [0, 1] \mapsto \big( \nabla_{\boldsymbol{x}} \ell(\gamma'(t)) \big)^f \in \mathbb{R}$ is well-defined, continuous on the closed space $[0, 1]$, differentiable on $(0, 1)$. Then applying the mean value theorem to this function yields

$$|(\nabla_{\boldsymbol{x}} \ell(\boldsymbol{x}_1))^f - (\nabla_{\boldsymbol{x}} \ell(\boldsymbol{x}_2))^f| \le \| \sup_{\boldsymbol{x} \in \mathbb{R}^F} (\nabla_{\boldsymbol{x}}^2 \ell(\boldsymbol{x}))^f \|_2 \|\boldsymbol{x}_1 - \boldsymbol{x}_2\|_2 \le \frac{\|\boldsymbol{\omega}\|_2}{2} \|\boldsymbol{x}_1 - \boldsymbol{x}_2\|_2 .$$

Then $\nabla_{\boldsymbol{x}} \ell$ is Lipschitz-continuous with constant $\sqrt{\sum_{f \le F} \frac{1}{4} \|\boldsymbol{\omega}\|_2^2} = \frac{\|\boldsymbol{\omega}\|_F^2}{2} > 0$. $\qquad \square$

**Theorem E.2.** High-probability upper bound on the joint imputation-downstream task performance (Theorem 3.1). *Under Assumptions B.4-B.6, for any initial matrix $X \in (\mathbb{R} \cup \{N/A\})^{N \times F}$, convex pointwise loss $\ell$ such that $\nabla \ell$ is Lipschitz-continuous, and $\beta \in [0,1]$, under the conditions mentioned in Theorem 2 from Yu et al. (2020)*

$$\max_{\boldsymbol{\alpha} \in \triangle_K} \sum_{s=1}^{t} (1-\beta) \Big( G_\star(\boldsymbol{\alpha}, X^{s-1}) - G_\star(\boldsymbol{\alpha}^s, X^{s-1}) \Big) - \frac{\beta}{N} \sum_{i \leq N} \Big( \ell((\boldsymbol{x}^{s-1})_i(\boldsymbol{\alpha})) - \ell((\boldsymbol{x}^{s-1})_i(\boldsymbol{\alpha}^s)) \Big)$$
$$\leq C_{(G,\ell)}^{AH} \sqrt{t} + (1-\beta) H^{miss} h^{-1} t \,,$$

*with probability $1 - 1/N \in (0,1)$, where $H^{miss} = \mathcal{O}(F + \ln N)$ depends on the missingness mechanism and $C_{(G,\ell)}^{AH}$ is the constant related to AdaHedge being applied with gains $\bar{g}_s(\cdot)$.*

*Proof.* Similarly to the proof of Theorem D.1, the application of the AdaHedge regret bound (Technical lemma 2), and the gradient trick on the concave function $(1-\beta)G(\cdot, X) + \beta\ell(\cdot)$

$$\sum_{s=1}^{t} (1-\beta) \Big( G(\boldsymbol{\alpha}^{\mathrm{PC}}, X^{s-1}) - G((\boldsymbol{\alpha}^s)^{\mathrm{PC}}, X^{s-1}) \Big) + \frac{\beta}{N} \sum_{i \leq N} \Big( \ell((\boldsymbol{x}^{s-1})_i(\boldsymbol{\alpha}^s)^{\mathrm{PC}}) - \ell((\boldsymbol{x}^{s-1})_i(\boldsymbol{\alpha}^{\mathrm{PC}})) \Big)$$
$$\leq C_{(G,\ell)}^{\mathrm{AH}} \sqrt{t} \,,$$

where $\boldsymbol{\alpha}^{\mathrm{PC}}$ and $(\boldsymbol{\alpha}^s)^{\mathrm{PC}}$ are the parameters updated with PCGrad (Yu et al., 2020). Assuming the three conditions in Theorem 2 from Yu et al. (2020) are all satisfied, which only depend on functions $G$ and $\ell$, then for $\boldsymbol{\theta} \in \{\boldsymbol{\alpha}, \boldsymbol{\alpha}^s\}$

$$(1-\beta)G(\boldsymbol{\theta}, X^{s-1}) - \frac{\beta}{N} \sum_{i \leq N} \ell((\boldsymbol{x}^{s-1})_i(\boldsymbol{\theta})) \leq (1-\beta)G(\boldsymbol{\theta}^{\mathrm{PC}}, X^{s-1}) - \frac{\beta}{N} \sum_{i \leq N} \ell((\boldsymbol{x}^{s-1})_i(\boldsymbol{\theta}^{\mathrm{PC}})) \,.$$

Finally, we apply the pointwise approximation in high probability of $G_\star$ by $G$ (Corollary F.6 for $\delta = 1/N$) that yields for any $\boldsymbol{\alpha} \in \triangle_K$

$$\sum_{s=1}^{t} (1-\beta) \Big( G_\star(\boldsymbol{\alpha}, X^{s-1}) - G_\star(\boldsymbol{\alpha}^s, X^{s-1}) \Big) + \frac{\beta}{N} \sum_{i \leq N} \Big( \ell((\boldsymbol{x}^{s-1})_i(\boldsymbol{\alpha}^s)) - \ell((\boldsymbol{x}^{s-1})_i(\boldsymbol{\alpha})) \Big)$$
$$\leq C_{(G,\ell)}^{\mathrm{AH}} \sqrt{t} + (1-\beta) C_{1/N^3}^{\mathrm{miss}} h^{-1} t \,.$$

$\square$

To implement PCGrad-F3I, we also need to compute $\nabla_{\boldsymbol{\alpha}} \ell((\boldsymbol{x}^{s-1})_i(\boldsymbol{\alpha}^s))$ at each iteration $s$ for each point $i$. By the chain rule,

$$\nabla_{\boldsymbol{\alpha}} \ell((\boldsymbol{x}^{s-1})_i(\boldsymbol{\alpha}^s)) = \nabla_{\boldsymbol{x}} \ell(\boldsymbol{x})_{|\boldsymbol{x}=(\boldsymbol{x}^{s-1})_i(\boldsymbol{\alpha}^s)} \nabla_{\boldsymbol{\alpha}} (\boldsymbol{x}^{s-1})_i(\boldsymbol{\alpha})_{|\boldsymbol{\alpha}=\boldsymbol{\alpha}^s} \,,$$

In particular, we give the gradient at any $\boldsymbol{\alpha}$ and $i \leq N$ for the log-loss with sigmoid classifier below

**Lemma E.3.** Gradient $\nabla_{\boldsymbol{\alpha}} \ell((\boldsymbol{x}^{s-1})_i(\boldsymbol{\alpha}))$ for Example 1. *The gradient at any $\boldsymbol{\alpha}$ for the log-loss $\ell$ with sigmoid classifier $C_{\boldsymbol{\omega}}$ where the true class for sample $\boldsymbol{x} \in \mathbb{R}^F$ is $y \in \{0,1\}$ is*

$$\nabla_{\boldsymbol{\alpha}} \ell(\boldsymbol{x}(\boldsymbol{\alpha})) = -y(1 - C_{\boldsymbol{\omega}}(\boldsymbol{x}(\boldsymbol{\alpha}))) \boldsymbol{\omega}^{\mathsf{T}} \widetilde{Z}_s^{n_i} \,,$$

*where $\widetilde{Z}_s^{n_i} \in \mathbb{R}^{F \times K}$ is the matrix which $k^{th}$ column is defined as $(\widetilde{z}_s^{n_i})_f^k = 0$ if $m_i^f = 0$, and otherwise, $(\widetilde{z}_s^{n_i})_f^k$ is the value of the feature $f$ for the $k^{th}$ closest neighbor to $(\boldsymbol{x}^{s-1})_i$ among rows $\widehat{\boldsymbol{x}}_2, \ldots, \widehat{\boldsymbol{x}}_N\}$ of $\widehat{X}$ $\{\widehat{\boldsymbol{x}}_1, \widehat{\boldsymbol{x}}_2, \ldots, \widehat{\boldsymbol{x}}_N\}$ (see Lemma C.3).*

## F   Technical lemmas

We consider below for any $i \leq N$ the matrix $\widetilde{Z}^{n_i} \in \mathbb{R}^{F \times K}$ where the $k^{\text{th}}$ column of $\widetilde{Z}^{n_i}$ is defined as $(\widetilde{\boldsymbol{z}}^{n_i})^k_f = 0$ if $m^f_i = 0$, otherwise $(\widetilde{z}^{n_i})^k_f$ is the value of the feature $f$ for the $k^{\text{th}}$ closest neighbor to $(\boldsymbol{x}^{s-1})_i$ among rows of $\widehat{X}$ $\{\widehat{\boldsymbol{x}}_1, \widehat{\boldsymbol{x}}_2, \ldots, \widehat{\boldsymbol{x}}_N\}$. That is, $(\widetilde{\boldsymbol{z}}^{n_i})^k$ is equal to the $k^{\text{th}}$ closest neighbor of $\boldsymbol{x}_i$ (by increasing order of distance) on missing coordinates of $\boldsymbol{x}_i$, and equal to zero otherwise. To upper-bound norms involving matrix $\widetilde{Z}^{n_i}$, we use the following lemma

**Technical lemma 1.** Upper bound on $\ell_2$ norms on $\widetilde{Z}^{n_i}$. *For any $i \leq N$ and any vectors $\boldsymbol{v} \in \mathbb{R}^K$ and $\boldsymbol{u} \in \mathbb{R}^F$,*

$$\|\widetilde{Z}^{n_i}\boldsymbol{v}\|_2 \leq \sqrt{2KS}\|\boldsymbol{v}\|_2 \text{ and } \|\boldsymbol{u}^{\mathsf{T}}\widetilde{Z}^{n_i}\|_2 \leq \sqrt{2KS}\|\boldsymbol{u}\|_2 .$$

*Proof.* Using the Cauchy-Schwartz inequality applied respectively $F$ and K times and if $\|M\|_F = \sqrt{\sum_i \sum_j |m^j_i|^2} = \sqrt{\sum_i \|\boldsymbol{m}_i\|^2_2} = \sqrt{\sum_j \|\boldsymbol{m}^j\|^2_2}$ is the Frobenius matrix norm of matrix $M$, then $\|\widetilde{Z}^{n_i}\|^2_F \leq 2KS$ and

$$\|\widetilde{Z}^{n_i}\boldsymbol{v}\|^2_2 = \sum_{f \leq F}\langle(\widetilde{Z}^{n_i})^{\mathsf{T}}_f, \boldsymbol{v}\rangle^2 \leq \underbrace{\sum_{f \leq F}\|(\widetilde{Z}^{n_i})^{\mathsf{T}}_f\|^2_2}_{=_{\text{def}}\|\widetilde{Z}^{n_i}\|^2_F} \|\boldsymbol{v}\|^2_2 \leq 2KS\|\boldsymbol{v}\|^2_2 .$$

$$\|\boldsymbol{u}^{\mathsf{T}}\widetilde{Z}^{n_i}\|^2_2 = \sum_{k \leq K}\langle\boldsymbol{u}, (\widetilde{Z}^{n_i})^k\rangle^2 \leq \|\boldsymbol{u}\|^2_2 \underbrace{\sum_{k \leq K}\|(\widetilde{Z}^{n_i})^k\|^2_2}_{=_{\text{def}}\|\widetilde{Z}^{n_i}\|^2_F} \leq 2KS\|\boldsymbol{u}\|^2_2 .$$

$\square$

**Technical lemma 2.** Regret of AdaHedge. *On the online learning problem with K elements, using gains $\boldsymbol{\alpha} \mapsto g^s(\boldsymbol{\alpha}) \triangleq \sum_{k \leq K}\alpha_k U_k$ for $s \leq t$, and denoting $\delta_t \triangleq \max_{s \leq t}(\max_{k \leq K} U_k - \min_{q \leq K} U_q)$, the regret at time $t > 1$ incurred by AdaHedge with predictions $(\boldsymbol{\alpha}^s)_{s \leq t}$ is*

$$\max_{\boldsymbol{\alpha} \in \triangle_K}\sum_{s=1}^t g^s(\boldsymbol{\alpha}) - g^s(\boldsymbol{\alpha}^s) \leq 2\delta_t\sqrt{t\log(K)} + 16\delta_t(2 + \log(K)/3) .$$

*Proof.* This statement stems directly from Theorem 8 and Corollary 17 in De Rooij et al. (2014) applied to the loss $\ell^s = -g^s$, and using the fact that $\alpha_k \leq 1$ for any $k \leq K$. $\square$

In several proofs, we need an upper bound on $\|\widehat{\boldsymbol{x}}_j - (\boldsymbol{x}^\star)_i\|^2_2$ for any pair $i, j \leq N$, where $\widehat{X}$ is a convex weighted K-nearest neighbor-imputed matrix and $X^\star$ is the corresponding full matrix. This is our most important lemma for analyzing nearest-neighbor algorithms. This result still holds for any missingness mechanism such that the random variable $\widehat{x}^f_i - (x^\star)^f_i$ is a zero-mean subgaussian for any $i \leq N$ and $f \leq F$, independent across *features*. We show below that this statement includes all three mechanisms mentioned in Assumptions B.1-B.3 as described in Algorithm 4.

**Technical lemma 3.** Concentration bound on the norm of the difference between $\widehat{\boldsymbol{x}}_j$ and $(\boldsymbol{x}^\star)_i$. *Under any assumption in Assumptions B.1-B.3, if we consider a subset of features $\mathcal{F} \subseteq \{1, 2, \ldots, F\}$ such that $\boldsymbol{x}^{|\mathcal{F}}$ is the restriction of $\boldsymbol{x} \in \mathbb{R}^F$ to features in $\mathcal{F}$, then*

$$\forall c \geq \frac{4\ln N}{(\sigma^{miss})^2}\left(1 + \sqrt{1 + \frac{4(\sigma^{miss})^2|\mathcal{F}|}{\ln N}}\right) \forall i, j \leq N, \; \|\widehat{\boldsymbol{x}}^{|\mathcal{F}}_j - (\boldsymbol{x}^\star)^{|\mathcal{F}}_i\|^2_2 \leq (\sigma^{miss})^2(|\mathcal{F}| + c) ,$$

*with probability $1 - \exp\left(-\frac{(\sigma^{miss}c)^2}{4(8|\mathcal{F}|+c)} + 2\ln N\right) \in [0, 1]$, where $\sigma^{miss} \triangleq \max(\sigma_2, \sigma^{GSM})$, $A \triangleq \sum_{k \leq K}(\alpha_k)^2$ and*

$$\sigma_2 \triangleq \sigma\sqrt{1 + A} \quad (\text{Assumptions B.1-B.2}) \quad \text{and} \quad \sigma^{GSM} \triangleq \sigma\sqrt{\frac{1 + 3A}{3}} \quad (\text{Assumption B.3}) .$$

*Proof.* We summarized the procedure according to which the data matrices $X^\star$ and $\widehat{X}$ are generated in Algorithm 4. In particular, we assumed that $(x^\star)_i^f \sim_{\text{iid}} \mathcal{N}(\mu_f, \sigma^2)$ for any $i \leq N, f \leq F$ and fixed $\sigma > 0$ (Assumption B.4), $\mu = (\mu_1, \ldots, \mu_F) \in \mathbb{R}^F$, and that $K \leq \min_{f \leq F} |\{i \leq N \mid m_i^f = 0\}|$ (Assumption B.5), where the last term is the number of samples which do not miss the value of feature $f$ in the data set. Based on this, we can assume the following independence relationships for any $i, j \leq F$, where $i \neq j$, and $f, f' \leq F$, where $f \neq f'$

$$(x^\star)_i^f \perp\!\!\!\perp (x^\star)_j^f \tag{7}$$

$$\widehat{x}_i^f \perp\!\!\!\perp (x^\star)_i^f \mid m_i^f = 1 \tag{8}$$

$$\widehat{x}_i^f \not\perp\!\!\!\perp (x^\star)_i^f \mid m_i^f = 0 \quad (\text{since } (\widehat{x}_i^f \mid m_i^f = 0) = (x^\star)_i^f) \tag{9}$$

$$\widehat{x}_i^f \not\perp\!\!\!\perp \widehat{x}_j^f \mid m_i^f = 1, m_j^f = 1 \quad (\text{the two points can share a neighbor}) \tag{10}$$

$$\widehat{x}_i^f \not\perp\!\!\!\perp \widehat{x}_j^f \mid m_i^f = 1, m_j^f = 0 \quad ((\boldsymbol{x}^\star)_j \text{ can be a neighbor of } \boldsymbol{x}_i \text{ for } f) \tag{11}$$

$$\widehat{x}_i^f \perp\!\!\!\perp \widehat{x}_j^f \mid m_i^f = 0, m_j^f = 0 \tag{12}$$

$$(x^\star)_i^f \perp\!\!\!\perp (x^\star)_j^{f'} \tag{13}$$

$$(x^\star)_i^f \perp\!\!\!\perp (x^\star)_j^{f'} \tag{14}$$

$$(x^\star)_i^f \perp\!\!\!\perp (x^\star)_i^{f'} \text{ and } \widehat{x}_i^f \perp\!\!\!\perp \widehat{x}_i^{f'} . \tag{15}$$

What is the distribution of random variable $(\widehat{x}_i^f \mid m_i^f = m)$ for $m \in \{0, 1\}$? If $m_i^f = 0$, that is, if the value for the feature $f$ and sample $i$ is not missing in the input matrix $X$, then $(\widehat{x}_i^f \mid m_i^f = 0)$ follows the same law as $(x^\star)_i^f$. Otherwise, if $m_i^f = 1$, then at the initial imputation step, $\widehat{x}_i^f \mid m_i^f = 1$ is the convex-weighted mean of *exactly* K independent random variables of distribution $\mathcal{N}(\mu_f, \sigma^2)$ (by Independence 7). [2] All in all,

$$\left(\widehat{x}_i^f \mid m_i^f = 0\right) = (x^\star)_i^f \sim \mathcal{N}(\mu_f, \sigma^2) \quad \text{and} \quad \left(\widehat{x}_i^f \mid m_i^f = 1\right) \sim \mathcal{N}\!\left(\mu_f, \sigma^2 \sum_{k \leq K} (\alpha_k)^2\right) . \tag{16}$$

Let $A \triangleq \sum_{k \leq K} (\alpha_k)^2$, and denote $\sigma_0 \triangleq \sigma$, $\sigma_1 \triangleq \sigma\sqrt{A}$, $\sigma_2 \triangleq \sqrt{\sigma_0^2 + \sigma_1^2}$, $\sigma_3(k) \triangleq \sigma_0\sqrt{A - 2\alpha_k + 1}$ for $k \leq K$ and $p_{if}^{\text{miss}} \triangleq \mathbb{P}(m_i^f = 1)$. Let us now consider the distribution of the random variable $\left(\widehat{x}_i^f - (x^\star)_i^f\right)$ for any $i, f$

$$\forall i \leq N, \forall f \leq F, \quad \left(\widehat{x}_i^f - (x^\star)_i^f \mid m_i^f = 0\right) = 0$$
$$\left(\widehat{x}_i^f - (x^\star)_i^f \mid m_i^f = 1\right) \sim \mathcal{N}(0, \sigma_2^2) \quad (\text{by Independence 8}) .$$

Similarly, for any $i, j \leq N$

$$\forall j \neq i, \forall f \leq F, \; \left(\widehat{x}_j^f - (x^\star)_i^f \mid m_j^f = 0\right) \sim \mathcal{N}(0, 2\sigma^2) \quad (\text{by Independence 7})$$
$$\left(\widehat{x}_j^f - (x^\star)_i^f \mid m_j^f = 1\right) \sim \begin{cases} \mathcal{N}(0, \sigma_2^2) & \text{if } \forall k \leq K, i \neq \mathcal{K}(\boldsymbol{x}_j, \widehat{X}, k) \\ \mathcal{N}(0, \sigma_3(k_{i,j,f})^2) & \text{otherwise} \end{cases},$$

because in the last case, $\widehat{x}_j^f - (x^\star)_i^f = \sum_{q \neq k_{i,j,f}} \alpha_q (x^\star)_{\mathcal{K}(\boldsymbol{x}_j, f, \widehat{X}, q)}^f + (\alpha_{k_{i,j,f}} - 1)(x^\star)_i^f$ where $k_{i,j,f}$ is the proximity rank to data point $f$ with respect to $f$ of data point $i$. Let us denote now $p_{ij} \triangleq \mathbb{P}\big(\forall k \leq K, i \neq$

---

[2]Due to the upper bound on K (Assumption B.5).

$\mathcal{K}(\boldsymbol{x}_j, \widehat{X}, k) \mid m_j^f = 1)$. The law of total probability gives

$$\forall i \leq N, \ \forall f \leq F, \ \forall x \neq 0, \qquad \mathbb{P}\Big( \big( \widehat{x}_i^f - (x^\star)_i^f \big) = x \Big) = p_{if}^{\text{miss}} \mathcal{N}(x; 0, \sigma_2^2) \tag{17}$$

$$\mathbb{P}\Big( \big( \widehat{x}_i^f - (x^\star)_i^f \big) = 0 \Big) = 1 + p_{if}^{\text{miss}} \underbrace{\Big( \mathcal{N}(0; 0, \sigma_2^2) - 1 \Big)}_{=1/\sqrt{2\pi\sigma_2^2}-1}$$

$$\forall i \neq j, \ \forall f \leq F, \ \forall x \in \mathbb{R}, \qquad \mathbb{P}\Big( \big( \widehat{x}_j^f - (x^\star)_i^f \big) = x \Big) = (1 - p_{if}^{\text{miss}}) \mathcal{N}(0, 2\sigma_0^2) \tag{18}$$

$$+ \quad p_{if}^{\text{miss}} \Big( p_{ij} \mathcal{N}(x; 0, \sigma_2^2) + (1 - p_{ij}) \mathcal{N}(x; 0, \sigma_3(k_{i,j,f})^2) \Big) .$$

Then, we show that the random variable $\widehat{x}_j^f - (x^\star)_i^f$ is a zero-mean $\sigma^{\text{miss}}$-subgaussian variable under Assumptions B.1-B.3, where $\sigma^{\text{miss}}$ depends on the missingness mechanism and the initial imputation algorithm. We recall that a zero-mean $\sigma$-subgaussian variable $X$ satisfies $\mathbb{E}[e^{\lambda X}] \leq e^{\sigma^2 \lambda^2 / 2}$ for all $\lambda \in \mathbb{R}$, with equality for any zero-mean Gaussian random variable of variance $\sigma^2$.

**Lemma F.1.** $\widehat{x}_j^f - (x^\star)_i^f$ *is a zero-mean $\sigma_2$-subgaussian random variable under Assumption B.1. For all $i \neq j \leq N$, $f \leq F$, under the MCAR assumption, $p_{jf}^{miss} = p \in (0, 1)$ is a constant and then $\widehat{x}_j^f - (x^\star)_i^f$ is a zero-mean $\sigma_2$-subgaussian random variable.*

*Proof.* First, let us denote $X_{ij}^f \triangleq \widehat{x}_j^f - (x^\star)_i^f$ for $i, j \leq N$ and $f \leq F$. Then using 17, it is clear that $X_{ii}^f$ is centered for any $i \leq N$. Similarly, due to 18 for any $i \neq j \leq N$

$$\mathbb{E}[X_{ij}^f] = (1 - p) \underbrace{\mathbb{E}[X_{ij}^f \mid m_j^f = 0]}_{=0} + p \underbrace{\mathbb{E}[X_{ij}^f \mid m_j^f = 1]}_{=0} = 0 .$$

Moreover, $\forall \lambda \in \mathbb{R}$, $\mathbb{E}[e^{\lambda X_{ii}^f}] = \mathbb{P}(m_i^f = 0) \cdot e^{\lambda \cdot 0} + \mathbb{P}(m_i^f = 1) \cdot \mathbb{E}_{x \sim \mathcal{N}(0, \sigma_2^2)}[e^{\lambda x}] = (1 - p) + p e^{\frac{\sigma_2^2 \lambda^2}{2}}$ so

$$\forall \lambda \in \mathbb{R}, \ \exp(\sigma_2^2 \lambda^2 / 2) - \mathbb{E}[e^{\lambda X_{ii}^f}] = (1 - p)(e^{\sigma_2^2 \lambda^2} - 1) \geq 0 .$$

Second, we notice that $\sigma_1 \leq \sigma_0 \leq \sigma_0 \sqrt{2} \leq \sigma_2$ (since $\boldsymbol{\alpha} \in \triangle_K$). It is easy to see that any $\sigma'$-subgaussian variable is also a $\sigma''$-subgaussian variable, where $\sigma' \leq \sigma''$. Then $X_{ij}^f \mid m_j^f = 1$ and $X_{ij}^f \mid m_j^f = 0$ are both $\sigma_2$-subgaussian. Then $X_{ij}^f$ is $\sigma_2$-subgaussian for any $i, j \leq N$. □

**Lemma F.2.** $\widehat{x}_j^f - (x^\star)_i^f$ *is a zero-mean $\sigma_2$-subgaussian random variable under Assumption B.2. For all $i \neq j \leq N$, $f \leq F$, under the MAR assumption, the missingness depends on a fixed subset of always observed values $F^O \subset \{1, 2, \ldots, F\}$: $\mathbb{P}(m_j^f = 1 \mid x_j^\star) = h\big( (x^\star)_j^{F^O}, f \big)$ where $(x^\star)_i^{F^O}$ is the restriction of $(x^\star)_i$ to rows in $F^O$ and $h$ some deterministic function.* [3] *Then $\widehat{x}_j^f - (x^\star)_i^f$ is a zero-mean $\sigma_2$-subgaussian random variable.*

*Proof.* Under Assumption B.2, for all $j \leq N$ and for all $f \in F^O$, $p_{jf}^{\text{miss}} = 0$ and for all $f \notin F^O$,

$$p_{jf}^{\text{miss}} = \int_{x_{f'}, f' \in F^O} h\big( [(x^\star)_j^{f'} = x^{f'}, \ f' \in F^O], f \big) \Pi_{f' \in F^O} \mathcal{N}(x^{f'}; \mu_{f'}, \sigma^2) dx .$$

By Independence 14 and similarly to the proof of Lemma F.1, $X_{ji}^f$ is then a zero-mean $\sigma_2$-subgaussian random variable for any $f \leq F$ and $i, j \leq N$. □

**Lemma F.3.** $\widehat{x}_j^f - (x^\star)_i^f$ *is a zero-mean $\sigma^{\text{GSM}}$-subgaussian random variable under Assumption B.3. For all $i \neq j \leq N$, $f \leq F$, under the Gaussian self-masking mechanism from Assumption 4 in Le Morvan et al. (2020), the probability of $x_i^f$ missing is given by*

$$\forall x \in \mathbb{R}, \ \mathbb{P}(m_i^f = 1 \mid (x^\star)_i^f = x) = p_i^{miss}(x, f) = K_f \exp\Big( -\frac{(x - \mu_f)^2}{\sigma^2} \Big) \text{ with } K_f \in (0, 1) .$$

---

[3] With an abuse in notation as we denote $x_i^\star$ both the random variable and its realization.

Then $\widehat{x}_j^f - (x^\star)_i^f$ is a zero-mean $\sigma^{GSM}$-subgaussian random variable, where $\sigma^{GSM} \triangleq \sigma\sqrt{\frac{1+3A}{3}}$.

*Proof.* For all $i \leq N$, $f \leq F$, and for any $x \neq 0$, by the law of total probability

$$\mathbb{P}(X_{ii}^f = x) = \mathbb{P}(X_{ii}^f = x|m_i^f = 1)\mathbb{P}(m_i^f = 1) + \underbrace{\mathbb{P}(X_{ii}^f = x|m_i^f = 0)}_{=0 \text{ because } x \neq 0}\mathbb{P}(m_i^f = 0) .$$

Then since $\mathbb{P}(X_{ii}^f = x|m_i^f = 1, (x^\star)_i^f = y) = \mathbb{P}(\widehat{x}_i^f = x + y|m_i^f = 1)$, using 16

$$\mathbb{P}(X_{ii}^f = x) = \int_{y \in \mathbb{R}} \mathbb{P}(X_{ii}^f = x|m_i^f = 1, (x^\star)_i^f = y)\mathbb{P}(m_i^f = 1|(x_i^\star)^f = y)\mathbb{P}((x^\star)_i^f = y)dy$$

$$= \int_{y \in \mathbb{R}} \mathcal{N}(x + y; \mu_f, \sigma^2 A)p_i^{\text{miss}}(y, f)\mathcal{N}(y; \mu_f, \sigma^2)dy \triangleq I_{i,f}(x)$$

$$\forall x \in \mathbb{R}, I_{i,f}(x) = \int_{y \in \mathbb{R}} \frac{K_f}{2\pi\sigma^2\sqrt{A}} \exp\left(-\frac{(x + y - \mu_f)^2}{2\sigma^2 A}\right) \exp\left(-\frac{(y - \mu_f)^2}{\sigma^2}\right) \exp\left(-\frac{(y - \mu_f)^2}{2\sigma^2}\right) dy$$

$$= \frac{K_f}{2\pi\sigma^2\sqrt{A}} \int_{y \in \mathbb{R}} \exp\left(-\frac{x^2 + 2x(y - \mu_f) + (y - \mu_f)^2 + 2A(y - \mu_f)^2 + A(y - \mu_f)^2}{2\sigma^2 A}\right) dy$$

$$= \frac{K_f}{2\pi\sigma^2\sqrt{A}} \int_{y \in \mathbb{R}} \exp\left(-\frac{1}{2\sigma^2 A}\left(x^2 + 2x(y - \mu_f) + (1 + 3A)(y - \mu_f)^2\right)\right) dy$$

$$= \frac{K_f}{2\pi\sigma^2\sqrt{A}} \int_{y \in \mathbb{R}} \exp\left(-\frac{1}{2\sigma^2 A}\left((\sqrt{1 + 3A}(y - \mu_f) + x/\sqrt{1 + 3A})^2 + 3A/(1 + 3A)x^2\right)\right) dy$$

$$= \frac{K_f}{2\pi\sigma^2\sqrt{A}} \exp\left(-\frac{3}{2\sigma^2(1 + 3A)}x^2\right) \int_{y \in \mathbb{R}} \exp\left(-\frac{1}{2\sigma^2 A}\left(\sqrt{1 + 3A}(y - \mu_f) + x/\sqrt{1 + 3A}\right)^2\right) dy$$

$$= \frac{K_f}{2\pi\sigma^2\sqrt{A}} \exp\left(-\frac{3}{2\sigma^2(1 + 3A)}x^2\right) \int_{y \in \mathbb{R}} \exp\left(-\frac{1 + 3A}{2\sigma^2 A}\left(y - \mu_f + x/(1 + 3A)\right)^2\right) dy$$

$$= \frac{K_f}{2\pi\sigma^2\sqrt{A}} \exp\left(-\frac{3}{2\sigma^2(1 + 3A)}x^2\right)\sqrt{\frac{2\pi\sigma^2 A}{1 + 3A}} = \frac{K_f}{\sqrt{2\pi\sigma^2(1 + 3A)}} \exp\left(-\frac{3}{2\sigma^2(1 + 3A)}x^2\right)$$

When $x = 0$, $X_{ii}^f$ follows the second law described at 17 and then

$$\mathbb{P}(X_{ii}^f = 0) = \mathbb{P}(X_{ii}^f = 0|m_i^f = 1)\mathbb{P}(m_i^f = 1) + \underbrace{\mathbb{P}(X_{ii}^f = 0|m_i^f = 0)}_{=1}\underbrace{\mathbb{P}(m_i^f = 0)}_{=1 - \mathbb{P}(m_i^f = 1)}$$

$$= \int_{y \in \mathbb{R}} \mathbb{P}(X_{ii}^f = 0|m_i^f = 1, (x^\star)_i^f = y)\mathbb{P}(m_i^f = 1 \mid (x^\star)_i^f = y)\mathbb{P}((x^\star)_i^f = y)dy$$

$$+ 1 - \int_{y \in \mathbb{R}} \mathbb{P}(m_i^f = 1 \mid (x^\star)_i^f = y)\mathbb{P}((x^\star)_i^f = y)dy$$

$$= \int_{y \in \mathbb{R}} \mathcal{N}(y; \mu_f, \sigma^2 A)p_i^{\text{miss}}(y, f)\mathcal{N}(y; \mu_f, \sigma^2)dy + 1 - \int_{y \in \mathbb{R}} p_i^{\text{miss}}(y, f)\mathcal{N}(y; \mu_f, \sigma^2)dy$$

$$= I_{i,f}(0) + 1 - \frac{K_f}{\sqrt{2\pi\sigma^2}} \int_{y \in \mathbb{R}} e^{-\frac{3}{2\sigma^2}(y - \mu_f)^2} dy$$

$$= \frac{K_f}{\sqrt{2\pi\sigma^2(1 + 3A)}} + 1 - \frac{K_f}{\sqrt{2\pi\sigma^2}}\sqrt{\frac{2\pi\sigma^2}{3}} = 1 + K_f\left(\frac{1}{\sqrt{2\pi\sigma^2(1 + 3A)}} - \frac{1}{\sqrt{3}}\right) .$$

That is

$$\forall x \neq 0, \ \mathbb{P}(\widehat{x}_i^f - (x^\star)_i^f = x) \ = \ \frac{K_f}{\sqrt{3}} \times \mathcal{N}(x; 0, (\sigma^{\text{GSM}})^2) \text{ where } \sigma^{\text{GSM}} \triangleq \sigma\sqrt{\frac{1 + 3A}{3}} \qquad (19)$$

$$\mathbb{P}(\widehat{x}_i^f - (x^\star)_i^f = 0) \ = \ 1 + K_f\left(\frac{1}{\sqrt{2\pi\sigma^2(1 + 3A)}} - \frac{1}{\sqrt{3}}\right) .$$

For the zero-mean variable X following the distribution described in 19, the moment-generating function (MGF) of $X$ is given by

$$\mathbb{E}[e^{tX}] = \int \mathbb{P}(X = x)e^{tx}dx$$

$$= \int_{x \neq 0} e^{tx} \frac{K_f}{\sqrt{6\pi(\sigma^{\text{GSM}})^2}} \exp\left(-\frac{x^2}{2(\sigma^{\text{GSM}})^2}\right) dx$$

$$= \frac{K_f}{\sqrt{3}} \mathbb{E}_{Y \sim \mathcal{N}(0, (\sigma^{\text{GSM}})^2)}[e^{tY}] = \frac{K_f}{\sqrt{3}} \exp\left(\frac{(\sigma^{\text{GSM}})^2 t^2}{2}\right) .$$

Choose $s = \sigma^{\text{GSM}}$. Then,

$$\exp(\frac{s^2 t^2}{2}) - \mathbb{E}[e^{tX}] = \exp\left(\frac{(\sigma^{\text{GSM}})^2 t^2}{2}\right) - \frac{K_f}{\sqrt{3}} \exp\left(\frac{(\sigma^{\text{GSM}})^2 t^2}{2}\right)$$

$$= \exp\left(\frac{(\sigma^{\text{GSM}})^2 t^2}{2}\right) \left(1 - \frac{K_f}{\sqrt{3}}\right) .$$

Clearly the minimum value, achieved at $t = 0$, is $1 - \frac{K_f}{\sqrt{3}} \geq 0$ since $K_f \in (0, 1)$ by definition. All in all, $X$ is a zero-mean $\sigma^{\text{GSM}}$-subgaussian variable. $\qquad \square$

**Lemma F.4.** If $X$ is $s$-subgaussian, then $\beta X$ is $\beta s$-gaussian when $\beta > 0$. *For any $\beta > 0$ and $X$ zero-mean $s$-subgaussian, $\beta X$ is zero-mean $\beta s$-subgaussian.*

*Proof.* If $X$ is a zero-mean $s$-subgaussian, then $\mathbb{E}[\beta X] = 0$ and

$$\forall t > 0, \ \mathbb{P}(|\beta X| \geq t) = \mathbb{P}(|X| \geq \beta^{-1}t) \leq 2\exp\left(-\frac{t^2}{2(\beta s)^2}\right) ,$$

and using Proposition 2.5.2 from Vershynin (2018), $\beta X$ is a (zero-mean) $\beta s$-subgaussian variable. $\qquad \square$

Finally, we determine a concentration bound on $\|\widehat{x}_j^{|\mathcal{F}} - (x^\star)_i^{|\mathcal{F}}\|_2^2$ for any $i, j \leq N$ and $\mathcal{F} \subseteq \{1, 2, \ldots, F\}$ under any of the Assumptions B.1-B.3. Let us set $\sigma^{\text{miss}} \triangleq \max(\sigma_2, \sigma^{\text{GSM}})$ and introduce the $|\mathcal{F}|$-dimensional random vector $\widetilde{X}_{ji}^{|\mathcal{F}} \triangleq \widehat{x}_j^{|\mathcal{F}} - (x^\star)_i^{|\mathcal{F}}$ for any $i, j \leq N$. The $|\mathcal{F}|$ coefficients of $\widetilde{X}_{ji}^{|\mathcal{F}}$ follow the distribution described in Equations 17-18. Then, the random vector $(\sigma^{\text{miss}})^{-1}\widetilde{X}_{ji}^{|\mathcal{F}}$ has $|\mathcal{F}|$ independent 1-subgaussian zero-mean coefficients. The independence holds by Independence 15 and 7. The coefficients are 1-subgaussian due to Lemma F.4. Using Theorem 3.1.1 from (Vershynin, 2018), which relies on Bernstein's inequality applied to the random variables $(\sigma^{\text{miss}})^{-1}\widetilde{X}_{ji}^f$, for any feature $f \in \mathcal{F}$ and samples $i, j \leq N$, for any constant $c > 0$

$$\mathbb{P}[(\sigma^{\text{miss}})^{-2}\|\widetilde{X}_{ji}^{|\mathcal{F}}\|_2^2 \geq |\mathcal{F}| + c] \leq \exp\left(-\frac{c^2}{4(8|\mathcal{F}| + c)}\right)$$

$$\implies \mathbb{P}[\|\widehat{x}_j^{|\mathcal{F}} - (x^\star)_i^{|\mathcal{F}}\|_2^2 \geq (\sigma^{\text{miss}})^2(|\mathcal{F}| + c)] \leq \exp\left(-\frac{(\sigma^{\text{miss}}c)^2}{4(8|\mathcal{F}| + c)}\right)$$

$$\implies \mathbb{P}[\cup_{i,j \leq N}\{\|\widehat{x}_j^{|\mathcal{F}} - (x^\star)_i^{|\mathcal{F}}\|_2^2 \geq (\sigma^{\text{miss}})^2(|\mathcal{F}| + c)\}] \leq \exp\left(-\frac{(\sigma^{\text{miss}}c)^2}{4(8|\mathcal{F}| + c)} + 2\ln N\right) ,$$

by applying an union bound on $\{1, 2, \ldots, N\}^2$. And then for any positive constant $c$ such that $2\ln N - (\sigma^{\text{miss}}c)^2/(4(8|\mathcal{F}| + c)) \leq 0$,

$$\mathbb{P}[\cap_{i,j \leq N} \|\widehat{\boldsymbol{x}}_j^{|\mathcal{F}} - (\boldsymbol{x}^\star)_i^{|\mathcal{F}}\|_2^2 \leq (\sigma^{\text{miss}})^2(|\mathcal{F}| + c)]$$
$$= 1 - \mathbb{P}[\cup_{i,j \leq N} \|\widehat{\boldsymbol{x}}_j^{|\mathcal{F}} - (\boldsymbol{x}^\star)_i^{|\mathcal{F}}\|_2^2 \geq (\sigma^{\text{miss}})^2(|\mathcal{F}| + c)]$$
$$\geq 1 - \exp\left(-\frac{(\sigma^{\text{miss}}c)^2}{4(8|\mathcal{F}| + c)} + 2\ln N\right) .$$

A positive such $c$ always exists, which can be shown by choosing $c$ such that

$$c \geq \frac{4\ln N}{(\sigma^{\text{miss}})^2}\left(1 + \sqrt{1 + 4(\sigma^{\text{miss}})^2|\mathcal{F}|/\ln N}\right) > 0 \implies 2\ln N - (\sigma^{\text{miss}}c)^2/(4(8|\mathcal{F}| + c)) \leq 0 .$$

Note that, similarly, by union bound on $\{1, 2, \ldots, N\}$, for such a $c$,

$$\mathbb{P}[\cap_{i \leq N} \|\widehat{\boldsymbol{x}}_i^{|\mathcal{F}} - (\boldsymbol{x}^\star)_i^{|\mathcal{F}}\|_2^2 \leq (\sigma^{\text{miss}})^2(|\mathcal{F}| + c)] \geq 1 - \exp\left(-\frac{(\sigma^{\text{miss}}c)^2}{4(8|\mathcal{F}| + c)} + \ln N\right) \in [0, 1] .$$

$\square$

**Corollary F.5.** *First concentration bound on $\|\widehat{\boldsymbol{x}}_j - (\boldsymbol{x}^\star)_i\|_2^2$. Under any assumption in Assumptions B.1-B.3, then*

$$\forall c \geq \frac{4\ln N}{(\sigma^{miss})^2}\left(1 + \sqrt{1 + \frac{4(\sigma^{miss})^2 F}{\ln N}}\right) \quad \forall i, j \leq N, \quad \|\widehat{\boldsymbol{x}}_j - (\boldsymbol{x}^\star)_i\|_2^2 \leq (\sigma^{miss})^2(F + c) ,$$

*with probability $1 - \exp\left(-\frac{(\sigma^{miss}c)^2}{4(8F+c)} + 2\ln N\right) \in [0, 1]$, where $\sigma^{miss} \triangleq \max(\sigma_2, \sigma^{GSM}) \propto \sigma$ is defined in Technical lemma 3.*

*Proof.* This statement holds by application of Lemma 3 with $\mathcal{F} = \{1, 2, \ldots, F\}$. $\square$

**Corollary F.6.** *Second concentration bound on $\|\widehat{\boldsymbol{x}}_j - (\boldsymbol{x}^\star)_i\|_2^2$ and $\|\widehat{\boldsymbol{x}}_i - (\boldsymbol{x}^\star)_i\|_2^2$. Under any assumption in Assumptions B.1-B.3, for $\sigma^{miss} \triangleq \max(\sigma_2, \sigma^{GSM}) \propto \sigma$ (Technical lemma 3), let us denote*

$$C_\delta^{miss} \triangleq (\sigma^{miss})^2 F + 2\ln(1/\delta)\left(1 + \sqrt{1 + 8(\sigma^{miss})^2 F/\ln(1/\delta)}\right) \text{ for } \delta \leq 1/N .$$

*Then, with probability $1 - \delta \in (0, 1)$, for all $i, j \leq N$, $\|\widehat{\boldsymbol{x}}_j - (\boldsymbol{x}^\star)_i\|_2^2 \leq C_{\delta/N^2}^{miss}$.*

*Proof.* We solve the following equation in $c > 0$ from Corollary F.5,

$$\delta = \exp\left(-\frac{(\sigma^{\text{miss}}c)^2}{4(8F + c)} + 2\ln N\right) \Leftrightarrow -(\sigma^{\text{miss}})^2 c^2 + (4\ln(N^2/\delta))c + 32F\ln(N^2/\delta) = 0 .$$

This equation has two real roots, one positive root being

$$c_\delta \triangleq \frac{4\ln(N^2/\delta)}{(\sigma^{\text{miss}})^2}\left(1 + \sqrt{1 + 8(\sigma^{\text{miss}})^2 F/\ln(N^2/\delta)}\right) .$$

Applying Corollary F.5 with $c = c_\delta$ when $\delta \in (0, 1)$ yields for any $j, i \leq N$, with probability $1 - \frac{\delta}{N^2} \in (0, 1)$,

$$\|\widehat{\boldsymbol{x}}_j - (\boldsymbol{x}^\star)_i\|_2^2 \leq C_{\delta/N^2}^{\text{miss}} .$$

Applying an upper bound on $\{1, 2, \ldots, N\}^2$ yields the expected result. $\square$

# G   Experimental study

This section dwells on details about the experimental setting and features full numerical tables of the corresponding figures in Section 4. Hyperparameter values for all the nearest neighbour approaches and baselines are reported in Table 2. We use $K = 5$ neighbors here for all the nearest neighbor imputers.

Table 2: Hyperparameters for nearest-neighbor approaches and baselines, unless otherwise specified. K is the number of neighbors in nearest-neighbor approaches. The names of the hyperparameters match the corresponding argument names in their implementation in Python (official, in scikit-learn (Pedregosa et al., 2011) or in HyperImpute (Jarrett et al., 2022), if present).

| Imputer | Hyperparameters |
|---|---|
| F3I | `n_neighbors`$= K$, `max_iter`$= 500$, $\eta= 0.001$, `S`$= 1$, $\beta= 0$ `max_iters`$= 500$, $\beta= 0$ |
| KNN (Troyanskaya et al., 2001) (uniform) | `n_neighbors`$= K$, `distance`='nan_euclidean' |
| KNN (Troyanskaya et al., 2001) (distance) | `n_neighbors`$= K$, `distance`='nan_euclidean' |
| GAIN (Yoon et al., 2018a) | `batch_size`$= 128$, `n_epochs`$= 100$, `hint_rate`$= 0.8$, `loss_alpha`$= 10$ |
| GRAPE (You et al., 2020) | `node_dim`$= 64$, `edge_dim`$= 16$, `nepochs`$= 20,000$ |
| HyperImpute (Jarrett et al., 2022) | `imputation_order`$= 2$, `baseline_imputer`$= 0$, `optimizer`='simple', `class_threshold`$= 5$, `optimize_thresh`$= 5,000$, `n_inner_iter`$= 40$, `select_patience`$= 5$ |
| MIRACLE (Kyono et al., 2021) | `lr`$= 0.001$, `batch_size`$= 1,024$, `num_outputs`$= 1$, `n_hidden`$= 32$, `reg_lambda`$= 1$, `reg_beta`$= 1$, `reg_m`$= 1.0$, `window`$= 10$, `max_steps`$= 400$, `seed_imputation`='mean' |
| NewImp (Chen et al., 2024) | `entropy_reg`$= 10$, `eps`$= 0.01$, `lr`$= 0.01$, `opt`='Adam', `niter`$= 50$, `kernel_func`='xRBF', `mlp_hidden`$= [256, 256]$, `score_net_epoch`$= 2,000$, `score_net_lr`$= 0.001$, `score_loss_type`='dsm', `bandwidth`$= 10$, `sampling_step`$= 500$, `batchsize`$= 128$, `n_pairs`$= 1$, `noise`$= 0.1$, `scaling`$= 0.9$ |
| Remasker (Du et al., 2023) | `batch_size`$= 64$, `max_epochs`$= 600$, `accum_iter`$= 1$, `mask_ratio`$= 0.5$, `embed_dim`$= 32$, `depth`$= 6$, `decoder_depth`$= 4$, `num_heads`$= 4$, `mlp_ratio`$= 4$, `encode_func`='linear', `weight_decay`$= 0.05$, `lr`=None, `blr`$= 0.001$, `min_lr`$= 0.00001$, `warmup_epochs`$= 40$ |
| TDM (Zhao et al., 2023) | `k`$= 2$, `depth`$= 3$, `im_lr`$= 0.01$, `proj_lr`$= 0.01$, `opt`='RMSprop', `niter`$= 2,000$, `batchsize`$= 128$, `n_pairs`$= 1$, `noise`$= 0.1$ |
| RF-GAP (Rhodes et al., 2023) | `prox_method`='rfgap' |

Note that, due to an error arising in the Gottlieb data set, we changed the implementation of TDM. The bug was that when two distinct samples were close enough together, the automatic differentiation of the Euclidean distance gave rise to NaN values. This error was fixed by modifying line 100 in `tdm.py` from `M_p = torch.cdist(X1_p, X2_p, p=2)` to `M_p = torch.cdist(X1_p, X2_p, p=2, compute_mode='donot_use_mm_for_euclid_dist')`. Our results are obtained with this modified implementation.

The implementations of the MCAR and MAR mechanisms come from Muzellec et al. (2020a) (with `opt`='logistic'). For a MCAR mechanism or MAR mechanism implemented by Muzellec et al. (2020a), it corresponds to the random probability of missing data. For the MNAR Gaussian self-masking and feature $f$, using the notation in Assumption B.3, we sample $K_f$ from $\mathcal{N}(\frac{3.5}{3}p^{\text{miss}}(1 - p^{\text{miss}}), 0.1)$ and we clip $K_f$ in $[0.01, 0.99]$ whenever necessary. Empirically, as long as $p^{\text{miss}}$, the expected missingness frequency is

not too extreme (*i.e.*, far from the bounds of $[0,1]$), the empirical probability of missingness is close to $p^{\mathrm{miss}}$. However, controlling this probability more finely in the case of a MNAR mechanism might break the not-missing-at-random property.

The experiments were run on remote cluster servers (processor QEMU Virtual v2.5+, 48 cores @2.20GHz, RAM 500GB). No GPU was used in our experiments. We also briefly dwell on the time complexity of the iterative imputation rounds in F3I (Algorithm 2). The time complexity of running the kNN imputer (Troyanskaya et al., 2001) with uniform weights and building the k-d tree on $N$ $F$-dimensional points is $\mathcal{O}(FN \log N)$, both steps being performed once. For each input point $\boldsymbol{x}$, Algorithm 1 first queries K nearest neighbors (each query has a time complexity of $\mathcal{O}(\log N)$) and then performs the imputation in at most $FK$ operations, for a total time complexity across all points of $\mathcal{O}(NK(\log N + F))$.

## G.1 Validation of theoretical results

### G.1.1 Empirical validation of error bounds for nearest neighbor approaches to imputation

Table 3 presents numerical results corresponding to Figure 1 for the validation of Theorem 2.3 for F3I (Algorithm 2) and other NN algorithms for $p^{\mathrm{miss}} = 25\%$. All values are rounded to the closest second decimal place in all tables of this section. We also show here that the results obtained for $p^{\mathrm{miss}} = 25\%$ and F3I in the main text still hold for other missingness frequencies ($p^{\mathrm{miss}} \in \{10\%, 50\%, 75\%\}$) and other nearest-neighbor algorithms, as illustrated by Figures 7-9 for the F3I algorithm in Algorithm 2 (relationship between $\sigma$ and the MSE loss is shown in Figure 10); Figures 11-13 for the distance-dependent kNN algorithm (Figure 14); and Figures 15-17 for the uniformly-weighted kNN algorithm (Figure 18). Corresponding tables are Tables 4-6. For the distance-dependent kNN algorithm, since weights $(\boldsymbol{\alpha}^i)_{i \leq N}$ are data-point-dependent, we define $A \triangleq \max_{i \leq N} \sum_{k \leq K} \alpha_k^i$.

Table 3: Empirical validation of Theorem 2.3 where $p^{\mathrm{miss}} = 25\%$.

| Algorithm | Missingness type | $\sigma$ | $NF \times \mathcal{L}^{\mathrm{MSE}}(\widehat{X}, X^\star)$ | $N \times C_\delta^{\mathrm{miss}}$ |
|---|---|---|---|---|
| F3I | MCAR (Assumption B.1) | 0.01 | 0.15 ±0.01 | 2,378.62 |
| | | 0.10 | 14.94 ±0.61 | 4,354.42 |
| | | 0.15 | 33.6 ±1.4 | 6,001.27 |
| | | 0.20 | 59.77 ±2.45 | 7,862.58 |
| | | 0.25 | 93.43 ±3.88 | 9,916.63 |
| | | 0.50 | 373.77 ±15.52 | 22,908.41 |
| | MAR (Assumption B.2) | 0.01 | 0.1 ±0.01 | 2,378.62 |
| | | 0.10 | 10.27 ±0.61 | 4,354.42 |
| | | 0.15 | 23.09 ±1.34 | 6,001.27 |
| | | 0.20 | 41.02 ±2.33 | 7,862.58 |
| | | 0.25 | 64.06 ±3.7 | 9,916.63 |
| | | 0.50 | 255.97 ±15.09 | 22,908.41 |
| | MNAR (Assumption B.3) | 0.01 | 0.06 ±0.0 | 2,378.62 |
| | | 0.10 | 5.8 ±0.37 | 4,354.42 |
| | | 0.15 | 13.06 ±0.81 | 6,001.27 |
| | | 0.20 | 23.21 ±1.45 | 7,862.58 |
| | | 0.25 | 36.26 ±2.27 | 9,916.63 |
| | | 0.50 | 144.87 ±9.06 | 22,908.41 |
| Distance kNN | MCAR (Assumption B.1) | 0.01 | 0.15 ±0.01 | 2,356.19 |
| | | 0.10 | 15.0 ±0.65 | 3,077.8 |
| | | 0.15 | 33.75 ±1.47 | 3,765.53 |
| | | 0.20 | 60.01 ±2.61 | 4,554.96 |
| | | 0.25 | 93.76 ±4.09 | 5,419.85 |
| | | 0.50 | 375.04 ±16.34 | 10,624.98 |
| | MAR (Assumption B.2) | 0.01 | 0.11 ±0.01 | 2,356.19 |

| | | | | |
|---|---|---|---|---|
| | | 0.10 | 10.59 ±0.61 | 3,077.8 |
| | | 0.15 | 23.8 ±1.38 | 3,765.53 |
| | | 0.20 | 42.27 ±2.44 | 4,554.96 |
| | | 0.25 | 66.03 ±3.76 | 5,419.85 |
| | | 0.50 | 264.25 ±15.57 | 10,624.98 |
| | MNAR (Assumption B.3) | 0.01 | 0.06 ±0.0 | 2,356.19 |
| | | 0.10 | 6.01 ±0.37 | 3,077.8 |
| | | 0.15 | 13.53 ±0.84 | 3,765.53 |
| | | 0.20 | 24.05 ±1.5 | 4,554.96 |
| | | 0.25 | 37.58 ±2.34 | 5,419.85 |
| | | 0.50 | 150.31 ±9.35 | 10,624.98 |
| Uniform kNN | MCAR (Assumption B.1) | 0.01 | 0.15 ±0.01 | 2,352.6 |
| | | 0.10 | 15.0 ±0.65 | 2,816.02 |
| | | 0.15 | 33.75 ±1.47 | 3,286.57 |
| | | 0.20 | 59.99 ±2.61 | 3,839.3 |
| | | 0.25 | 93.74 ±4.08 | 4,450.16 |
| | | 0.50 | 374.96 ±16.33 | 8,109.04 |
| | MAR (Assumption B.2) | 0.01 | 0.11 ±0.01 | 2,352.6 |
| | | 0.10 | 10.58 ±0.61 | 2,816.02 |
| | | 0.15 | 23.79 ±1.37 | 3,286.57 |
| | | 0.20 | 42.25 ±2.44 | 3,839.3 |
| | | 0.25 | 66.01 ±3.75 | 4,450.16 |
| | | 0.50 | 264.16 ±15.55 | 8,109.04 |
| | MNAR (Assumption B.3) | 0.01 | 0.06 ±0.0 | 2,352.6 |
| | | 0.10 | 6.01 ±0.37 | 2,816.02 |
| | | 0.15 | 13.52 ±0.84 | 3,286.57 |
| | | 0.20 | 24.04 ±1.5 | 3,839.3 |
| | | 0.25 | 37.56 ±2.34 | 4,450.16 |
| | | 0.50 | 150.22 ±9.35 | 8,109.04 |

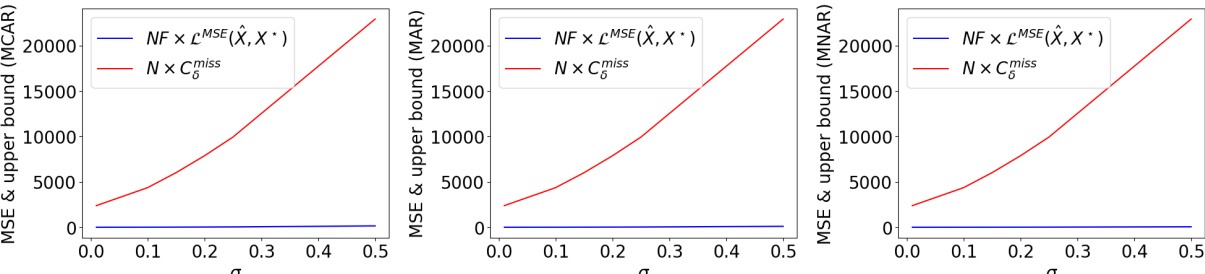

Figure 7: Empirical validation of Theorem 2.3 by comparing the value of the upper bound $N \times C^{\mathrm{miss}}$ and $NF \times \mathcal{L}^{\mathrm{MSE}}(X^t, X^\star)$ where $t$ is the final round for F3I and $p^{\mathrm{miss}} = 10\%$. Left: MCAR setting. Center: MAR setting. Right: MNAR setting.

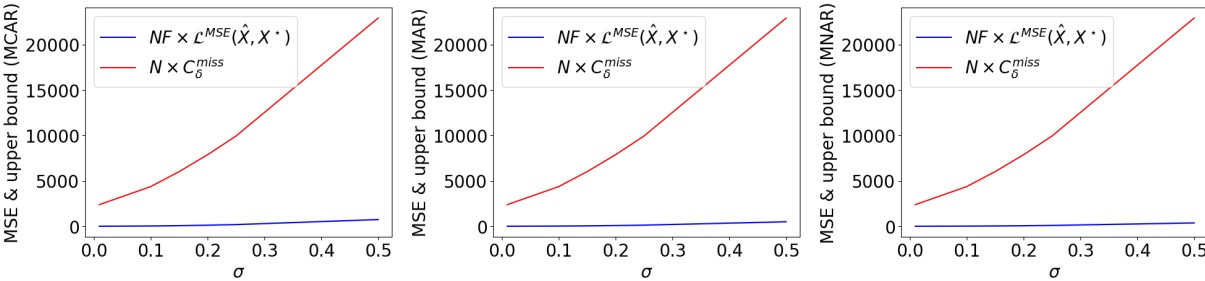

Figure 8: Empirical validation of Theorem 2.3 by comparing the value of the upper bound $N \times C^{\mathrm{miss}}$ and $NF \times \mathcal{L}^{\mathrm{MSE}}(X^t, X^\star)$ where $t$ is the final round for F3I and $p^{\mathrm{miss}} = 50\%$. Left: MCAR setting. Center: MAR setting. Right: MNAR setting.

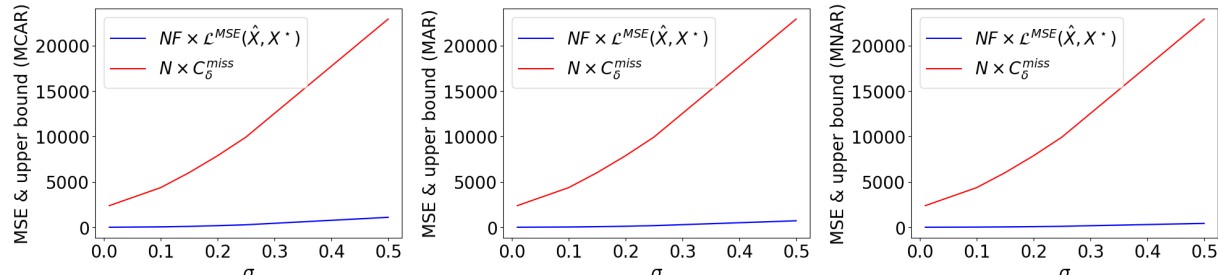

Figure 9: Empirical validation of Theorem 2.3 by comparing the value of the upper bound $N \times C^{\mathrm{miss}}$ and $NF \times \mathcal{L}^{\mathrm{MSE}}(X^t, X^\star)$ where $t$ is the final round for F3I and $p^{\mathrm{miss}} = 75\%$. Left: MCAR setting. Center: MAR setting. Right: MNAR setting.

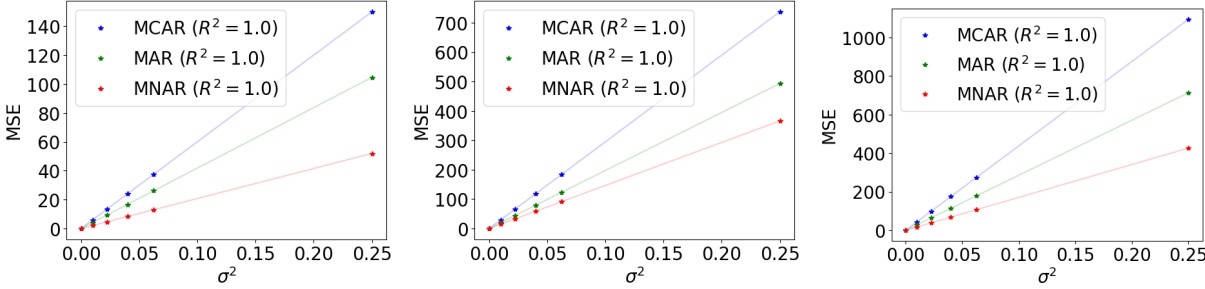

Figure 10: $NF \times \mathcal{L}^{\mathrm{MSE}}(X^t, X^\star)$ is linear in $\sigma^2$ regardless of the missingness mechanism and missingness frequency $p^{\mathrm{miss}}$ for F3I (left: 10%, center: 50%, right: 75%).

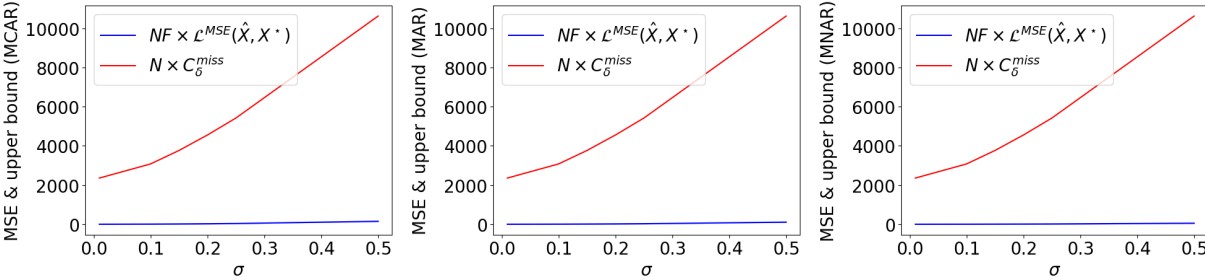

Figure 11: Empirical validation of Theorem 2.3 by comparing the value of the upper bound $N \times C^{\mathrm{miss}}$ and $NF \times \mathcal{L}^{\mathrm{MSE}}(X^t, X^\star)$ where $t$ is the final round for the distance-dependent kNN algorithm and $p^{\mathrm{miss}} = 10\%$. Left: MCAR setting. Center: MAR setting. Right: MNAR setting.

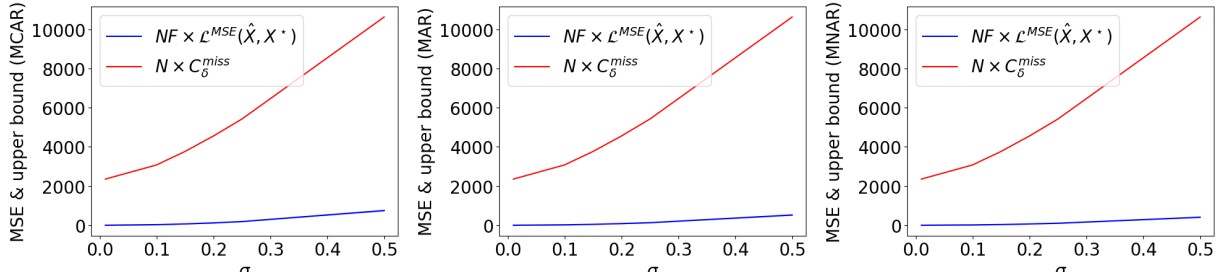

Figure 12: Empirical validation of Theorem 2.3 by comparing the value of the upper bound $N \times C^{\mathrm{miss}}$ and $NF \times \mathcal{L}^{\mathrm{MSE}}(X^t, X^\star)$ where $t$ is the final round for the distance-dependent kNN algorithm and $p^{\mathrm{miss}} = 50\%$. Left: MCAR setting. Center: MAR setting. Right: MNAR setting.

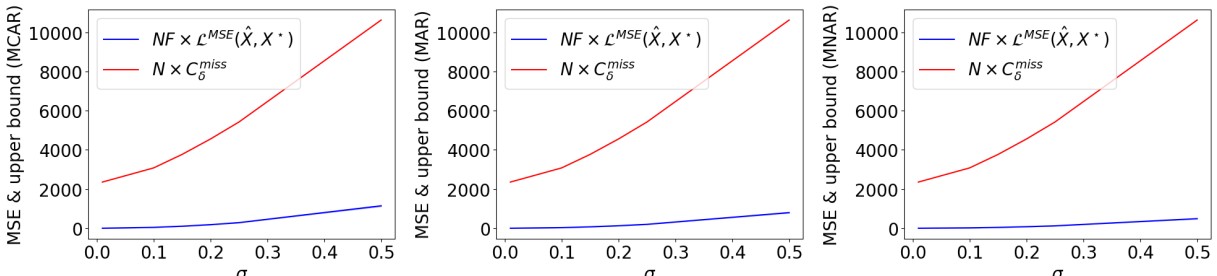

Figure 13: Empirical validation of Theorem 2.3 by comparing the value of the upper bound $N \times C^{\mathrm{miss}}$ and $NF \times \mathcal{L}^{\mathrm{MSE}}(X^t, X^\star)$ where $t$ is the final round for the distance-dependent kNN algorithm and $p^{\mathrm{miss}} = 75\%$. Left: MCAR setting. Center: MAR setting. Right: MNAR setting.

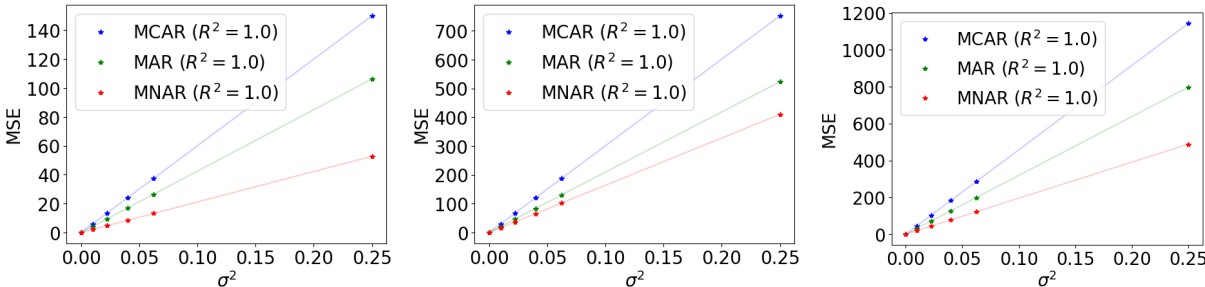

Figure 14: $NF \times \mathcal{L}^{\mathrm{MSE}}(X^t, X^\star)$ is linear in $\sigma^2$ regardless of the missingness mechanism and missingness frequency $p^{\mathrm{miss}}$ for the distance-dependent kNN algorithm (left: 10%, center: 50%, right: 75%).

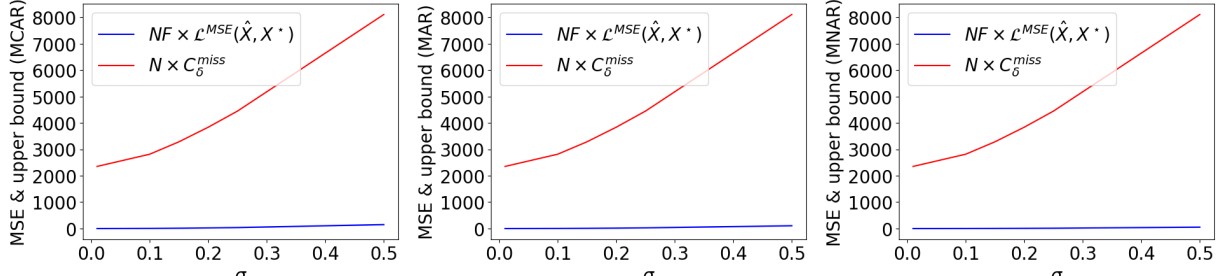

Figure 15: Empirical validation of Theorem 2.3 by comparing the value of the upper bound $N \times C^{\mathrm{miss}}$ and $NF \times \mathcal{L}^{\mathrm{MSE}}(X^t, X^\star)$ where $t$ is the final round for the uniform kNN algorithm and $p^{\mathrm{miss}} = 10\%$. Left: MCAR setting. Center: MAR setting. Right: MNAR setting.

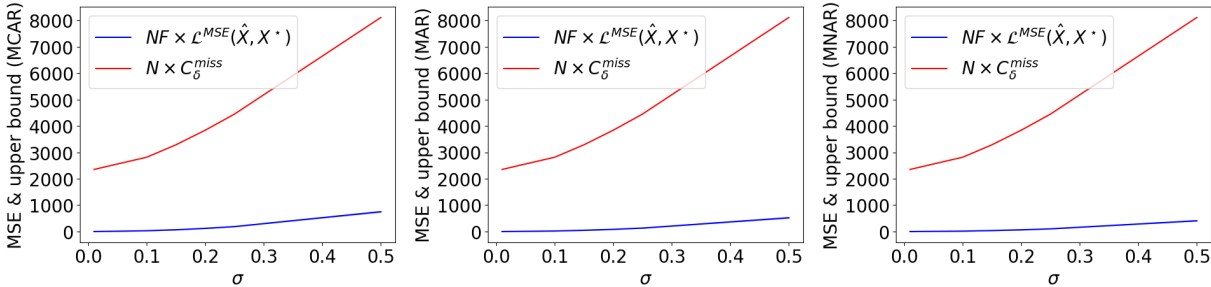

Figure 16: Empirical validation of Theorem 2.3 by comparing the value of the upper bound $N \times C^{\mathrm{miss}}$ and $NF \times \mathcal{L}^{\mathrm{MSE}}(X^t, X^\star)$ where $t$ is the final round for the uniform kNN algorithm and $p^{\mathrm{miss}} = 50\%$. Left: MCAR setting. Center: MAR setting. Right: MNAR setting.

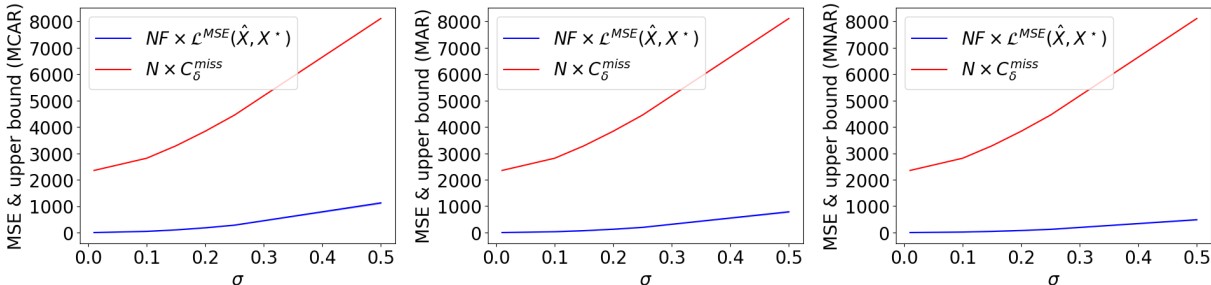

Figure 17: Empirical validation of Theorem 2.3 by comparing the value of the upper bound $N \times C^{\mathrm{miss}}$ and $NF \times \mathcal{L}^{\mathrm{MSE}}(X^t, X^\star)$ where $t$ is the final round for the uniform kNN algorithm and $p^{\mathrm{miss}} = 75\%$. Left: MCAR setting. Center: MAR setting. Right: MNAR setting.

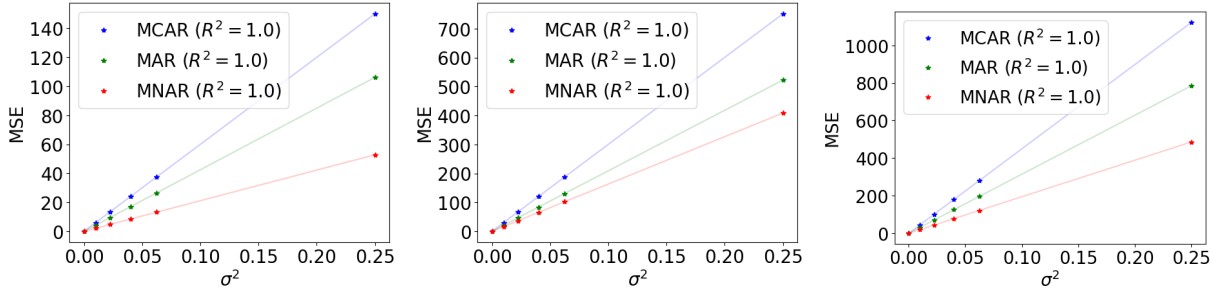

Figure 18: $NF \times \mathcal{L}^{\mathrm{MSE}}(X^t, X^\star)$ is linear in $\sigma^2$ regardless of the missingness mechanism and missingness frequency $p^{\mathrm{miss}}$ for the uniform kNN algorithm (left: 10%, center: 50%, right: 75%).

Table 4: Empirical validation of Theorem 2.3 on F3I (Algorithm 2)

| Missingness frequency | Missingness type | $\sigma$ | $NF \times \mathcal{L}^{\mathrm{MSE}}(\widehat{X}, X^\star)$ | $N \times C_\delta^{\mathrm{miss}}$ |
|---|---|---|---|---|
| 10% | MCAR (Assumption B.1) | 0.01 | 0.06 ±0.0 | 2378.62 |
| | | 0.10 | 6.0 ±0.44 | 4,354.42 |
| | | 0.15 | 13.51 ±1.0 | 6,001.27 |
| | | 0.20 | 24.01 ±1.77 | 7,862.58 |
| | | 0.25 | 37.53 ±2.77 | 9,916.63 |
| | | 0.50 | 149.97 ±11.13 | 22,908.41 |
| | MAR (Assumption B.2) | 0.01 | 0.04 ±0.0 | 2378.62 |
| | | 0.10 | 4.17 ±0.34 | 4,354.42 |
| | | 0.15 | 9.39 ±0.78 | 6,001.27 |
| | | 0.20 | 16.72 ±1.37 | 7,862.58 |
| | | 0.25 | 26.14 ±2.14 | 9,916.63 |
| | | 0.50 | 104.58 ±8.71 | 22,908.41 |
| | MNAR (Assumption B.3) | 0.01 | 0.02 ±0.0 | 2,378.62 |
| | | 0.10 | 0.02 ±0.0 | 2,378.62 |
| | | 0.15 | 4.67 ±0.58 | 6,001.27 |
| | | 0.20 | 8.3 ±1.05 | 7,862.58 |
| | | 0.25 | 12.97 ±1.63 | 9,916.63 |
| | | 0.50 | 51.95 ±6.43 | 22,908.41 |
| 50% | MCAR (Assumption B.1) | 0.01 | 0.29 ±0.01 | 2,378.62 |
| | | 0.10 | 29.49 ±0.85 | 4,354.42 |
| | | 0.15 | 66.36 ±1.87 | 6,001.27 |

| Missingness frequency | Missingness type | $\sigma$ | $NF \times \mathcal{L}^{\mathrm{MSE}}(\widehat{X}, X^\star)$ | $N \times C_\delta^{\mathrm{miss}}$ |
|---|---|---|---|---|
| | | 0.20 | 117.92 ±3.29 | 7,862.58 |
| | | 0.25 | 184.29 ±5.13 | 9,916.63 |
| | | 0.50 | 737.13 ±20.77 | 22,908.41 |
| | MAR (Assumption B.2) | 0.01 | 0.2 ±0.01 | 2,378.62 |
| | | 0.10 | 19.77 ±0.63 | 4,354.42 |
| | | 0.15 | 44.45 ±1.49 | 6,001.27 |
| | | 0.20 | 78.97 ±2.62 | 7,862.58 |
| | | 0.25 | 123.4 ±4.13 | 9,916.63 |
| | | 0.50 | 493.58 ±16.36 | 22,908.41 |
| | MNAR (Assumption B.3) | 0.01 | 0.15 ±0.01 | 2,378.62 |
| | | 0.10 | 14.67 ±0.54 | 4,354.42 |
| | | 0.15 | 32.96 ±1.22 | 6,001.27 |
| | | 0.20 | 58.6 ±2.19 | 7,862.58 |
| | | 0.25 | 91.55 ±3.38 | 9,916.63 |
| | | 0.50 | 366.28 ±13.71 | 22,908.41 |
| 75% | MCAR (Assumption B.1) | 0.01 | 0.44 ±0.01 | 2,378.62 |
| | | 0.10 | 43.78 ±1.0 | 4,354.42 |
| | | 0.15 | 98.49 ±2.23 | 6,001.27 |
| | | 0.20 | 175.1 ±3.9 | 7,862.58 |
| | | 0.25 | 273.57 ±6.16 | 9,916.63 |
| | | 0.50 | 1094.5 ±24.44 | 22,908.41 |
| | MAR (Assumption B.2) | 0.01 | 0.31 ±0.01 | 2,378.62 |
| | | 0.10 | 28.57 ±0.65 | 4,354.42 |
| | | 0.15 | 64.28 ±1.52 | 6,001.27 |
| | | 0.20 | 114.23 ±2.68 | 7,862.58 |
| | | 0.25 | 178.45 ±4.21 | 9,916.63 |
| | | 0.50 | 713.69 ±16.79 | 22,908.41 |
| | MNAR (Assumption B.3) | 0.01 | 0.17 ±0.01 | 2,378.62 |
| | | 0.10 | 17.04 ±0.58 | 4,354.42 |
| | | 0.15 | 38.33 ±1.32 | 6,001.27 |
| | | 0.20 | 68.2 ±2.42 | 7,862.58 |
| | | 0.25 | 106.58 ±3.78 | 9,916.63 |
| | | 0.50 | 426.44 ±15.28 | 22,908.41 |

Table 5: Empirical validation of Theorem 2.3 on the distance-dependent kNN algorithm

| Missingness frequency | Missingness type | $\sigma$ | $NF \times \mathcal{L}^{\mathrm{MSE}}(\widehat{X}, X^\star)$ | $N \times C_\delta^{\mathrm{miss}}$ |
|---|---|---|---|---|
| 10% | MCAR (Assumption B.1) | 0.01 | 0.06 ±0.0 | 2,356.19 |
| | | 0.10 | 6.0 ±0.43 | 3,077.8 |
| | | 0.15 | 13.5 ±0.96 | 3,765.53 |
| | | 0.20 | 23.99 ±1.7 | 4,554.96 |
| | | 0.25 | 37.49 ±2.66 | 5,419.85 |
| | | 0.50 | 149.96 ±10.63 | 10,624.98 |
| | MAR (Assumption B.2) | 0.01 | 0.04 ±0.0 | 2,356.19 |
| | | 0.10 | 4.24 ±0.36 | 3,077.8 |
| | | 0.15 | 9.54 ±0.81 | 3,765.53 |
| | | 0.20 | 16.97 ±1.43 | 4,554.96 |
| | | 0.25 | 26.51 ±2.24 | 5,419.85 |
| | | 0.50 | 106.21 ±9.09 | 10,624.98 |
| | MNAR (Assumption B.3) | 0.01 | 0.02 ±0.0 | 2,356.19 |
| | | 0.10 | 2.11 ±0.26 | 3,077.8 |

| | | σ | $NF \times \mathcal{L}^{\mathrm{MSE}}(\widehat{X}, X^\star)$ | $N \times C_\delta^{\mathrm{miss}}$ |
|---|---|---|---|---|
| | | 0.15 | 4.75 ±0.58 | 3,765.53 |
| | | 0.20 | 8.45 ±1.03 | 4,554.96 |
| | | 0.25 | 13.2 ±1.61 | 5,419.85 |
| | | 0.50 | 52.79 ±6.46 | 10,624.98 |
| 50% | MCAR (Assumption B.1) | 0.01 | 0.3 ±0.01 | 2,356.19 |
| | | 0.10 | 30.07 ±0.85 | 3,077.8 |
| | | 0.15 | 67.65 ±1.92 | 3,765.53 |
| | | 0.20 | 120.27 ±3.41 | 4,554.96 |
| | | 0.25 | 187.92 ±5.33 | 5,419.85 |
| | | 0.50 | 751.68 ±21.33 | 10,624.98 |
| | MAR (Assumption B.2) | 0.01 | 0.21 ±0.01 | 2,356.19 |
| | | 0.10 | 20.92 ±0.73 | 3,077.8 |
| | | 0.15 | 47.11 ±1.71 | 3,765.53 |
| | | 0.20 | 83.81 ±2.95 | 4,554.96 |
| | | 0.25 | 130.86 ±4.67 | 5,419.85 |
| | | 0.50 | 522.97 ±19.0 | 10,624.98 |
| | MNAR (Assumption B.3) | 0.01 | 0.16 ±0.01 | 2,356.19 |
| | | 0.10 | 16.39 ±0.56 | 3,077.8 |
| | | 0.15 | 36.88 ±1.25 | 3,765.53 |
| | | 0.20 | 65.56 ±2.22 | 4,554.96 |
| | | 0.25 | 102.44 ±3.47 | 5,419.85 |
| | | 0.50 | 409.75 ±13.88 | 10,624.98 |
| 75% | MCAR (Assumption B.1) | 0.01 | 0.46 ±0.01 | 2,356.19 |
| | | 0.10 | 45.75 ±1.16 | 3,077.8 |
| | | 0.15 | 102.94 ±2.61 | 3,765.53 |
| | | 0.20 | 183.01 ±4.64 | 4,554.96 |
| | | 0.25 | 285.95 ±7.24 | 5,419.85 |
| | | 0.50 | 1,143.8 ±28.98 | 10,624.98 |
| | MAR (Assumption B.2) | 0.01 | 0.32 ±0.01 | 2,356.19 |
| | | 0.10 | 31.89 ±0.79 | 3,077.8 |
| | | 0.15 | 71.74 ±1.83 | 3,765.53 |
| | | 0.20 | 127.52 ±3.22 | 4,554.96 |
| | | 0.25 | 199.31 ±5.02 | 5,419.85 |
| | | 0.50 | 797.45 ±20.26 | 10,624.98 |
| | MNAR (Assumption B.3) | 0.01 | 0.2 ±0.01 | 2,356.19 |
| | | 0.10 | 19.54 ±0.53 | 3,077.8 |
| | | 0.15 | 43.97 ±1.19 | 3,765.53 |
| | | 0.20 | 78.17 ±2.12 | 4,554.96 |
| | | 0.25 | 122.15 ±3.31 | 5,419.85 |
| | | 0.50 | 488.58 ±13.23 | 10,624.98 |

Table 6: Empirical validation of Theorem 2.3 on the uniform kNN algorithm.

| Missingness frequency | Missingness type | σ | $NF \times \mathcal{L}^{\mathrm{MSE}}(\widehat{X}, X^\star)$ | $N \times C_\delta^{\mathrm{miss}}$ |
|---|---|---|---|---|
| 10% | MCAR (Assumption B.1) | 0.01 | 0.06 ±0.0 | 2,352.6 |
| | | 0.10 | 6.0 ±0.43 | 2,816.02 |
| | | 0.15 | 13.5 ±0.96 | 3,286.57 |
| | | 0.20 | 24.0 ±1.71 | 3,839.3 |
| | | 0.25 | 37.5 ±2.67 | 4,450.16 |
| | | 0.50 | 149.99 ±10.66 | 8109.04 |

| | | | | |
|---|---|---|---|---|
| | MAR (Assumption B.2) | 0.01 | 0.04 ±0.0 | 2,352.6 |
| | | 0.10 | 4.24 ±0.36 | 2,816.02 |
| | | 0.15 | 9.54 ±0.81 | 3,286.57 |
| | | 0.20 | 16.97 ±1.43 | 3,839.3 |
| | | 0.25 | 26.51 ±2.25 | 4,450.16 |
| | | 0.50 | 106.21 ±9.12 | 8,109.04 |
| | MNAR (Assumption B.3) | 0.01 | 0.02 ±0.0 | 2,352.6 |
| | | 0.10 | 2.11 ±0.26 | 2,816.02 |
| | | 0.15 | 4.75 ±0.58 | 3,286.57 |
| | | 0.20 | 8.45 ±1.03 | 3,839.3 |
| | | 0.25 | 13.2 ±1.61 | 4,450.16 |
| | | 0.50 | 52.79 ±6.45 | 8,109.04 |
| 50% | MCAR (Assumption B.1) | 0.01 | 0.3 ±0.01 | 2,352.6 |
| | | 0.10 | 30.04 ±0.85 | 2,816.02 |
| | | 0.15 | 67.59 ±1.92 | 3,286.57 |
| | | 0.20 | 120.16 ±3.4 | 3,839.3 |
| | | 0.25 | 187.75 ±5.32 | 4,450.16 |
| | | 0.50 | 751.0 ±21.28 | 8,109.04 |
| | MAR (Assumption B.2) | 0.01 | 0.21 ±0.01 | 2,352.6 |
| | | 0.10 | 20.91 ±0.73 | 2,816.02 |
| | | 0.15 | 47.07 ±1.7 | 3,286.57 |
| | | 0.20 | 83.74 ±2.93 | 3,839.3 |
| | | 0.25 | 130.74 ±4.63 | 4,450.16 |
| | | 0.50 | 522.46 ±18.91 | 8,109.04 |
| | MNAR (Assumption B.3) | 0.01 | 0.16 ±0.01 | 2,352.6 |
| | | 0.10 | 16.34 ±0.56 | 2,816.02 |
| | | 0.15 | 36.77 ±1.26 | 3,286.57 |
| | | 0.20 | 65.37 ±2.23 | 3,839.3 |
| | | 0.25 | 102.15 ±3.49 | 4,450.16 |
| | | 0.50 | 408.59 ±13.95 | 8,109.04 |
| 75% | MCAR (Assumption B.1) | 0.01 | 0.45 ±0.01 | 2,352.6 |
| | | 0.10 | 44.91 ±1.04 | 2,816.02 |
| | | 0.15 | 101.05 ±2.34 | 3,286.57 |
| | | 0.20 | 179.65 ±4.15 | 3,839.3 |
| | | 0.25 | 280.7 ±6.49 | 4,450.16 |
| | | 0.50 | 1122.81 ±25.95 | 8,109.04 |
| | MAR (Assumption B.2) | 0.01 | 0.31 ±0.01 | 2352.6 |
| | | 0.10 | 31.41 ±0.75 | 2816.02 |
| | | 0.15 | 70.69 ±1.75 | 3286.57 |
| | | 0.20 | 125.61 ±3.03 | 3839.3 |
| | | 0.25 | 196.33 ±4.75 | 4450.16 |
| | | 0.50 | 785.34 ±19.23 | 8109.04 |
| | MNAR (Assumption B.3) | 0.01 | 0.19 ±0.01 | 2,352.6 |
| | | 0.10 | 19.45 ±0.53 | 2,816.02 |
| | | 0.15 | 43.75 ±1.18 | 3,286.57 |
| | | 0.20 | 77.78 ±2.1 | 3,839.3 |
| | | 0.25 | 121.54 ±3.29 | 4,450.16 |
| | | 0.50 | 486.14 ±13.16 | 8,109.04 |

### G.1.2 Empirical validation of regret analysis on F3I (Algorithm 2)

We show that Theorem D.1 is experimentally validated, by looking at the upper bound for the expected cumulative regret stated in Theorem D.1. Using a missingness frequency of 25% again, we run 100 times F3I on synthetic data sets for all three missingness mechanisms and track the values of $\max_{\boldsymbol{\alpha} \in \triangle_K} \sum_{s=1}^{t} G_*(\boldsymbol{\alpha}, X^{s-1}) - G_*(\boldsymbol{\alpha}^s, X^{s-1})$ and its upper bound $C^{\mathrm{AH}}\sqrt{t} + H^{\mathrm{miss}}h^{-1}t$ across iterations, where $t$ is the final step of F3I (that can change across iterations). We compute the value of $\max_{\boldsymbol{\alpha} \in \triangle_K} \sum_{s=1}^{t} G_*(\boldsymbol{\alpha}, X^{s-1})$ by solving the related convex problem with function `minimize` in Python package `scipy.optimize` (Virtanen et al., 2020) after running F3I

$$\min_{\boldsymbol{\alpha} \in \mathbb{R}^K} - \sum_{s=1}^{t} G_{\star}(\boldsymbol{\alpha}, X^{s-1}) \quad \text{such that} \quad \forall k \leq K, \ \alpha_k \geq 0 \quad \text{and} \quad \sum_{k \leq K} \alpha_k = 1 \ ,$$

where $G_{\star}$ is computed with respect to the true complete points $\{(\boldsymbol{x}^{\star})_1, \ldots, (\boldsymbol{x}^{\star})_N\}$ and $(\boldsymbol{x}^s)_i = \mathtt{Impute}((\boldsymbol{x}^{s-1})_i; \boldsymbol{\alpha}^s)$ if $s \geq 1$ and $\widehat{X}$ is the naively imputed matrix. Figure 19 and Table 7 show that the upper bound is always valid across those experiments. Some random data sets among the 100 might be harder than the others, incurring larger regret. However, Figure 19 shows that the upper bound adapts to these instances. The large gap between the empirical and theoretical peaks in hardness might be due, as for Theorem 2.3, to the conservative estimates given by Bernstein's inequality.

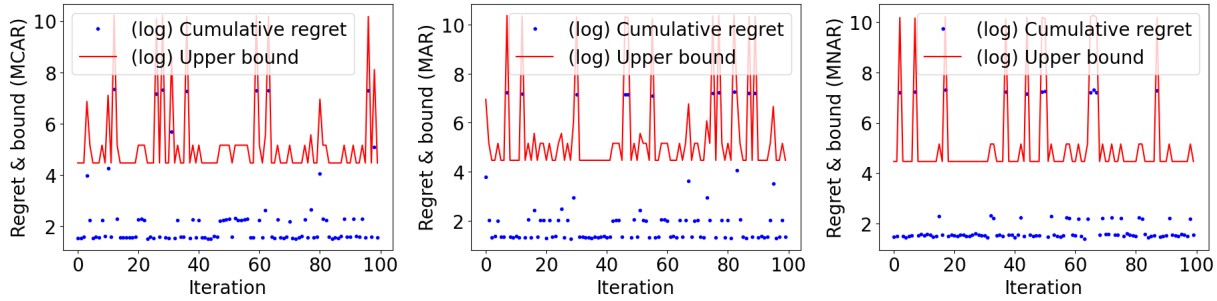

Figure 19: Cumulative regret for F3I and upper bound from Theorem D.1 across 100 iterations in log-values. Left: MCAR setting. Center: MAR setting. Right: MNAR setting.

Table 7: Empirical validation of Theorem D.1 by comparing the value of the upper bound and the average and standard deviation value of the cumulative regret $\max_{\boldsymbol{\alpha} \in \triangle_K} \sum_{s=1}^{t} G_*(\boldsymbol{\alpha}, X^{s-1}) - G_*(\boldsymbol{\alpha}^s, X^{s-1})$ across iterations where $t$ is the final round for F3I (the maximum number of rounds is set to 500). All values are rounded to the closest second decimal place, except for the time round $t$, which is rounded to the closest integer. Theorem D.1 states that $\max_{\boldsymbol{\alpha} \in \triangle_K} \sum_{s=1}^{t} G_*(\boldsymbol{\alpha}, X^{s-1}) - G_*(\boldsymbol{\alpha}^s, X^{s-1}) \leq C^{\mathrm{AH}}\sqrt{t} + H^{\mathrm{miss}}h^{-1}t$ with probability $1 - 1/2,500 \approx 0.9996$.

| Missingness type | | # Iterations | Cumulative regret on $G$ | Upper bound from Theorem D.1 |
|---|---|---|---|---|
| MCAR | (Assumption B.1) | 24 ±78 | 113.76 ±371.39 | 2,067.60 ±6,734.50 |
| MAR | (Assumption B.2) | 40 ±111 | 153.06 ±417.40 | 3,542.96 ±9,659.89 |
| MNAR | (Assumption B.3) | 35 ±96 | 158.61 ±438.45 | 3,015.86 ±8,336.24 |

### G.1.3 Empirical validation of regret analysis on PCGradF3I (Algorithm 2 with a downstream task loss)

Continuing from Section 4.1, we validate the upper bound on the cumulative regret in Theorem 3.1 for all three missingness mechanisms we studied, similar to what was done for Theorem D.1. As in our proofs (see Appendix E), we consider the log-loss $\ell$ with the sigmoid classifier $C_{\boldsymbol{\omega}}$, satisfying all our assumptions, and $\beta = 0.5$. We consider a missingness frequency of 50%. To obtain the value of arg $\max_{\boldsymbol{\alpha} \in \triangle_K} \sum_{s=1}^{t} \overline{G}(\boldsymbol{\alpha}, X^{s-1})$,

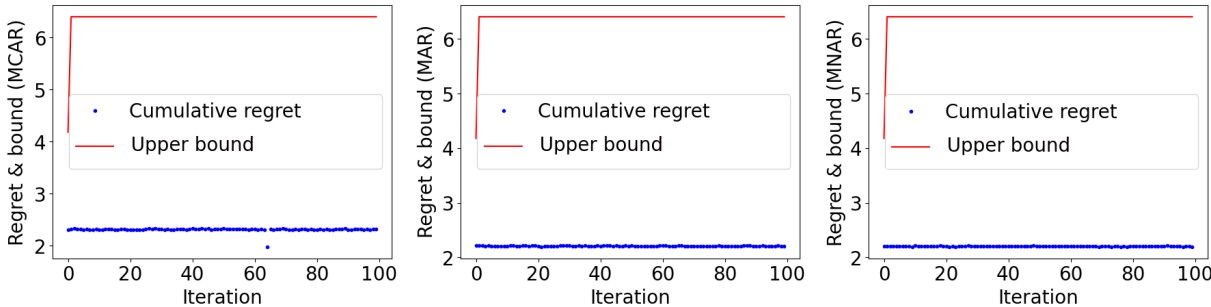

Figure 20: Cumulative regret for F3I and upper bound from Theorem 3.1 across 100 iterations for $\beta = 0.5$. The blue points are always below the red lines. Left: MCAR setting. Center: MAR setting. Right: MNAR setting.

Table 8: Empirical validation of Theorem 3.1 by comparing the value of the upper bound and the average and standard deviation value of the cumulative regret $\max_{\boldsymbol{\alpha} \in \triangle_K} \sum_{s=1}^{t} \overline{G}(\boldsymbol{\alpha}, X^{s-1}) - \overline{G}(\boldsymbol{\alpha}^s, X^{s-1})$ across iterations where $t$ is the final round for F3I in the last epoch. All values are rounded to the closest second decimal place. The time round is fixed to $T = 3$ in PCGrad-F3I and $\beta = 0.5$. Theorem D.1 states that $\max_{\boldsymbol{\alpha} \in \triangle_K} \sum_{s=1}^{t} \overline{G}(\boldsymbol{\alpha}, X^{s-1}) - \overline{G}(\boldsymbol{\alpha}^s, X^{s-1}) \leq C_{(G,\ell)}^{\mathrm{AH}} \sqrt{t} + (1 - \beta) H^{\mathrm{miss}} h^{-1} t$ with probability $1 - 1/2,500 \approx 0.9996$.

| Missingness type | | $t$ | Cumulative regret on $\overline{G}$ | $C_{(G,\ell)}^{\mathrm{AH}} \sqrt{t} + (1 - \beta) H^{\mathrm{miss}} h^{-1} t$ |
|---|---|---|---|---|
| MCAR | (Assumption B.1) | 3 | 2.31 ±0.03 | 6.38 ±0.22 |
| MAR | (Assumption B.2) | 3 | 2.21 ±0.01 | 6.37 ±0.22 |
| MNAR | (Assumption B.3) | 3 | 2.21 ±0.00 | 6.38 ±0.22 |

where $\overline{G}$ is defined as

$$\overline{G} : \boldsymbol{\alpha} \in \triangle_K, X \in \mathbb{R}^{N \times F} \mapsto (1 - \beta) G_\star(\boldsymbol{\alpha}, X) - \frac{\beta}{N} \sum_{i \leq N} \ell(\boldsymbol{x}_i(\boldsymbol{\alpha})) \, ,$$

we solve the following optimization problem by solving the related convex problem with function `minimize` in Python package `scipy.optimize` (Virtanen et al., 2020), considering the $(X^{s-1})_{s \leq t}$ and parameter of the sigmoid classifier $C_\omega$ in the last epoch in PCGrad-F3I

$$\min_{\boldsymbol{\alpha} \in \mathbb{R}^K} - \sum_{s=1}^{t} \overline{G}(\boldsymbol{\alpha}, X^{s-1}) \quad \text{such that} \quad \forall k \leq K, \ \alpha_k \geq 0 \quad \text{and} \quad \sum_{k \leq K} \alpha_k = 1 \, .$$

The gradient and the Hessian matrix of the objective function $\overline{G}$ with respect to $\boldsymbol{\alpha}$ are obtained by combining the results from Lemmas C.3, Lemma C.4 and Section E. Figure 20 and Table 8 indeed show that the upper bound reliably holds on the cumulative regret for $\overline{G}$.

### G.2 Evaluation of empirical performance

We add here the numerical details for the results presented in Figures 4- 6.

#### G.2.1 Empirical performance of nearest neighbor-based imputers compared to other baselines

Tables 9-10 provide full numerical results for Figure 4, along with additional metrics Mean Absolute Error (MAE) and Wasserstein distance (WD) (Givens & Shortt, 1984). Additionally, we also benchmark the nearest-neighbor approaches against the other baselines across all three missingness scenarios (MCAR, MAR and MNAR) for nine datasets at a missingness rate of $p^{\mathrm{miss}} = 30\%$ in Table 11.

Table 9: Average and standard deviation values of imputation metrics and runtime across 10 seeds. kNN-d (respectively, kNN-u) is the distance-dependent (resp., uniformly) weighted kNN algorithm.

| Data set | Algorithm | RMSE ↓ | MAE ↓ | WD ↓ | Runtime ↓ |
|---|---|---|---|---|---|
| BreastCancer | **F3I** | 0.08 ±0.00 | 0.05 ±0.00 | 0.06 ±0.01 | 0.73 ±0.10 |
| | **kNN-d** | 0.08 ±0.01 | 0.05 ±0.00 | 0.07 ±0.01 | **0.04 ±0.01** |
| | **kNN-u** | 0.08 ±0.01 | 0.05 ±0.00 | 0.07 ±0.01 | **0.04 ±0.01** |
| | GAIN | 0.08 ±0.00 | 0.05 ±0.00 | 0.08 ±0.01 | 26.34 ±7.74 |
| | GRAPE | 0.24 ±0.01 | 0.18 ±0.01 | 0.21 ±0.02 | 4,944 ±618 |
| | HyperImpute | **0.05 ±0.00** | **0.03 ±0.00** | **0.04 ±0.00** | 12.05 ±2.32 |
| | MIRACLE | 3.99 ±0.21 | 3.93 ±0.21 | 9.44 ±0.50 | 10.52 ±0.43 |
| | NewImp | 0.30 ±0.04 | 0.23 ±0.03 | 0.55 ±0.07 | 1,594 ±379 |
| | Remasker | 0.10 ±0.01 | 0.07 ±0.00 | 0.13 ±0.01 | 1,399 ±303 |
| | TDM | 0.28±0.03 | 0.22 ±0.02 | 0.24 ±0.05 | 347 ±55 |
| Diabetes | **F3I** | 0.26 ±0.01 | 0.18 ±0.00 | 0.18 ±0.01 | 0.16 ±0.02 |
| | **kNN-d** | 0.24 ±0.01 | 0.16 ±0.00 | 0.20 ±0.01 | **0.02 ±0.00** |
| | **kNN-u** | **0.23 ±0.01** | 0.16 ±0.00 | 0.21 ±0.01 | **0.02 ±0.00** |
| | GAIN | 0.24 ±0.01 | 0.17 ±0.01 | 0.27 ±0.01 | 32.59 ±5.86 |
| | GRAPE | 0.38 ±0.01 | 0.30 ±0.00 | 0.38 ±0.02 | 2,009 ±336 |
| | HyperImpute | 0.24 ±0.01 | **0.13 ±0.00** | **0.14 ±0.02** | 25.99 ±7.05 |
| | MIRACLE | 4.37 ±0.16 | 4.29 ±0.16 | 10.33 ±0.38 | 4.68 ±0.32 |
| | NewImp | 0.81 ±0.04 | 0.74 ±0.05 | 1.79 ±0.11 | 1,256 ±328 |
| | Remasker | **0.23 ±0.01** | 0.14 ±0.01 | 0.16 ±0.01 | 751 ±193 |
| | TDM | 0.29 ±0.01 | 0.23 ±0.01 | 0.21 ±0.01 | 334 ±57 |
| Gottlieb | **F3I** | 0.04 ±0.00 | 0.02 ±0.00 | 0.02 ±0.00 | 2.62 ±0.66 |
| | **kNN-d** | 0.04 ±0.03 | 0.02 ±0.00 | 0.02 ±0.00 | **0.27 ±0.03** |
| | **kNN-u** | 0.04 ±0.00 | 0.02 ±0.00 | 0.03 ±0.00 | 0.28 ±0.03 |
| | GAIN | 0.04 ±0.00 | 0.02 ±0.00 | 0.02 ±0.00 | 245 ±16.14 |
| | GRAPE | 0.13 ±0.07 | 0.10 ±0.01 | 0.12 ±0.02 | 91,778 ±8,362 |
| | HyperImpute | **0.03 ±0.00** | **0.01 ±0.00** | **0.01 ±0.00** | 21.09 ±5.14 |
| | MIRACLE | 4.42 ±0.17 | 4.37 ±0.16 | 10.46 ±0.40 | 245 ±21.34 |
| | NewImp | 37.20 ±16.91 | 17.58 ±6.87 | 42.05 ±16.50 | 6,731 ±1,545 |
| | Remasker | 0.15 ±0.03 | 0.11 ±0.03 | 0.23 ±0.07 | 31,652 ±2,869 |
| | TDM | 0.14 ±0.00 | 0.11 ±0.00 | 0.11 ±0.05 | 3,870 ±489 |
| HeartDisease | **F3I** | 0.35 ±0.03 | 0.26 ±0.02 | - | 0.11 ±0.02 |
| | **kNN-d** | 0.34 ±0.03 | 0.24 ±0.02 | - | **0.01 ±0.00** |
| | **kNN-u** | **0.33 ±0.03** | 0.24 ±0.02 | - | **0.01 ±0.00** |
| | GAIN | 0.35 ±0.03 | 0.25 ±0.02 | - | 19.46 ±6.53 |
| | GRAPE | 0.56 ±0.02 | 0.45 ±0.01 | - | 1,606 ±255 |
| | HyperImpute | 0.38 ±0.04 | **0.22 ±0.02** | - | 34.51 ±4.62 |
| | MIRACLE | 4.43 ±0.18 | 4.35 ±0.18 | - | 4.98 ±0.48 |
| | NewImp | 0.65 ±0.06 | 0.53 ±0.06 | - | 767 ±272 |
| | Remasker | 0.34 ±0.03 | 0.25 ±0.02 | - | 457 ±133 |
| | TDM | 0.38 ±0.02 | 0.30 ±0.02 | - | 369 ±46.53 |
| Ionosphere | **F3I** | 0.20 ±0.01 | 0.11 ±0.01 | 0.17 ±0.01 | 0.40 ±0.29 |
| | **kNN-d** | **0.19 ±0.01** | **0.10 ±0.00** | **0.16 ±0.01** | **0.02 ±0.00** |
| | **kNN-u** | **0.19 ±0.01** | **0.10 ±0.00** | 0.17 ±0.02 | **0.02 ±0.00** |
| | GAIN | 0.21 ±0.01 | 0.13 ±0.01 | 0.20 ±0.02 | 32.27 ±6.04 |
| | GRAPE | 0.42 ±0.01 | 0.34 ±0.01 | 0.37 ±0.05 | 3,819 ±706 |
| | HyperImpute | 0.20 ±0.01 | 0.12 ±0.01 | 0.20 ±0.04 | 24.14 ±4.49 |
| | MIRACLE | 4.84 ±0.26 | 4.77 ±0.26 | 11.50 ±0.59 | 8.19 ±0.53 |
| | NewImp | 0.50 ±0.05 | 0.40 ±0.04 | 0.96 ±0.09 | 1,173 ±310 |

| | | | | |
|---|---|---|---|---|
| Remasker | 0.29 ±0.02 | 0.20 ±0.02 | 0.30 ±0.04 | 935 ±229 |
| TDM | 0.38 ±0.01 | 0.29 ±0.01 | 0.27 ±0.02 | 408 ±50 |

Table 10: Average and standard deviation values of imputation metrics and runtime across 10 different random seeds. kNN-d (respectively, kNN-u) is the distance-dependent (resp., uniformly) weighted kNN algorithm.

| Data set | Algorithm | RMSE ↓ | MAE ↓ | WD ↓ | Runtime ↓ |
|---|---|---|---|---|---|
| Spambase | **F3I** | 0.07 ±0.00 | **0.02 ±0.00** | **0.02 ±0.00** | 36.79 ±0.00 |
| | **kNN-d** | **0.06 ±0.00** | **0.02 ±0.00** | **0.02 ±0.00** | 3.49 ±0.00 |
| | **kNN-u** | **0.06 ±0.00** | **0.02 ±0.00** | 0.03 ±0.00 | **3.44 ±0.00** |
| | GAIN | 0.08 ±0.00 | 0.03 ±0.00 | 0.07 ±0.00 | 14.11 ±0.00 |
| | GRAPE | 0.09 ±0.00 | 0.04 ±0.00 | 0.06 ±0.00 | 20,667 ±0.00 |
| | HyperImpute | **0.06 ±0.00** | 0.03 ±0.00 | 0.04 ±0.00 | 45.08 ±0.00 |
| | MIRACLE | **0.06 ±0.00** | 0.03 ±0.00 | 0.06 ±0.00 | 62.16 ±0.00 |
| | NewImp | 0.13 ±0.00 | 0.10 ±0.00 | 0.24 ±0.00 | 46,563 ±0.00 |
| | Remasker | **0.06 ±0.00** | **0.02 ±0.00** | 0.03 ±0.00 | 13,650 ±0.00 |
| | TDM | 0.20 ±0.00 | 0.15 ±0.00 | 0.29 ±0.00 | 482.73 ±0.00 |
| Wine (White) | **F3I** | 0.11 ±0.00 | 0.07 ±0.00 | 0.07 ±0.00 | 27.95 ±6.15 |
| | **kNN-d** | **0.08 ±0.00** | **0.05 ±0.00** | 0.05 ±0.00 | 2.10 ±0.45 |
| | **kNN-u** | 0.09 ±0.00 | 0.06 ±0.00 | 0.07 ±0.01 | **2.07 ±0.42** |
| | GAIN | 0.11 ±0.01 | 0.08 ±0.01 | 0.13 ±0.01 | 24.97 ±6.06 |
| | GRAPE | 0.22 ±0.00 | 0.17 ±0.00 | 0.27 ±0.01 | 14,437 ±3,174 |
| | HyperImpute | **0.08 ±0.00** | **0.05 ±0.00** | **0.04 ±0.00** | 47.02 ±18.59 |
| | MIRACLE | 0.10 ±0.01 | 0.07 ±0.00 | 0.07 ±0.01 | 37.02 ±9.48 |
| | NewImp | 0.44 ±0.04 | 0.36 ±0.04 | 0.87 ±0.09 | 204,520 ±910 |
| | Remasker | **0.08 ±0.00** | **0.05 ±0.00** | 0.06 ±0.00 | 8,211 ±151 |
| | TDM | 0.16 ±0.01 | 0.13 ±0.01 | 0.10 ±0.01 | 258.66 ±28.38 |
| Wine (Red) | **F3I** | 0.12 ±0.01 | 0.09 ±0.01 | 0.09 ±0.00 | 2.05 ±0.45 |
| | **kNN-d** | **0.09 ±0.00** | **0.06 ±0.00** | 0.06 ±0.01 | 0.18 ±0.04 |
| | **kNN-u** | 0.10 ±0.00 | 0.07 ±0.00 | 0.08 ±0.01 | **0.17 ±0.03** |
| | GAIN | 0.12 ±0.01 | 0.09 ±0.01 | 0.13 ±0.02 | 3.46 ±1.22 |
| | GRAPE | 0.26 ±0.01 | 0.20 ±0.01 | 0.29 ±0.02 | 3,449 ±583 |
| | HyperImpute | **0.09 ±0.00** | **0.06 ±0.00** | **0.04 ±0.00** | 27.51 ±5.86 |
| | MIRACLE | 0.21 ±0.02 | 0.15 ±0.01 | 0.14 ±0.02 | 15.85 ±2.44 |
| | NewImp | 0.49 ±0.02 | 0.42 ±0.02 | 1.03 ±0.06 | 11,459 ±2,815 |
| | Remasker | 0.10 ±0.00 | 0.07 ±0.00 | 0.09 ±0.00 | 812.88 ±180 |
| | TDM | 0.20 ±0.01 | 0.16 ±0.01 | 0.11 ±0.01 | 223.31 ±49.37 |
| Libras | **F3I** | 0.08 ±0.00 | 0.06 ±0.00 | 0.07 ±0.00 | 0.83 ±0.22 |
| | **kNN-d** | 0.08 ±0.00 | 0.06 ±0.00 | 0.07 ±0.01 | **0.05 ±0.00** |
| | **kNN-u** | 0.09 ±0.00 | 0.07 ±0.00 | 0.08 ±0.01 | **0.05 ±0.01** |
| | GAIN | 0.05 ±0.00 | 0.03 ±0.00 | 0.05 ±0.00 | 24.11 ±1.11 |
| | GRAPE | 0.29 ±0.01 | 0.24 ±0.01 | 0.14 ±0.01 | 13,037 ±25 |
| | HyperImpute | **0.02 ±0.00** | **0.01 ±0.00** | **0.02 ±0.00** | 33.17 ±3.89 |
| | MIRACLE | 4.11 ±0.27 | 4.03 ±0.26 | 9.80 ±0.76 | 59.95 ±2.81 |
| | NewImp | 0.38 ±0.04 | 0.31 ±0.04 | 0.73 ±0.08 | 4,312 ±34 |
| | Remasker | 0.16 ±0.01 | 0.13 ±0.01 | 0.23 ±0.03 | 1,442 ±5.86 |
| | TDM | 0.31 ±0.01 | 0.25 ±0.01 | 0.14 ±0.02 | 687 ±5.93 |

Table 11: Average runtime (in seconds) and RMSE score across 10 iterations across the three missingness mechanisms, for a missing rate of 30%. RMSE values are rounded to the closest second decimal place.

| Data set | Algorithm | MNAR | | MCAR | | MAR | |
| | | RMSE | Time | RMSE | Time | RMSE | Time |
|---|---|---|---|---|---|---|---|
| BreastCancer | **F3I** | 0.08 ±0.00 | 0.92 | 0.07 ±0.00 | 0.49 | 0.09 ±0.01 | 0.82 |
| | GAIN | 0.08 ±0.01 | 7.73 | 0.07 ±0.00 | 5.17 | 0.09 ±0.00 | 6.58 |
| | HyperImpute | **0.05 ±0.00** | 10.65 | **0.05 ±0.00** | 12.00 | **0.06 ±0.00** | 10.65 |
| | Remasker | 0.11 ±0.01 | 535 | 0.10 ±0.01 | 501 | 0.12 ±0.01 | 528 |
| Diabetes | **F3I** | 0.22 ±0.03 | 0.31 | 0.25 ±0.01 | 0.33 | 0.24 ±0.06 | 0.23 |
| | GAIN | 0.22 ±0.03 | 4.97 | **0.23 ±0.00** | 6.30 | **0.21 ±0.04** | 4.84 |
| | HyperImpute | **0.19 ±0.03** | 20.84 | 0.24 ±0.01 | 38.06 | **0.21 ±0.06** | 23.01 |
| | Remasker | 0.20 ±0.03 | 186 | **0.23 ±0.00** | 184 | **0.21 ±0.04** | 187 |
| Gottlieb | **F3I** | 0.04 ±0.00 | 4.09 | 0.04 ±0.00 | 4.15 | 0.04 ±0.00 | 3.31 |
| | GAIN | 0.04 ±0.00 | 101 | 0.04 ±0.00 | 123 | 0.04 ±0.00 | 104 |
| | HyperImpute | **0.02 ±0.00** | 20.98 | **0.03 ±0.00** | 23.31 | **0.03 ±0.01** | 25.38 |
| | Remasker | 0.20 ±0.04 | 11,094 | 0.15 ±0.03 | 11,380 | 0.17 ±0.03 | 11,121 |
| HeartDisease | **F3I** | 0.37 ±0.03 | 0.15 | 0.35 ±0.03 | 0.14 | − | 0.08 |
| | GAIN | 0.37 ±0.03 | 6.03 | 0.35 ±0.03 | 5.33 | − | 5.99 |
| | HyperImpute | 0.39 ±0.04 | 90.79 | 0.36 ±0.04 | 89.10 | − | 48.84 |
| | Remasker | **0.35 ±0.03** | 179 | **0.34 ±0.03** | 188 | − | 185 |
| Ionosphere | **F3I** | **0.19 ±0.01** | 0.29 | **0.20 ±0.01** | 0.21 | **0.17 ±0.01** | 0.17 |
| | GAIN | 0.21 ±0.01 | 7.14 | 0.21 ±0.01 | 5.86 | 0.19 ±0.01 | 6.66 |
| | HyperImpute | **0.19 ±0.01** | 43.02 | **0.20 ±0.01** | 50.52 | 0.18 ±0.01 | 26.04 |
| | Remasker | 0.24 ±0.01 | 255 | 0.26 ±0.02 | 258 | 0.24 ±0.02 | 257 |
| Spambase | **F3I** | 0.06 ±0.01 | 150 | 0.05 ±0.00 | 59.55 | 0.06 ±0.01 | 116 |
| | GAIN | 0.06 ±0.01 | 26.00 | 0.05 ±0.01 | 11.14 | 0.06 ±0.01 | 20.60 |
| | HyperImpute | **0.05 ±0.00** | 119 | 0.05 ±0.00 | 61.83 | **0.05 ±0.01** | 104.87 |
| | Remasker | **0.05 ±0.00** | 8,355 | 0.05 ±0.00 | 8,316 | **0.05 ±0.01** | 11,831 |
| Wine (White) | **F3I** | 0.11 ±0.01 | 25.92 | 0.10 ±0.01 | 19.93 | 0.10 ±0.01 | 14.67 |
| | GAIN | 0.11 ±0.01 | 6.16 | 0.10 ±0.01 | 4.73 | 0.10 ±0.01 | 5.02 |
| | HyperImpute | **0.08 ±0.00** | 26.60 | **0.08 ±0.00** | 30.74 | **0.07 ±0.00** | 36.70 |
| | Remasker | **0.08 ±0.00** | 2,836 | **0.08 ±0.00** | 2,225 | 0.08 ±0.01 | 2,691 |
| Wine (Red) | **F3I** | 0.12 ±0.01 | 2.90 | 0.12 ±0.01 | 2.44 | 0.11 ±0.01 | 1.01 |
| | GAIN | 0.12 ±0.00 | 4.85 | 0.12 ±0.01 | 4.31 | 0.13 ±0.01 | 3.60 |
| | HyperImpute | **0.09 ±0.00** | 28.10 | **0.09 ±0.00** | 22.58 | **0.10 ±0.01** | 24.29 |
| | Remasker | 0.10 ±0.00 | 822 | **0.09 ±0.00** | 682 | **0.10 ±0.01** | 787 |
| Libras | **F3I** | 0.08 ±0.00 | 0.33 | 0.08 ±0.00 | 0.33 | 0.07 ±0.00 | 0.22 |
| | GAIN | 0.05 ±0.00 | 11.02 | 0.04 ±0.00 | 11.45 | 0.04 ±0.00 | 10.90 |
| | HyperImpute | **0.02 ±0.00** | 10.31 | **0.02 ±0.00** | 11.84 | **0.02 ±0.00** | 9.45 |
| | Remasker | 0.12 ±0.02 | 771 | 0.12 ±0.02 | 803 | 0.13 ±0.02 | 733 |

Table 12: Average of RMSE score over 10 seeds (rounded to the closest second decimal place) on the Ionosphere dataset for varying missing rates (in %) and missingness mechanisms (MCAR, MAR, MNAR). kNN-d (respectively, kNN-u) is the distance-dependent (resp., uniformly) weighted kNN algorithm.

|  | % | F3I | kNN-d | kNN-u | HyperImpute | MIRACLE | GAIN | Remasker | GRAPE |
|---|---|---|---|---|---|---|---|---|---|
| M | 10 | 0.19 ±0.0 | 0.19 ±0.0 | 0.19 ±0.0 | 0.19 ±0.0 | 4.77 ±0.1 | 0.21 ±0.0 | 0.29 ±0.0 | 0.40 ±0.0 |
| C | 25 | 0.19 ±0.0 | 0.19 ±0.0 | 0.19 ±0.0 | 0.19 ±0.0 | 4.62 ±0.2 | 0.20 ±0.0 | 0.27 ±0.0 | 0.41 ±0.0 |
| A | 50 | 0.20 ±0.0 | 0.19 ±0.0 | 0.19 ±0.0 | 0.20 ±0.0 | 4.43 ±0.3 | 0.21 ±0.0 | 0.26 ±0.0 | 0.43 ±0.0 |
| R | 75 | 0.21 ±0.0 | 0.20 ±0.0 | 0.20 ±0.0 | 0.21 ±0.0 | 4.60 ±0.7 | 0.23 ±0.0 | 0.26 ±0.0 | 0.46 ±0.0 |
|  | 90 | 0.22 ±0.0 | 0.21 ±0.0 | 0.22 ±0.0 | 0.22 ±0.0 | 3.94 ±0.6 | 0.40 ±0.0 | 0.27 ±0.0 | 0.46 ±0.0 |
| M | 10 | 0.17 ±0.0 | 0.17 ±0.0 | 0.17 ±0.0 | 0.18 ±0.0 | 4.68 ±0.2 | 0.19 ±0.0 | 0.29 ±0.0 | 0.40 ±0.0 |
| A | 25 | 0.18 ±0.0 | 0.17 ±0.0 | 0.17 ±0.0 | 0.18 ±0.0 | 5.05 ±0.2 | 0.18 ±0.0 | 0.30 ±0.0 | 0.43 ±0.0 |
| R | 50 | 0.21 ±0.0 | 0.20 ±0.0 | 0.20 ±0.0 | 0.21 ±0.0 | 4.66 ±0.3 | 0.21 ±0.0 | 0.26 ±0.0 | 0.42 ±0.0 |
|  | 75 | 0.20 ±0.0 | 0.19 ±0.0 | 0.19 ±0.0 | 0.20 ±0.0 | 4.58 ±0.5 | 0.21 ±0.0 | 0.26 ±0.0 | 0.42 ±0.0 |
|  | 90 | 0.22 ±0.0 | 0.21 ±0.0 | 0.22 ±0.0 | 0.22 ±0.0 | 4.71 ±0.6 | 0.27 ±0.0 | 0.32 ±0.0 | 0.44 ±0.0 |
| M | 10 | 0.19 ±0.0 | 0.18 ±0.0 | 0.18 ±0.0 | 0.19 ±0.0 | 4.73 ±0.1 | 0.20 ±0.0 | 0.29 ±0.0 | 0.41 ±0.0 |
| N | 25 | 0.19 ±0.0 | 0.19 ±0.0 | 0.19 ±0.0 | 0.20 ±0.0 | 4.42 ±0.3 | 0.20 ±0.0 | 0.26 ±0.0 | 0.42 ±0.0 |
| A | 50 | 0.19 ±0.0 | 0.18 ±0.0 | 0.18 ±0.0 | 0.19 ±0.0 | 4.42 ±0.2 | 0.21 ±0.0 | 0.25 ±0.0 | 0.43 ±0.0 |
| R | 75 | 0.20 ±0.0 | 0.19 ±0.0 | 0.20 ±0.0 | 0.21 ±0.0 | 4.48 ±0.6 | 0.26 ±0.0 | 0.26 ±0.1 | 0.46 ±0.0 |
|  | 90 | 0.21 ±0.0 | 0.21 ±0.0 | 0.21 ±0.0 | 0.22 ±0.0 | 3.86 ±0.4 | 0.40 ±0.0 | 0.28 ±0.0 | 0.45 ±0.0 |

Table 13: Average of RMSE score over 10 seeds (rounded to the closest second decimal place) on the BreastCancer dataset for varying missing rates (in %) and missingness mechanisms (MCAR, MAR, MNAR). kNN-d (respectively, kNN-u) is the distance-dependent (resp., uniformly) weighted kNN algorithm.

|  | % | F3I | kNN-d | kNN-u | HyperImpute | MIRACLE | GAIN | Remasker | GRAPE |
|---|---|---|---|---|---|---|---|---|---|
| M | 10 | 0.07 ±0.0 | 0.07 ±0.0 | 0.07 ±0.0 | 0.04 ±0.0 | 4.15 ±0.1 | 0.07 ±0.0 | 0.11 ±0.0 | 0.22 ±0.0 |
| C | 25 | 0.07 ±0.0 | 0.07 ±0.0 | 0.07 ±0.0 | 0.04 ±0.0 | 4.16 ±0.1 | 0.07 ±0.0 | 0.10 ±0.0 | 0.24 ±0.0 |
| A | 50 | 0.07 ±0.0 | 0.07 ±0.0 | 0.07 ±0.0 | 0.05 ±0.0 | 4.15 ±0.4 | 0.07 ±0.0 | 0.09 ±0.0 | 0.23 ±0.0 |
| R | 75 | 0.08 ±0.0 | 0.08 ±0.0 | 0.08 ±0.0 | 0.06 ±0.0 | 3.99 ±0.6 | 0.08 ±0.0 | 0.09 ±0.0 | 0.24 ±0.0 |
|  | 90 | 0.08 ±0.0 | 0.09 ±0.0 | 0.09 ±0.0 | 0.07 ±0.0 | 3.39 ±0.4 | 0.13 ±0.0 | 0.09 ±0.0 | 0.24 ±0.0 |
| M | 10 | 0.08 ±0.0 | 0.09 ±0.0 | 0.09 ±0.0 | 0.06 ±0.0 | 4.25 ±0.1 | 0.08 ±0.0 | 0.12 ±0.0 | 0.26 ±0.0 |
| A | 25 | 0.08 ±0.0 | 0.09 ±0.0 | 0.09 ±0.0 | 0.05 ±0.0 | 4.21 ±0.1 | 0.08 ±0.0 | 0.11 ±0.0 | 0.26 ±0.0 |
| R | 50 | 0.08 ±0.0 | 0.09 ±0.0 | 0.09 ±0.0 | 0.06 ±0.0 | 4.12 ±0.2 | 0.08 ±0.0 | 0.10 ±0.0 | 0.24 ±0.0 |
|  | 75 | 0.08 ±0.0 | 0.09 ±0.0 | 0.09 ±0.0 | 0.06 ±0.1 | 3.92 ±0.3 | 0.09 ±0.0 | 0.10 ±0.0 | 0.24 ±0.0 |
|  | 90 | 0.10 ±0.0 | 0.10 ±0.0 | 0.10 ±0.0 | 0.08 ±0.0 | 3.82 ±0.4 | 0.10 ±0.0 | 0.12 ±0.0 | 0.25 ±0.0 |
| M | 10 | 0.08 ±0.0 | 0.09 ±0.0 | 0.09 ±0.0 | 0.05 ±0.0 | 4.21 ±0.1 | 0.08 ±0.0 | 0.12 ±0.0 | 0.25 ±0.0 |
| N | 25 | 0.08 ±0.0 | 0.08 ±0.0 | 0.08 ±0.0 | 0.05 ±0.0 | 4.47 ±0.2 | 0.08 ±0.0 | 0.11 ±0.0 | 0.25 ±0.0 |
| A | 50 | 0.08 ±0.0 | 0.08 ±0.0 | 0.08 ±0.0 | 0.06 ±0.0 | 4.39 ±0.4 | 0.09 ±0.1 | 0.10 ±0.0 | 0.25 ±0.0 |
| R | 75 | 0.08 ±0.0 | 0.08 ±0.0 | 0.09 ±0.0 | 0.06 ±0.0 | 3.71 ±0.5 | 0.11 ±0.0 | 0.09 ±0.0 | 0.23 ±0.0 |
|  | 90 | 0.09 ±0.0 | 0.09 ±0.0 | 0.09 ±0.0 | 0.08 ±0.0 | 4.03 ±0.4 | 0.15 ±0.0 | 0.10 ±0.0 | 0.24 ±0.0 |

Moreover, we also benchmark the nearest-neighbor approaches against the other baselines across all three missingness scenarios (MCAR, MAR and MNAR) for the Ionosphere and BreastCancer datasets for missingness rates from 10%-90% in Tables 12 and 13 respectively, corresponding to Figure 5.

### G.2.2 Empirical evaluation of the extension to downstream tasks

Table 14 provides full numerical details for Figure 6.

Table 14: Average and standard deviation of Area Under the Curve (AUC) values (rounded to the closest second decimal place) across 20 runs on the joint imputation-classification task (MCAR scenario, $p^{\text{miss}} = 50\%$). kNN-d (respectively, kNN-u) is the distance-dependent (resp., uniformly) weighted kNN algorithm. *Remasker and HyperImpute are extremely slow when combined with a MLP, especially on the largest data set PREDICT, which is why the values are missing.

| Task | Algorithm | BreastCancer | Ionosphere | MNIST | PREDICT |
|---|---|---|---|---|---|
| Joint Imputation-Classification | GRAPE | 0.539 ±0.059 | 0.738 ±0.060 | 0.998 ±0.001 | 0.481 ±0.017 |
| | NeuMiss | 0.461 ±0.016 | 0.675 ±0.056 | 0.813 ±0.232 | 0.501 ±0.003 |
| | **PCGradF3I** | 0.595 ±0.064 | 0.732 ±0.046 | **1.000 ±0.000** | 0.482 ±0.018 |
| Separate Imputation-Classification | HyperImpute | 0.571 ±0.151 | 0.757 ±0.061 | —* | —* |
| | **kNN-d** | 0.671 ±0.132 | 0.760 ±0.046 | 0.922 ±0.177 | 0.475 ±0.021 |
| | **kNN-u** | 0.594 ±0.137 | 0.750 ±0.061 | 0.922 ±0.177 | 0.468 ±0.019 |
| | **F3I** | 0.550 ±0.059 | 0.734 ±0.049 | 0.922 ±0.177 | 0.495 ±0.013 |
| | Mean | 0.500 ±0.000 | 0.740 ±0.038 | 0.575 ±0.178 | 0.474 ±0.019 |
| | Remasker | 0.500 ±0.000 | 0.807 ±0.040 | —* | —* |
| | RF-GAP | **0.851 ±0.074** | **0.879 ±0.048** | **1.000 ±0.000** | **0.545 ±0.028** |
| | | Spambase | HeartDisease | Wine (Red) | |
| Joint Imputation-Classification | GRAPE | 0.441 ±0.061 | 0.518 ±0.045 | 0.527 ±0.095 | |
| | NeuMiss | 0.499 ±0.001 | 0.535 ±0.051 | 0.527 ±0.070 | |
| | **PCGradF3I** | 0.608 ±0.021 | 0.492 ±0.061 | 0.506 ±0.094 | |
| Separate Imputation-Classification | HyperImpute | 0.215 ±0.025 | 0.582 ±0.130 | 0.591 ±0.043 | |
| | **kNN-d** | 0.273 ±0.014 | **0.665 ±0.079** | 0.544 ±0.119 | |
| | **kNN-u** | 0.269 ±0.018 | 0.530 ±0.087 | 0.582 ±0.042 | |
| | **F3I** | 0.364 ±0.013 | 0.493 ±0.050 | 0.558 ±0.105 | |
| | Mean | 0.331 ±0.039 | 0.500 ±0.000 | **0.722 ±0.076** | |
| | Remasker | 0.251 ±0.029 | 0.500 ±0.000 | 0.500 ±0.000 | |
| | RF-GAP | **0.725 ±0.212** | 0.501 ±0.019 | 0.500 ±0.000 | |

