# OpenReview forum: "Nearest-Neighbor Imputation with Error Guarantees and Extensions for Mixed-Type Data and Joint Learning"
_TMLR — Withdrawn by Authors_

### Review · Reviewer_oB6s · 2026-06-12

**Summary Of Contributions:**

The paper studies nearest-neighbor methods for filling in missing data. The authors argue that these methods are still strong, fast, and easier to analyse than many newer imputation methods. They give theoretical error bounds for weighted nearest-neighbor imputation under different missing-data settings, and propose F3I, a method that learns how much weight to give to each neighbor. They also extend the method to joint imputation and prediction, and to data with both numerical and categorical features.

A key strength of the paper is that it takes a simple method seriously and shows that it can be competitive while being much faster than more complex alternatives. The paper is also useful because it combines theory, practical algorithms, and experiments on both synthetic and real datasets.

The main weakness is that some claims feel stronger than the evidence. The theory relies on fairly restrictive assumptions, and the real-world experiments do not fully show when those assumptions matter. The paper would be stronger with clearer statistical comparisons, more ablations, and a more controlled runtime analysis. In particular, the authors should better isolate whether the learned weights in F3I are really responsible for the reported gains.

Overall, this is a useful and interesting paper. It shows that nearest-neighbor imputation remains a strong baseline and possibly a practical method in its own right, but some claims should either be narrowed or supported with more direct evidence.

**Audience:**

Yes

**Audience Explanation:**

I think some readers in the TMLR audience would be interested in this paper. Missing-data imputation is a common problem in machine learning, and the paper makes a useful point: simple nearest-neighbor methods can still be competitive, fast, and easier to analyse than many more complex alternatives. The theoretical error bounds, the learned-weight version of kNN imputation, and the extensions to downstream prediction and mixed-type data are all relevant to readers working on practical machine learning methods.

**Broader Impact Concerns:**

Imputation methods can be used in sensitive domains such as healthcare, finance, or social data analysis. In such settings, poor imputation can introduce bias, hide uncertainty, or make downstream predictions appear more reliable than they really are. This is especially relevant when missingness is not random, since missing values may reflect social, clinical, or measurement processes that should not simply be treated as noise. Perhaps the authors should briefly acknowledge that imputed values should not be treated as observed ground truth, especially in high-stakes applications. They could also mention that downstream users should check performance across relevant subgroups and report uncertainty when imputation affects decisions.

**Claims And Evidence:**

No

**Claims Explanation:**

The main claims are supported by relevant evidence. The paper gives a clear theoretical analysis of weighted nearest-neighbor imputation, proposes practical extensions, and evaluates the methods on both synthetic and real datasets. The experiments generally support the claim that nearest-neighbor imputation can be competitive while being much faster than more complex alternatives.
However, some claims should be stated more carefully. The theoretical guarantees rely on restrictive assumptions, so they do not fully explain the real-world results. The empirical evidence is also not always strong enough to support the broadest claims, especially about state-of-the-art performance, the benefit of learned weights, and practical runtime advantages. These would be more convincing with clearer statistical comparisons, stronger ablations, and more controlled runtime reporting.
Overall, I think the evidence is accurate and relevant, but not always as complete as it could be. The authors should narrow some claims or add evidence where the current support is weaker.

**Requested Changes:**

First, the authors should make the main claims more precise. In particular, they should separate more clearly what is proved by the theory from what is shown empirically. The theoretical results rely on specific assumptions about the data distribution and the missingness mechanism. These assumptions are useful for analysis, but they are unlikely to hold exactly in the real-world datasets. I therefore suggest that the authors explicitly state, near the theoretical results and again in the discussion, that the guarantees apply under these assumptions, while the real-data experiments show empirical robustness outside this idealized setting. This would avoid giving the impression that the theory directly explains all empirical results.

Second, the empirical comparisons would be stronger with clearer statistical evidence. The current results are useful, but the paper would benefit from paired comparisons across datasets, confidence intervals, win/loss/tie summaries, and critical difference diagrams. For example, the authors could report, for each metric, how often F3I is better, similar to, or worse than each main baseline, and complement this with average-rank plots showing which differences are statistically meaningful across datasets. They could also report paired differences rather than only raw scores. This would make it easier to judge whether the observed improvements are consistent across the benchmark suite or whether they are driven by a small number of datasets.


Third, the authors should strengthen the ablation study around F3I. Since one of the main ideas of the paper is to learn neighbor weights, it would be useful to isolate the contribution of this component more directly. The authors could compare F3I against uniform kNN, distance-weighted kNN, fixed learned weights, random weights, and a version of F3I without the distribution-preserving objective. For the joint-learning version, they could also compare PCGrad-F3I with the same method without PCGrad. These ablations would help show whether the proposed ingredients are actually responsible for the gains.

Fourth, the runtime comparison should be made more controlled and more central. Since speed is one of the strongest motivations for the paper, the authors should report wall-clock time, hardware, memory use, tuning budget, and implementation details for all methods. It would also be useful to include scaling experiments where one parameter is varied at a time, such as the number of samples, the dimensionality of the data, the missingness rate, or the number of clusters in the data distribution, while the other factors are kept fixed. This would show more clearly how the runtime behaves as the problem becomes larger or structurally more complex. Such experiments would make the practical efficiency claim much more convincing, especially for high-dimensional datasets.


Fifth, the authors should narrow some broad statements about state-of-the-art performance and mixed-type data, unless they add more experiments. The current evidence supports the view that nearest-neighbor imputation is a strong and efficient alternative under the tested settings. However, it does not fully establish broad superiority across all missingness mechanisms, dataset types, or mixed-data regimes. For mixed-type data, it would be useful to test more categorical-heavy datasets and more missingness patterns, or otherwise state the conclusion more cautiously.

Finally, the authors should improve the discussion of when the method is expected to work well or poorly. For example, the paper could say more about high-dimensional settings, sparse data, strongly correlated features, high missingness rates, and datasets where nearest-neighbor structure may be weak. This would help readers understand the boundary conditions of the method and would make the contribution more useful in practice.

---

> ### Author Response · Authors · 2026-06-18
> **Asking for clarification**
>
> We thank the reviewer for their work. We are currently working on the requested experiments and changes to the manuscript. There are a few questions where we would like the reviewer's input:
>
> - What does the reviewer refer to when mentioning as a variant algorithm F3I with "fixed learned weights''? F3I is already learning weights and applying them during the imputation procedure.
>
> - Likewise, it is unclear to us what F3I would be without the distribution-preserving objective. Function G guides the weight learning procedure.
>
> - Hardware and implementation details are already available in Appendix G of the manuscript (before section G.1) in addition to the supplementary material provided.
>
> - What does the reviewer mean by "mixed-data regimes''?
>
> We thank again the reviewer for helping us to improve the submission.

---

> > ### Comment · Reviewer_oB6s · 2026-06-24
> >
> > By “fixed learned weights”, I meant a simple variant where the weights learned by F3I are learned once, or on a validation/training split, and then kept fixed during imputation, rather than being updated iteratively. The point would be to separate the benefit of the final weight pattern from the benefit of the full iterative learning procedure.
> >
> > By “F3I without the distribution-preserving objective”, I meant an ablation where the update is not driven by the density-ratio objective (G), or where this term is replaced by a simpler objective. I understand that (G) is central to the proposed method, so this may be better framed as an ablation of the contribution of (G), rather than as a fully meaningful alternative algorithm.
> >
> > On hardware and implementation details my suggestion is only that the main paper should make the runtime setup easier to see, since speed is one of the main claims of the submission.
> >
> > By “mixed-data regimes”, I meant datasets with different balances and structures of continuous and categorical variables. For example, it would be useful to know how the method behaves when most variables are categorical, when there are many categorical levels, when categorical and continuous variables have very different scales, or when missingness affects categorical and continuous variables differently. My concern was not just whether mixed-type data are supported, but whether the empirical evidence shows where this extension works well and where it may be less reliable.

---

> ### Author Response · Authors · 2026-07-16
>
> Due to large sizes of tables and figures, we put all tables and figures in an anonymous repository: [https://anonymous.4open.science/r/TMLR_Review-BFE7/TMLR_Response.pdf](https://anonymous.4open.science/r/TMLR_Review-BFE7/TMLR_Response.pdf).
> # Assumptions of theoretical results
> We will postpone the proof on the high-probability upper bound on the MSE loss to the Appendix, to leave space to write in Section 2.1: "Note that, *albeit the theoretical results below rely on specific assumptions on data distribution and missingness mechanisms,* the *kNN* algorithms still work well in practice *on real-world data*, even without those assumptions being satisfied, as reported in the literature and in Section 4.", in Section 4.1: "We first empirically show that Theorem 2.3 is valid *under the set of assumptions described in Section 2.*", and in the discussion (section 6): "First, we proposed a general proof strategy for deriving upper bounds on the mean-squared-error loss on imputed values, which we obtained through a new concentration bound on the norm between the imputed and corresponding complete data points. *However, note that this result rely on a set of specific assumptions on the data distribution and missingness mechanisms.*", "*Based on the prior concentration bound, we* derived high-probability guarantees on the optimization of the underlying data distribution-preserving function*, under the same assumptions as before.*'', "Finally, our experimental study highlights the good performance-computational cost tradeoff achieved by nearest-neighbor algorithms for imputation-only and joint imputation-classification tasks despite their simplicity *and outside our idealized theoretical setting*, while empirically confirming our error bounds *in a context where all the theoretical assumptions are valid*.".
>
> # Statistical comparisons across datasets
> We thank the reviewer for this suggestion. In order to measure whether the paired differences were significant, we conducted the one-sided one-sample Wilcoxon signed rank test on the paired difference of performance metrics and runtime of the best-performing nearest neighbour method (here $\text{Metric}\_\text{kNN}$ and $\text{Runtime}\_\text{kNN}$) and the best performing non-Nearest Neighbour method (here $\text{Metric}\_\text{other}$ and $\text{Runtime}\_\text{other}$) for the 10 random seed values. The "best" models in each category were chosen by picking the model with the maximum number of minimum average performance metric values for each metric. Ties were broken with the minimum average runtime values. For models where both performance metrics and runtime were equal, ties were broken arbitrarily. The two hypothesis set for the corresponding one-sided tests
> are:
> 1. $H_0:\text{Metric}\_\text{kNN}=\text{Metric}\_\text{other}$ against $H_1:\text{Metric}\_\text{kNN}>\text{Metric}\_\text{kNN}\;,$ where Metric corresponds to RMSE, MAE or WD,
>  2. $H_0:\text{Runtime}\_\text{kNN}=\text{Runtime}\_\text{other}$ against $H_1:\text{Runtime}\_\text{kNN}<\text{Runtime}\_\text{kNN}\;.$
>
>  The results are presented in Table 13.

---

> > ### Author Response · Authors · 2026-07-16
> >
> > # Refined ablation study in F3I and PCGradF3I
> > In addition to uniform-weight and distance-dependent weight-kNNs, we now compare F3I to a variant called "F3I Random" where weights are randomly sampled at each iteration (sampling K integer values from the interval $[1,K]$ uniformly, and then normalizing by the sum of those values). See Tables 2-8. We also consider a F3I variant with fixed weights, that is, where the full convex problem is solved and then the resulting weights are applied to the data once, and a variant where the G objective is replaced by the MSE loss compared to the average value of each feature. The corresponding expression (to maximize), gradient and hessian of this objective are given below (using notation from the paper):
> > $$ G^\text{MSE}(\boldsymbol{\alpha}, X) = \frac{-1}{N_\text{mask}} \sum_{i,f \mid m^f_i=1} (x_i^f(\boldsymbol{\alpha})-\overline{x}^f)^2 \text{ and }\overline{x}^f = \frac{1}{|\{i \mid m^f_i = 0\}|}\sum_{i, m^f_i = 0} x^f_i \text{ and } N_\text{mask} = |\{(i,f) \mid m^f_i=1\}|$$
> > $$\nabla_{\boldsymbol{\alpha}} G^\text{MSE}(\boldsymbol{\alpha}, X) = \frac{-2}{N_\text{mask}} \sum_{i,f \mid m^f_i=1} (x^f_i(\boldsymbol{\alpha})-\overline{x}^f)^\top (\widetilde{Z}^{n_i})^f \text{ and } \nabla^2_{\boldsymbol{\alpha}} G^\text{MSE}(\boldsymbol{\alpha}, X) = \frac{-2}{N_\text{mask}} \sum_{i,f \mid m^f_i=1} ((\widetilde{Z}^{n_i})^f)^\top (\widetilde{Z}^{n_i})^f$$
> > These results confirm what was observed in the plots and tables of the submission: all kNN algorithms have roughly the same imputation performance across data sets and metrics.
> > The difference between F3I variants and other kNN algorithms might be explained by the use of Chebyshev distance, instead of Euclidean distance, which is more robust to outliers and less prone to the curse of dimensionality in high-dimensional settings. The weight optimization part is the most useful when combined with a downstream task, as shown in Figure 6 of the manuscript where the difference between PCGradF3I and F3I is more noticeable.
> > We also ablated PCGradF3I by removing the gradient surgery aspect incurred by PCGrad. The results are presented in Table 20, where we observe that there is not much difference between the gradient surgery version and the ablated joint training version of the algorithm.
> > # Runtime comparison and scaling experiments
> > Note that the wall-clock time (called runtime in our submission and corresponding columns in Appendix), hardware specifications (page 36 of the submission) and implementation details (Section G and supplementary material) are already reported. Moreover, in addition to wall-clock time, we also report the memory use in the following scaling experiments. We first consider the same synthetic data sets as described in the submission, and modify the data generation procedure as follows: given a number of clusters $c$, we translate the mean of the Gaussian distribution (sampled from $\mathcal{N}(0,1)$) generating $\lceil 1/cN \rceil$ samples in the cluster $i_c$ by $i_c+\varepsilon$, generating $\lceil 1/cN \rceil \times d$ values for feature $f$ in cluster $i_c$ from distribution $\mathcal{N}(\mu_f+i_c, \sigma^2)$. We choose $\varepsilon=1$ for distinct clusters, and a smaller value to obtain data where the clustering structure is weaker (*i.e.*, clusters are not separated). If needed, supplementary samples are generated within the last cluster to obtain exactly $N$ samples. Then, we vary the number of samples, the dimensionality of the data, the missingness rate, the number of clusters and their separability (controlled by $\varepsilon$ in the data distribution, and run kNN algorithms (uniform, distance-based, F3I) and top baselines (HyperImpute, Remasker, GAIN, GRAPE, MIRACLE). Corresponding results are located in Tables 9-13. We comment on these results in paragraph 6 below.
> > # Mixed-type imputation
> > We performed additional tests on different missingness mechanisms and rates. The additional experiments in Tables 14-18 support our conclusion that in every single missingness mechanism and rate, MI-F3I outperforms HyperImpute while costing much less in runtime. These datasets have a diverse number and cardinality of categorical variables, as shown in Table 1.

---

> > > ### Author Response · Authors · 2026-07-16
> > >
> > > # Applicability of kNN algorithms
> > > As observed in Tables 9-13 on synthetic data sets, kNN algorithms are more adapted to large sample settings (large N) compared to other state-of-the-art baselines: although their memory usage is higher, their runtime is smaller (Table 9). As the dimensionality of data increases (Table 10), kNN algorithms remain slightly less performant than HyperImpute. Among kNN algorithms, F3I variants are more time-consuming, whereas classical kNN algorithms remain much faster than other baselines. As expected, when the missingness rate increases (Table 11), all performance metrics degrade for all algorithms, but F3I variants remain noticeably better than baselines, while being more computationally competitive time-wise. When the number of clusters increases (while the provided number of neighbors $K=3$ remains the same) in Table 12, the performance and time/memory consumption remain the same. As $100/20=5>K$, this is expected. Finally, we also looked at the case where the nearest-neighbor structure might be weak, as $\varepsilon$ decreases (Table 13). The fact that the clustering structure is weaker does not seem again to impact performance and time/memory consumption significantly. We will add the related experiments and comments to the submission. Note that the results in those tables might be biased due to the synthetic nature of the data.
> > > # Impact concerns
> > > We agree with the reviewer's opinion. We will add to the Broader Impact Statement that: "*Moreover, as noticed by prior works [[Le Morvan & Varoquaux, 2025: https://openreview.net/forum?id=D1Y2XFgsPI](https://openreview.net/forum?id=D1Y2XFgsPI), imputation methods maximizing downstream (e.g., classification) performance might introduce biases to imputed values and predictions, as good imputation does not necessarily correlate with good prediction. Moreover, imputed values do not provide uncertainty metrics. As such, imputed values should not be considered ground truth, especially in high-stakes applications, but a prediction about expected values. These predictions are likely to be sensitive to specific subgroups of samples, and performance/uncertainty across subgroups should be carefully assessed when making critical decisions.*"

---

### Review · Reviewer_Xurn · 2026-07-03

**Summary Of Contributions:**

The authors present a theoretical framework for missing data imputation using weighted nearest-neighbors (NN) approaches. They derive theoretical bounds for the imputation MSE when using NN to impute data under strong assumptions. Their F3I method extend this framework to mixed-type data and to downstream tasks (e.g. supervised learning) via PCGrad. Finally, these theoretical findings are validated through experiments in synthetic and real-world data, whose results corroborate the claim that NN imputation can be a cost-effective solution for missing data imputation.

**Strengths:**
The paper expands upon weighted NN imputation theory. In particular, the high probability upper bounds on the imputation mean squared error and downstream task error are the main takeaways of the paper. These contributions represent a step towards the direct minimization of the true imputation error by deriving theoretical guarantees.

**Weaknesses:**
Some assumptions made in the paper are too strong and may not enable missing data imputation in the proposed synthetic benchmarks, let alone real-world data. In addition, practical considerations were not completely explained concerning the proposed F3I method and its downstream variant.

**Additional Comments:**

- Have the authors considered benchmarking another downstream task such as regression?
- Can the authors provide some clue towards theoretical guarantees for categorical-type data?
- Are the authors familiar with the work by Pashmchi et al. (2025) on kNNSampler? Do you think their stochastic approach to NN imputation could benefit from F3I in any capacity?

**Audience:**

Yes

**Audience Explanation:**

The theoretical high-probability upper bounds on NN imputation are useful to TMLR's audience, especially those working in missing data or NN theory. On the other hand, the proposed imputation methods can be attractive to practitioners in the audience provided some practical considerations on implementation details and computational cost are addressed.

**Broader Impact Concerns:**

No comments.

**Claims And Evidence:**

No

**Claims Explanation:**

While the authors present solid theoretical studies on the properties of NN imputation, their claims of cost-effectiveness and broad applicability of the proposed method (F3I) are disputable. I believe further clarification of the training cost of F3I and its implementation details, as well as a revision of the necessary assumptions for the theory's development could change this verdict.

**Requested Changes:**

## Major
- Assumption B.4 is too strong because it implies that there is no correlation between features. Under this assumption, it can be shown that mean imputation would be the optimal strategy. Are there any means to relax this assumption and arrive at reasonable imputation bounds that may apply in more general scenarios?
- I believe some practical details of Section 3.1 need to be clarified. How is $G(\alpha, \mathbf{x})$ computed in practice? Is it necessary to fit a kernel density estimator at every iteration of F3I? How does this scale with dataset size?
- The authors claim that F3I leverages the cost-effectiveness of NN imputation to achieve error metrics comparable to state-of-the-art imputation methods. However, some results in Appendix Section G.2.1 contradict this statement, showing that F3I can take as much time to run as more demanding methods. Can the authors explain why they happened? Notable examples include:
  - Wine (White) and Spambase datasets in Table 10, where F3I takes as longer as GAIN to achieve similar performance.
  - Diabetes and HeartDisease datasets in Table 9, F3I can take 10 times longer to train than standard kNN to obtain slightly worse results.


## Minor
- There is a reference to Lemma 2.2 in pp. 6,  l. 31, however no such Lemma exists in the text. Does it refer instead to Corollary 2.2? If so, please modify the text so as to not cause confusion.
- Consider adding runtime units in the captions of Tables 9 and 10, as Table 11's already does.
- Consider plotting the MSE and upper bound in Figure 1 using a log-scale in the y-axis.

---

> ### Author Response · Authors · 2026-07-16
>
> Due to large sizes of tables and figures, we put all tables and figures in an anonymous repository: [https://anonymous.4open.science/r/TMLR_Review-BFE7/TMLR_Response.pdf](https://anonymous.4open.science/r/TMLR_Review-BFE7/TMLR_Response.pdf).
>
> We thank the reviewer for their review and positive feedback about the contributions of the manuscript. We aim at addressing their remaining concerns below.
>
> # Theoretical assumptions on data.
>
> Note first that the synthetic benchmarks perfectly match all the theoretical assumptions, which is the reason why we could empirically validate the upper bounds derived on the MSE and regret. There are two independence assumptions pertaining to cross-feature independance. Independence (14) (Independences (13) and (14) are the same, we will merge them in the next revision of the manuscript) is needed to prove that the variable $\hat{x}^f_j - (x^\star)^f_i$ where $i \neq j$ is a subgaussian variable only in the MAR setting. Independence (15) is needed for applying the Bernstein-related concentration bound for sub-exponential sums. We can relax Independence (15) in two ways. The first approach, in the absence of specific structures, leads to a bound linear in the number of features $F$, trading weaker assumptions for a worse bound. The proof goes as follows (starting after Lemma F.4 in the manuscript and thus replacing the Bernstein-like bound):
>
> for any $i,j \leq N$ and $f \leq F$,
>
> $$||\boldsymbol{x}\_{i}-\hat{\boldsymbol{x}}\_j \|\_2 \leq \sqrt{F} \max\_{f \leq F} |x^f\_{i}-\hat{x}^f\_j|\;,$$
>
> by considering the maximum value over a sum. Then, by using the previous inequality, then an union bound and finally the definition of each $x^f_{i}-\hat{x}^f_j$ being a zero-mean $\sigma^\text{miss}$-subgaussian variable,
>
> $$ \forall t, \text{Prob}( \|\boldsymbol{x}\_{i}-\hat{\boldsymbol{x}}\_j \|_2 \geq t) \leq 2F\exp(-t^2/(2F(\sigma^\text{miss})^2))\;. $$
>
> However, we can't do much better as a general rule, see for instance the case where all coordinates of $\boldsymbol{x}^f\_{i}-\hat{\boldsymbol{x}}\_j$ have the same value. If a more restrictive assumption is allowed, we can do better. The second approach assumes that all coordinates of $\boldsymbol{x}^f\_{i}-\hat{\boldsymbol{x}}\_j$ are bounded by some quantity $R$ in addition to being zero-mean $\sigma^\text{miss}$-subgaussian (but not independent). That is, we are under event $E = \{ \forall i,j \leq N, \forall f \leq F, |x^f\_{i}-\hat{x}^f\_j| \leq R\}$. Note that, by union bound and definition of the coordinates being zero-mean $\sigma^\text{miss}$-subgaussian, the probability of this event is
>
> $$\text{Prob}((E)^c) \leq 2F\exp(-R^2/(2(\sigma^\text{miss})^2))\;.$$
>
> For $R \leq \sigma^\text{miss}\sqrt{2\ln(2F)}$, event $E$ holds almost surely. Having bounded values is realistic: real-life measurements might belong to plausible ranges, and by the principle of kNN imputation, the imputed values would never go outside those ranges. Then we aim at proving that under event $E$, the vector $\boldsymbol{x}\_{i}-\hat{\boldsymbol{x}}\_j$ is zero-mean subgaussian, and then we will apply the definition of being subgaussian vector to obtain a better concentration bound. $\boldsymbol{x}_{i}-\hat{\boldsymbol{x}}_j$ being subgaussian means that, for all $\boldsymbol{u}$ such that $\|u\|\_2=1$, $\boldsymbol{u}^\top (\boldsymbol{x}\_{i}-\hat{\boldsymbol{x}}\_j)$ is zero-mean subgaussian. For any such $\boldsymbol{u}$,
>
> $$ \boldsymbol{u}^\top (\boldsymbol{x}\_{i}-\hat{\boldsymbol{x}}\_j) = \sum\_{f \leq F} u^f (x^f\_{i}-\hat{x}^f\_j) \leq \|\boldsymbol{u}\|\_1 R \leq \sqrt{F} \|u\|\_2 R = R\sqrt{F} \;, $$
>
> using Jensen's inequality. Moreover, $E[ \boldsymbol{u}^\top (\boldsymbol{x}\_{i}-\hat{\boldsymbol{x}}\_j)] = \sum_{f \leq F} u^fE[ x^f\_{i}-\hat{x}^f\_j] = 0$. Hence $\boldsymbol{u}^\top (\boldsymbol{x}\_{i}-\hat{\boldsymbol{x}}\_j)$ is a bounded, centered variable, and we can apply Hoeffding's lemma
>
> $$ \forall \lambda, E(\exp(\lambda \boldsymbol{u}^\top (\boldsymbol{x}\_{i}-\hat{\boldsymbol{x}}\_j))) \leq \exp(\lambda^2 (2 R\sqrt{F})^2/8) = \exp(\lambda^2 R^2F/2)\;,$$
>
> so $\boldsymbol{u}^\top (\boldsymbol{x}\_{i}-\hat{\boldsymbol{x}}\_j)$ is a zero-mean $R\sqrt{F}$-subgaussian variable. So $\boldsymbol{x}\_{i}-\hat{\boldsymbol{x}}\_j$ is a zero-mean $R\sqrt{F}$-subgaussian vector. Then we can use Proposition 6.2.1 from [1] to conclude that
>
> $$ \exists C > 0, \forall t \geq 0, \text{Prob}( \|\boldsymbol{x}\_{i}-\hat{\boldsymbol{x}}\_j \|\_2 \geq t \mid E) \leq \exp(-(t-CRF)^2/(C^2R^2F))\;. $$
>
> Using $R = \sigma^\text{miss}\sqrt{2\ln(2F)}$ and hence $\text{Prob}(E)=1$,
>
> $$ \exists C > 0, \forall t \geq 0, \text{Prob}( \|\boldsymbol{x}\_{i}-\hat{\boldsymbol{x}}\_j \|\_2 \geq t) \leq \exp(-(t-C\sigma^\text{miss}\sqrt{2\ln(2F)}F)^2/(2\ln(2F)F(C\sigma^\text{miss})^2))\;. $$

---

> > ### Author Response · Authors · 2026-07-16
> >
> > All in all, it is possible to get rid of the assumptions on cross-feature independence in the MCAR and MNAR settings, leading to a bound linear in $F$ without assumptions about structure, and to a better bound with a boundedness assumption on the feature values.
> >
> > [1] Vershynin, R. (2018). High-dimensional probability: An introduction with applications in data science (Vol. 47). Cambridge university press.
> >
> > # Practical implementation of F3I.
> > Indeed, $G(\boldsymbol{\alpha}, X)$ is computed by running by fitting a kernel density estimator for each newly imputed data set. In practice, runtime for F3I remains below 1 second per run for synthetic data sets up to $1,000$ samples and $5,000$ features in synthetic data sets.
> >
> > # About experimental conclusions.
> > We make the following claims in the paper: (1) (weighted) kNN algorithms are able to compete with the state-of-the-art; (2) introducing F3I allows us to extend kNN algorithms to perform joint imputation-downsteam tasks and mixed-type imputation, going beyond classical kNN algorithms. We do not claim that F3I is always competitive runtime/performance-wise with respect to the state-of-the-art (including kNN algorithms) for (non mixed-type) imputation tasks, as noted in Table 10. As for the Diabetes and HeartDisease datasets in Table 9, note that the fastest non-NN imputation algorithm, which is MIRACLE in both cases, is up to $30$ times slower, while being almost $16\%$ worse RMSE-wise. The reason why F3I might take more time on some data sets is when the early stopping criterion is not fulfilled, leading F3I to fully exhaust its iteration budget.
> > # Minor concerns.
> > Indeed, Lemma 2.2 is actually Corollary 2.2, we will fix this in a future revision. We will also mention that the runtime in Tables 9 and 10 is in seconds. We will plot the MSE upper bound in Figure 1 in log-scale.
> >
> > ## Q1. Benchmarking another downstream task.
> > The PCGradF3I algorithm and related theoretical analysis can be applied without changes to regression, provided that the considered regression loss function satisfies the required assumptions. We wrote results for the $R^2$ score and MSE values on the scikit-learn Diabetes dataset in Table 21 by replacing the softmax layer in the MLP architecture by a linear layer in our implementation. This example shows that our approach can be readily applied to regression tasks. However, we decided to restrict our study to classification for the sake of clarity.
> >
> > ## Q2. About theoretical guarantees for categorical-type data.
> > Since the imputation model is no longer continuous in $\boldsymbol{\alpha}$, the regret bound for the AdaHedge learner do not hold anymore. An interesting future work would either extend our bounds to the mixed-type imputation case, or propose a theoretically-backed algorithm that enables mixed-type imputation.
> >
> > ## Q3. About kNNSampler (Pashmchi et al., 2025).
> > We thank the reviewer for bringing this paper to our attention. The principle of kNNSampler is different from NN algorithms considered in our submission, as the imputed value is sampled from the observed feature values across the K nearest neighbors instead of being a convex sum of all observed feature values across those neighbors. A variant of kNNSampler can be considered a part of the weighted kNN family introduced in our submission, by defining the weight function as one that randomly chooses a weight vector in $\{(p\_1, p\_2, \dots, p\_K) \in \{0,1\}^K \mid \sum\_{i} p\_i = 1 \text{ and } \exists i \leq K, p\_i=1 \}$ (see their Algorithm 1). However, note that the setting is slightly different from ours, as the kNNSampler paper assumes the existence of $n$ samples without any missing values (their Equation (1)) from which the set of $K$ nearest neighbors is computed, and it does not consider a MNAR setting. Moreover, the authors of the kNNSampler also propose a theoretical analysis of their algorithm in terms of convergence of distributions, rather than in terms of pointwise error as we do, and evaluate the performance of imputation differently from the rest of the literature on imputation in terms of data sets, baselines and metrics (see the state-of-the-art that we compared against in our experiments). However, their results could be adapted to our framework of weighted kNN algorithms to derive convergence properties at distributional level, provided we considered their setting instead of ours.

---

### Review · Reviewer_AMjb · 2026-07-08

**Summary Of Contributions:**

The paper studies nearest-neighbor methods for missing-value imputation. It has three main components. First, it argues empirically that simple nearest-neighbor imputers are competitive with recent imputation methods while being much faster. Second, it proposes a general weighted nearest-neighbor imputation algorithm and claims high-probability MSE bounds under MCAR, MAR, and MNAR mechanisms. Third, it introduces F3I, which learns global nearest-neighbor weights through a KDE-based density-ratio objective, and extends this idea to downstream classification and mixed-type data.

The main strength is empirical. The experiments suggest that nearest-neighbor imputation is a strong and efficient baseline. This is useful, since simple baselines are often under-emphasized in missing-data benchmarks.

The main weakness is theoretical. The central concentration bound does not justify the treatment of adaptive nearest-neighbor donors as fixed independent Gaussian variables. The main MSE theorem applies this bound beyond its proved scope. The regret analysis also uses an invalid squared-distance inequality. As a result, the claimed theoretical guarantees are not established.

**Additional Comments:**

I recommend rejection.

The paper contains a potentially useful empirical message: nearest-neighbor imputation is a strong and fast baseline. However, the paper presents theoretical guarantees as a main contribution, and the main guarantees are not established. In particular, the concentration argument does not handle adaptive nearest-neighbor selection, Theorem 2.3 applies that argument beyond its valid scope, and the regret analysis uses an invalid squared-distance inequality.

These are substantive issues with the central claims, not presentation issues. In its present form, I do not think the paper should be accepted.

**Audience:**

Yes

**Audience Explanation:**

The empirical finding is potentially useful. The paper gives evidence that simple nearest-neighbor imputers can be competitive with more complex imputation methods while being much faster. This message is relevant to practitioners and researchers working on tabular missing-data problems.

However, the interest is mainly in this empirical observation. The current theoretical framing is too strong. The experiments do not clearly isolate the added value of F3I over standard kNN, and the theoretical guarantees are not established.

**Claims And Evidence:**

No

**Claims Explanation:**

The empirical evidence supports a weaker claim: nearest-neighbor imputers are often fast and competitive. It does not clearly support the stronger claim that F3I, and in particular its KDE-based weight-learning mechanism, gives a substantial improvement over standard uniform or distance-weighted kNN.

More importantly, the theoretical claims are not supported by the proofs. The first three issues below are decision-critical.

1. Technical Lemma 3 uses an unjustified distributional identity.
In Eq. (16), the proof states
$$
(\widehat x_i^f\mid m_i^f=1)
\sim
N!\left(\mu_f,\sigma^2\sum_{k=1}^K\alpha_k^2\right).
$$
This would be true if the donor indices were fixed and the donor values were independent Gaussian variables. In Algorithm 1, however, the donors are selected by a nearest-neighbor search on the same data. In F3I, the weights $\alpha_k$ are also learned from the imputed data. The donor values are therefore adaptive, not fixed independent draws. The subsequent concentration argument also relies on independence across imputed coordinates, which is not established.

2. Theorem 2.3 applies Corollary 2.2 outside its scope.
After Jensen's inequality, the proof contains terms of the form
$$
{1\over F}\sum_{k=1}^K\alpha_k
\sum_{f^f=1}
\left(
\widehat x^f_{K(\widehat x_i,X,f,k)}-(x_i^\star)^f
\right)^2 .
$$
The donor index depends on both $f$ and $k$. The proof rewrites this as a norm of a pseudo-neighbor vector and applies Corollary 2.2, but the required pseudo-vectors are not valid outputs covered by that corollary. The argument for later iterations has a further gap, because later imputed matrices are functions of previous imputations, updated neighbor searches, and learned weights.

3. The regret proof uses a false squared-distance inequality.
In Theorem D.1, the proof uses
$$
|x-x_i^\star|_2^2-|x-\widehat x_i|_2^2
\leq
|\widehat x_i-x_i^\star|_2^2 .
$$
This is false. For example, in one dimension with $x=R$, $x_i^\star=0$, and $\widehat x_i=\varepsilon$, the left-hand side is $2R\varepsilon-\varepsilon^2$, which can be much larger than $\varepsilon^2$. A valid inequality must include a term depending on the evaluation point $x$.

There are additional problems. The Gaussian assumption and the deterministic bounded-norm assumption are incompatible without conditioning on a high-probability event. The MNAR proof does not correctly condition on the observed donor pool. The KDE objective is high-dimensional and its computational cost is not clearly included in the complexity analysis. The concavity proof for $G$ is incomplete. Lemma E.1 has a sign error in the claimed Lipschitz constant.

For these reasons, the main theoretical guarantees are not convincing.

**Requested Changes:**

The issues below are not minor revisions. They affect the main theoretical claims of the paper. In my view, they cannot be adequately addressed by clarification or local fixes in the present submission. A substantially different paper, with new proofs and weaker claims, would be needed.

1. The main concentration argument would need to be replaced. Technical Lemma 3 treats nearest-neighbor donor values as fixed independent Gaussian samples. This is not valid for data-dependent nearest-neighbor selection, and even less so for F3I, where the weights are learned from the imputed data.

2. Theorem 2.3 would need a new proof. The present proof applies Corollary 2.2 to feature-wise pseudo-neighbors and to later iterative outputs of Algorithm 1. These objects are not covered by the concentration result.

3. The regret analysis would need to be replaced. Theorem D.1 uses a false squared-distance inequality. A correct argument would require additional assumptions and would change the bound.

4. The assumptions would need to be reformulated. The paper combines non-degenerate Gaussian data with a deterministic bounded-norm assumption. These are incompatible without conditioning on a high-probability event or changing the data model.

5. The MNAR part would need a new argument. Under Gaussian self-masking, the observed donor pool is conditioned or reweighted. The present proof does not account for this conditioning together with nearest-neighbor selection.

6. The role of the high-dimensional KDE objective would need to be reconsidered. Its bandwidth choice and computational cost are central to F3I, but are not clearly justified. Direct evaluation appears to have $O(N^2F)$ cost.

7. The experimental claims would need to be weakened. The results support the usefulness of nearest-neighbor imputation as a fast baseline. They do not clearly establish that F3I improves substantially over uniform or distance-weighted kNN.

For these reasons, I do not view the paper as suitable for acceptance after ordinary revision. The authors may be able to develop a future submission by refocusing the paper as an empirical study of nearest-neighbor imputation, or by providing substantially new theory for a restricted setting. But the present theoretical framing is not supported.

---

> ### Author Response · Authors · 2026-07-16
>
> We thank the reviewer for their detailed review about our submission. We aim at addressing their concerns below.
>
> # About experimental conclusions.
> We make the following claims in the paper: (1) (weighted) kNN algorithms are able to compete with the state-of-the-art; (2) introducing F3I allows us to extend kNN algorithms to perform joint imputation-downsteam tasks and mixed-type imputation, going beyond classical kNN algorithms; (3) learning weights with the F3I procedure leads to weights that differ from classical kNN algorithms. We do not claim that F3I is always competitive performance-wise with respect to other kNN algorithms on imputation tasks.
>
> # About Technical Lemma 3 and Corollary 2.2 (points 1 and 2 of the review).
>
> We thank the reviewer for pointing out these issues in our proofs. We are still working on resolving these problems.
>
> # About the triangular inequality (point 3 of the review).
> Thank you for catching this error. The correct triangular inequality is
> $$ \forall x \in R^F, \forall i \leq N, \|x-(x^\star)\|^2\_2 - \|x-\hat{x}\_i\|^2\_2 \leq 2\|x-\hat{x}\_i\|\_2 \sqrt{C^\text{miss}\_{1/N^3}} + C^\text{miss}\_{1/N^3}\;, $$
> with probability $1-1/N$. We are still working on solving the subsequent issue in the regret proof.
>
> # Conditioning on a high-probability event for the Gaussian assumption and deterministic bounded-norm assumption
>  Under Assumption B.4, one can condition on event $E^\star = \{\forall i \leq N, \|\boldsymbol{x}\_i\|\_2 \leq S'\}$ where $S' >0$ (on $E^\star$, $\|\boldsymbol{x}\_i\|^2\_2 \leq (S')^2 = S$) of probability greater than $1-\delta$. $\delta$ is such that
> $$ \text{Prob}((E^\star)^c) = \text{Prob}(\bigcup\_{i \leq N} \|\boldsymbol{x}\_i\|\_2 \geq S') \leq \delta\;.$$
> Note that $\sigma^{-1}\|\boldsymbol{x}\_i\|^2\_2$ is distributed according to the noncentral chi-squared distribution with F degrees of freedom and with noncentrality parameter $\lambda = \|\boldsymbol{\mu}\|^2\_2$. Thus for all $t<1/2$, $E[\exp(t\|\sigma^{-1}\boldsymbol{x}\_i\|^2\_2)] = \exp(-F/2\log(1-2t)+\frac{t\|\boldsymbol{\mu}\|^2\_2}{1-2t})$. Using a Markov inequality, for all $t<1/2$,
> $$\text{Prob}(\|\boldsymbol{x}\_i\|\_2 \geq S') = \text{Prob}(\exp(t\sigma^{-1}\|\boldsymbol{x}\_i\|^2\_2) \geq \exp(t\sigma^{-1}(S')^2)) \leq \exp(-t\sigma^{-1}(S')^2)E[\exp(t\|\sigma^{-1}\boldsymbol{x}\_i\|^2\_2)$$
> $$\text{Prob}(\|\boldsymbol{x}\_i\|\_2 \geq S') \geq \exp(-t\sigma^{-1}(S')^2-F/2\log(1-2t)+\frac{t\|\boldsymbol{\mu}\|^2\_2}{1-2t})\;.$$
> Then by a union bound, for all $S'>0$,
> $$ \text{Prob}((E^\star)^c) \leq \sum\_{i \leq N} \text{Prob}(\|\boldsymbol{x}\_i\|\_2 \geq S) \leq \delta := \inf\_{0<t<1/2} \exp(\log(N)-t\sigma^{-1}(S')^2-F/2\log(1-2t)+\frac{t\|\boldsymbol{\mu}\|^2\_2}{1-2t}) \;.$$
> We will add this term to all probability terms in our proofs.
> # Incomplete concavity proof for G and justification of the bandwidth
>  To show that G is (strictly) concave, it is enough to show that the Hessian matrix $\nabla^2\_{\boldsymbol{\alpha}} G(\boldsymbol{\alpha}, X)$ is negative semi-definite (or definite). In the proof of Proposition C.2, we show that there exists a bandwidth value $h\_0 > 0$ such that for all bandwidth $h > h\_0$,  the Hessian matrix $\nabla^2\_{\boldsymbol{\alpha}} G(\boldsymbol{\alpha}, X)$ is negative definite, assuming $\eta < 4KN$.
>
> # Sign error in Lemma E.1.
> The final constant is indeed $H(1-\beta)+L\beta\sqrt{KS}$.
> # Computational cost of the KDE objective on highly-dimensional data.
> It is correct that, for exact estimations of the kernel density (for instance, with k-d-trees, which we implemented as we faced datasets with small to moderate dimensions), the computational cost can be up to $\mathcal{O}(N^2F)$. However, there are many approximate kernel density estimation techniques, including Nystr\"{o}m or Random Fourier Features approximations, where the computation cost is in $\mathcal{O}(NFr)$, where $r$ is the rank of the approximation. For extremely large datasets (both in $N$ and $F$), this step could also be replaced by learned densities with normalizing flows. We will mention this in our revision.

---

### Note · Authors · 2026-07-21

**Comment:**

We agree with Reviewer AMjb's assessment and would like to withdraw the manuscript so that we can address the issues identified in the proofs. We sincerely thank the reviewers for their careful evaluation of our work and their valuable feedback.

**Withdrawal Confirmation:**

I have read and agree with the venue's withdrawal policy on behalf of myself and my co-authors.